# Certifying Deep Network Risks and Individual Predictions with PAC-Bayes Loss via Localized Priors

**Wen Dong**[*]
Air Force Research Laboratory,
Wright–Patterson AFB, OH 45433
wendong@gmail.com

## Abstract

As machine learning increasingly relies on large, opaque foundation models powering generative and agentic AI, deploying these systems in safety-critical contexts demands rigorous generalization guarantees beyond training data. PAC-Bayes theory provides principled certificates linking training performance to generalization risk, yet existing approaches remain impractical: simple theoretical priors yield vacuous bounds, while data-dependent priors require costly second-stage training or introduce bias. To bridge this critical gap, we propose a *localized PAC-Bayes prior*—a structured, computationally efficient prior softly concentrated around parameters favored during standard training. By integrating this localized prior directly into the standard training objective, we deliver practically tight generalization certificates with minimal workflow disruption. Under standard neural tangent kernel assumptions, our bound shrinks as networks widen and datasets grow, becoming negligible in realistic regimes. Empirically, we demonstrate tight generalization certificates on tasks ranging from image classification (MNIST, CIFAR, ImageNet) and NLP fine-tuning (GLUE) to semantic segmentation (Cityscapes), typically within three percentage points of test error at ImageNet scale. Additionally, our approach provides rigorous guarantees for individual predictions, selective rejection of uncertain predictions, adversarial robustness, and accurate calibration—directly addressing key requirements for trustworthy AI deployment.

## 1 Introduction

Large-scale neural networks now read chest X-rays, flag fraudulent transactions, steer driver-assistance systems, and complete sentences for hundreds of millions of users. A single held-out accuracy number no longer suffices for regulators and practitioners deploying these models in safety-critical contexts. Aviation (EU's *AI Roadmap 2.0*), medical (U.S. FDA's 2025 draft guidance), and automotive (ISO PAS 8800:2024) standards all explicitly require rigorous *learning-assurance arguments* that bound the model's post-deployment error rate. Industry responses—such as robustness checks provided by major tool builders and responsible AI practices promoted by leading companies—address only local or point-wise guarantees, leaving open the distribution-level risk question central to real-world safety and trustworthiness.

Probably Approximately Correct (PAC) learning theory quantifies generalization rigorously, yet classical PAC complexity measures (VC dimension, Rademacher complexity) yield overly conservative bounds for modern, heavily over-parameterized networks. PAC-Bayes, introduced by McAllester [1, 2] and further developed by Langford [3, 4] and Catoni [5, 6], sharpens this analysis by comparing the *learned parameter distribution* $\rho$ with a reference distribution $\pi$. The gap between training and test loss reduces to an information term $\mathrm{KL}(\rho\|\pi)$. However, naive PAC-Bayes

---

[*]Affiliation provided for identification only; work performed in personal capacity on personal time.

39th Conference on Neural Information Processing Systems (NeurIPS 2025).

priors produce vacuous bounds on large-scale architectures: the resulting KL divergence can reach thousands of nats, rendering the method practically unusable.

We address this fundamental limitation by employing a *localized prior*, an approach inspired by Catoni's original work [6] and subsequent extensions [7, 8]. Specifically, our localized prior is defined as $\pi_{\text{loc}}(\theta) \propto \pi(\theta) \exp[-\xi \lambda r_S(\theta)]$, with $0 < \xi < 1$. Here, $\pi(\theta)$ is the original data-independent prior, $r_S(\theta)$ is the empirical loss, and the parameters $\lambda, \xi$ control the strength of the shift toward promising parameters. This prior closely mirrors the ideal Gibbs posterior distribution $\rho(\theta) \propto \pi(\theta) \exp[-\lambda r_S(\theta)]$, which emerges naturally as the limit case when $\xi = 1$. Thus, the factor $\xi < 1$ introduces controlled softening, preventing overfitting by keeping the prior more dispersed than the fully empirical Gibbs posterior. Optimizing $\xi$ and $\lambda$ as part of standard SGD seamlessly integrates PAC-Bayes regularization into training, yielding tight and practically meaningful generalization bounds for modern neural networks.

Empirically, our localized PAC-Bayes bound integrates seamlessly into standard training by directly replacing traditional loss objectives with PAC-Bayes-based counterparts. On benchmarks ranging from classical image classification (MNIST, CIFAR-10/100, ImageNet) to modern tasks such as Cityscapes semantic segmentation and GLUE NLP fine-tuning, our method consistently provides rigorous, tight, and meaningful certificates. The certification overhead is minimal, comparable to adding just one training epoch, while offering reliable individual-level guarantees, selective prediction strategies, and robust adversarial input detection.

Theoretically, we confirm that our bound converges favorably with increasing data and network size, establishing clear relationships between localization parameters, network width, and sample size. Informally stated, under standard scaling conditions, the KL term shrinks linearly in network width and inversely with the sample size, vanishing entirely in infinite-width limits (Theorem 3.2).

In summary, we (i) introduce a localized, trajectory-aware PAC-Bayes bound, transforming PAC-Bayes from theoretical curiosity to practical diagnostic, (ii) demonstrate straightforward, effective integration into real-world deep-learning pipelines, and (iii) validate our approach across diverse vision, language, and control tasks, aligning closely with emerging regulatory requirements.

## 2 PAC-Bayes Preliminaries and Notation

Machine-learning papers typically report empirical training loss, yet practical deployment demands guarantees on unseen data. The difference between these defines a *deviation event*. Formally, given a dataset $S = \{(x_i, y_i)\}_{i=1}^{N}$ drawn from an unknown distribution $D$, a predictor parameterized by $\theta$, and a per-example loss $\ell(\theta; x_i, y_i) \in [0, 1]$, we denote the empirical loss by $r_S(\theta) = \frac{1}{N} \sum_i \ell(\theta; x_i, y_i)$ and the population (true) loss by $R(\theta) = \mathbf{E}_{(x,y) \sim D}[\ell(\theta; x, y)]$. Our central question: how improbable is the event $R(\theta) - r_S(\theta) \gg 0$?

The simplest way to bound such deviations uses Markov's inequality, controlling the tail of a nonnegative random variable $X$ via its expectation: $\mathbf{P}[X > k\, \mathbf{E}X] \leq 1/k$. To sharpen control, we transform deviations through a monotone mapping—specifically, exponentiation—to amplify large deviations. Explicitly, we rewrite the deviation probability using exponentiation and then apply Markov:

$$\mathbf{P}[r_S(\theta) - R(\theta) > \varepsilon] = \mathbf{P}[e^{\lambda N(r_S(\theta) - R(\theta))} > e^{\lambda N \varepsilon}] \leq e^{-\lambda N \varepsilon}\, \mathbf{E}[e^{\lambda N(r_S(\theta) - R(\theta))}].$$

Union bounds and classical complexity measures (e.g., VC-dimension, Rademacher complexity [9]) extend these guarantees from single predictors to finite hypothesis sets. However, classical PAC bounds falter when faced with large, complex hypothesis spaces—such as deep neural networks, explicitly parameterized by many-layered compositions of affine transformations and nonlinear activations. Classical PAC bounds require clipping or softening losses to lie within $[0, 1]$, yet even after we clipped/softened and scaled the cross entropy loss to be within $[0, 1]$ to satisfy these artificial constraints, traditional combinational or covering-number arguments rapidly explode due to the massive complexity inherent in deep architectures.

PAC-Bayes circumvents these issues by considering distributions over predictors rather than finite sets. The central technical tool is the Donsker-Varadhan variational identity, which provides a sharp

variational bound on exponential moments under a reference distribution $\pi$:

$$\log \mathbf{E}_\pi \left[ e^{f(\theta)} \right] = \sup_\rho \left\{ \mathbf{E}_\rho[f(\theta)] - \mathrm{KL}(\rho \| \pi) \right\}.$$

Choosing $f(\theta) = \lambda(r_S(\theta) - R(\theta))$, and applying Jensen's inequality to push expectation inside, the exponential tail bound above becomes:

$$\mathbf{P}_S \left( \mathbf{E}_\rho \left[ r_S(\theta) - R(\theta) \right] > \varepsilon \right) = \mathbf{P}_S \left( e^{\lambda \mathbf{E}\rho[r_S(\theta) - R(\theta)]} > e^{\lambda \varepsilon} \right) \leq e^{-\lambda N \varepsilon} \mathbf{E}_S \left[ e^{\lambda \mathbf{E}_\rho[r_S(\theta) - R(\theta)]} \right]$$

$$\overset{\text{(Jensen)}}{\leq} e^{-\lambda \varepsilon} \mathbf{E}_{S,\rho} \left[ e^{\lambda(r_S(\theta) - R(\theta))} \right] \overset{\text{(DV)}}{\leq} e^{-\lambda \varepsilon} \exp \left( \sup_\rho \left\{ \mathbf{E}_\rho[\lambda(r_S(\theta) - R(\theta))] - \mathrm{KL}(\rho | \pi) \right\} \right).$$

Setting this probability equal to $\delta$, we finally obtain the simplified PAC-Bayes inequality [10, 6], explicitly linking good generalization to small divergence of the posterior $\rho$ from a fixed prior $\pi$: with high probability $1 - \delta$,

$$\mathbf{E}_\rho[R(\theta)] \leq \mathbf{E}_\rho[r_S(\theta)] + \frac{\mathrm{KL}(\rho \| \pi) + \log(1/\delta)}{\lambda}. \tag{1}$$

Original PAC-Bayes results [1, 3] instead chose $f(\theta) = \mathrm{kl}(r_S(\theta) | R(\theta))$ to directly bound KL-divergences between empirical and population risks, but evaluating such KL divergences is challenging. Thus, the linearized Hoeffding-based form above is preferred in practice.

Still, PAC-Bayes bounds remain challenging for deep networks, since naïve priors (like isotropic Gaussians) differ substantially from actual SGD-trained parameters, inflating the KL term dramatically and producing trivial guarantees. Recent efforts include complex data-dependent priors—weight compression [11], variational priors learned through auxiliary optimization [8], and differential-privacy priors [12]—each effective but computationally expensive.

A simple way to view Catoni's localization [6] is as the PAC-Bayes analogue of classical capacity-near-the-data ideas: SVM margin/SRM reasoning [9], local Rademacher complexities that shrink around empirical minimizers [13, 14], and variance-aware (empirical-Bernstein) concentration [15]. In PAC-Bayes, we implement this by exponential tilting $\pi_*(\theta) \propto \pi(\theta) \exp\left[-\xi \lambda r_S(\theta)\right]$: this is the same mechanism as Gibbs/exponential weights in aggregation [16] and is handled analytically via the Donsker–Varadhan change of measure, which turns the tilt into a transparent compensation term $\log \pi \left[ \exp\left(-\xi \lambda r_S\right) \right]$ added to the usual $\mathrm{KL}(\rho \| \pi)$. The temperature $\xi \in (0,1)$ softly interpolates between the base prior ($\xi \to 0$) and the empirical Gibbs posterior ($\xi \to 1$), and the bound tracks this interpolation with only a mild $(1 - \xi)^{-1}$ factor. This perspective both connects localization to standard non-PAC-Bayes tools and reassures that the "extra strength" from data dependence is explicitly accounted for by a log-partition term rather than hidden in heuristics [17, 4, 18, 7, 8].

Insights from Neural Tangent Kernel theory [19] suggest the KL divergence of the SGD trajectory from initialization accumulates gradually, with small incremental increases per step—implying that total KL cost shrinks as network width grows.

A more detailed and extended review of PAC-Bayes theory and related derivations can be found in Appendix C. In the next section, we rigorously formalize these localized PAC-Bayes ideas, developing a practical generalization certificate validated experimentally at ImageNet scale.

## 3 Empirical PAC-Bayes Bound for Neural Networks

The following theorem guarantees—with high probability—a bound on a neural network's true (population) error based solely on the training data, sampled network parameters, and explicit statistical penalties. Specifically, our bound decomposes into four intuitive components: (1) empirical risk estimated via sampling from a learned posterior distribution, (2) penalties due to Monte Carlo sampling uncertainty, (3) complexity measured as the divergence between the learned posterior and a tilted data-dependent prior (similar to variational inference objectives), and (4) normalization terms ensuring uniform statistical validity.

Let $\rho(\theta)$ denote a posterior distribution over neural network parameters, which we optimize directly with respect to the empirical loss $r_S(\theta)$ and, in practice, its mini-batch proxy $\hat{r}_m(\theta)$ (Component 1). To measure complexity, we use a tilted prior distribution (Component 3): $\pi_{\exp[-\xi \lambda r_S(\theta)]}(\theta) \propto \pi(\theta) \exp[-\xi \lambda r_S(\theta)]$, constructed by softly aligning a fixed, data-independent reference distribution

$\pi(\theta)$ toward low-loss regions of parameter space, moderated by scaling parameters $0 < \xi < \alpha < 1$ and inverse-temperature parameter $\lambda$.

We estimate the empirical loss and the log-moment under the prior via independent samples $\theta_i \sim \rho$ and $\tilde{\theta}_j \sim \pi$:

$$\widehat{\rho[\hat{r}_m]} = \frac{1}{K} \sum_{i=1}^{K} \hat{r}_m(\theta_i), \quad \pi \widehat{[e^{-\xi\lambda\hat{r}_m}]} = \frac{1}{M} \sum_{j=1}^{M} e^{-\xi\lambda\hat{r}_m(\tilde{\theta}_j)}. \tag{2}$$

These finite-sample estimates introduce explicit uncertainty penalties into the bound (Component 2). The parameter $\epsilon \in (0, 1)$ represents our total confidence budget, split among different rare events. The parameters $\alpha$ and $\zeta$ specify the granularity of our discretization, enabling optimization of parameters $\xi$ and $\lambda$ to minimize the PAC-Bayes bound.

**Theorem 3.1** (Empirical PAC-Bayes Bound). *For any confidence budget $\epsilon \in (0, 1)$, localization parameters $0 < \xi < \alpha < 1$, scaling parameter $\lambda \in [1, 2N]$, and discretization parameter $\zeta \in (\xi, 1)$, let $\kappa = g(\lambda/(\zeta N))$. With probability at least $1 - \epsilon$:*

$$\mathbf{E}_\rho[R(\theta)] \leq \left(1 - \frac{\xi}{\alpha}\right)^{-1} \left(1 - \frac{\alpha+\xi}{\alpha-\xi}\kappa\frac{\lambda}{\zeta N}\right)^{-1} \left\{\widehat{\rho[\hat{r}_m]} + \frac{1}{\lambda}\left(\mathrm{KL}(\rho\|\pi) + \log\pi\widehat{[e^{-\xi\lambda\hat{r}_m}]}\right) + \delta\right\}, \tag{3}$$

*where* $\delta = \sqrt{\frac{\log(6|\Lambda||\Xi|/\epsilon)}{2K}} + (1+\xi)\sqrt{\frac{\log(6|\Lambda||\Xi|/\epsilon)}{2m}} + \frac{1}{\lambda}\log\left(1 + \frac{(1-e^{-\xi\lambda})}{\pi\widehat{[e^{-\xi\lambda\hat{r}_m}]}}\sqrt{\frac{\log(6|\Lambda||\Xi|/\epsilon)}{2M}}\right) + \frac{1+\xi/\alpha}{\lambda}\log\frac{2|\Lambda||\Xi|}{\epsilon}.$

*Here $g(x) = \frac{e^x - x - 1}{x^2}$, and $|\Lambda|, |\Xi|$ are the sizes of the $\lambda$- and $\xi$-grids. The denominator satisfies $0 < 1 - \frac{\alpha+\xi}{\alpha-\xi}g(\frac{\lambda}{\zeta N})\frac{\lambda}{\zeta N} \leq 1$.*

*Proof sketch.* The proof starts from Catoni's localized learning lemma: for any posterior $\rho$, tilting a base prior $\pi$ by $\exp(-\xi\lambda r_S)$ yields a high-probability inequality that upper-bounds $\rho[R]$ by a combination of $\rho[r_S]$, a divergence term, and a small curvature penalty coming from a Bernstein mgf control; algebraically expanding $\mathrm{KL}(\rho\|\pi_{\exp}(-\xi\lambda r_S)) = \mathrm{KL}(\rho\|\pi) + \xi\lambda\rho[r_S] + \log\mathbf{E}_\pi[e^{-\xi\lambda r_S}]$ exposes exactly the three quantities we estimate in practice (empirical risk, structural complexity, and the localization normalizer). To make $(\lambda, \xi)$ tunable by SGD without data-splitting, we first uniformize over discrete grids for $\lambda$ and $\xi$ (union bound) and then relax back to continuous values using the nearest grid points, which introduces the transparent contraction factors in the denominator and the small grid-size logs in the numerator (this is the source of the $(1 - \xi/\alpha)^{-1}$ and the $g(\lambda/(\zeta N))$ term). Finally, each expectation is replaced by a Monte-Carlo proxy with a one-sided concentration correction that preserves the direction of the bound: we use Hoeffding for $\widehat{\rho[\hat{r}_m]}$, and a prior-sampling upper confidence bound for $\log\mathbf{E}_\pi[e^{-\xi\lambda r_S}]$ (monotonicity of log keeps it one-sided), while mini-batching replaces $r_S$ by $\hat{r}_m$ with an added $(1 + \xi)$ radius. the resulting bound is tight because it automatically adapts to concentrate around models with low training loss (localization) and adjusts to the variance of the loss (curvature), avoiding overly conservative estimates. Moreover, it provides a practical drop-in objective, resembling standard training with just a well-calibrated regularization term, making it easy to optimize via standard gradient-based methods. The formal statement and complete proof of Theorem 3.1 can be found in Appendix A.1. $\square$

Our PAC-Bayes bound explicitly avoids the vacuousness problem common in prior PAC-Bayes analyses of deep networks and reduces generalization analysis to standard variational inference (VI) — or, in the special case of a point posterior, maximum a posteriori (MAP) learning — leveraging familiar optimization methods with interpretable guarantees. Specifically, the empirical risk and KL complexity terms jointly form a VI objective: $\frac{\lambda\mathbf{E}_\rho[r_S(\theta)] + \mathrm{KL}\left(\rho\|\pi_{\exp[-\xi\lambda r_S(\theta)]}\right)}{\lambda(1-\xi)}$. Its optimal posterior is exactly the Gibbs distribution given explicitly by the Donsker–Varadhan formula: $\rho(\theta) \propto \pi(\theta)\exp(-\lambda r_S(\theta))$. At optimality, the objective equals $\frac{\log Z_{\xi\lambda} - \log Z_\lambda}{\lambda(1-\xi)}$, where $Z_{\beta\lambda} = \int \pi(\theta)\exp(-\beta\lambda r_S(\theta))\,d\theta$. For risks bounded in $[0, 1]$, this quantity naturally lies in $[0, 1]$ (since $e^{-(1-\xi)\lambda} \leq Z_\lambda/Z_{\xi\lambda} \leq 1$), ensuring the bound is inherently non-vacuous. In fact, even the data-independent variational principle remains non-vacuous at optimality, though it is less optimal than the data-dependent version: $\inf_\rho\left\{\mathbf{E}_\rho[r_S(\theta)] + \frac{1}{\lambda}\mathrm{KL}(\rho\|\pi)\right\} = -\frac{1}{\lambda}\log\mathbf{E}_{\theta\sim\pi}[e^{-\lambda r_S(\theta)}] \in [0, 1]$, since $\mathbf{E}_{\theta\sim\pi}[e^{-\lambda r_S(\theta)}] \in [e^{-\lambda}, 1]$. Its optimal distribution coincides exactly with the Gibbs posterior of the data-dependent case (with the same inverse temperature $\lambda$). Additional penalty terms explicitly account for confidence and finite-sample estimation effects.

The variance-sensitive denominator $0 < 1 - \frac{\alpha+\xi}{\alpha-\xi} g\left(\frac{\lambda}{\zeta N}\right) \frac{\lambda}{\zeta N} \leq 1$, structured as a "linear term minus variance-driven penalty," arises naturally from the Bernstein-type inequality in our PAC-Bayes bound. A smaller denominator indicates greater uncertainty in the empirical risk estimate, thereby enforcing stronger regularization to avoid overly confident posterior solutions. Hyperparameters $\alpha$ and $\zeta$ discretize the optimization space for $\lambda$ and $\xi$, allowing optimization while preventing overfitting. Because $g(x)$ is almost linear when its argument is small, the combined term $g(\frac{\lambda}{\zeta N})\frac{\lambda}{\zeta N}$ collapses to roughly $\frac{1}{2}\frac{\lambda}{\zeta N}$ whenever the denominator is kept near 1. Allowing that denominator to fall by at most $\varepsilon$ from one then leads to the easy rule of thumb $\frac{\lambda}{\zeta N} \leq \frac{2\epsilon(\alpha-\xi)}{\alpha+\xi}$.

The previous theorem provides a rigorous yet computationally challenging generalization guarantee. To operationalize this result practically, we propose a computationally efficient stochastic gradient descent (SGD) algorithm with partitioned posterior sampling, detailed in Algorithm 1. At each iteration, the algorithm partitions the current mini-batch into $K$ sub-batches and independently samples network parameters for each partition. This parallelized approach yields stable Monte Carlo estimates of empirical posterior risk $\widehat{\rho[\hat{r}_m]}$, the structural term $\text{KL}(\rho|\pi)$, and the localization normalizer $\log \mathbf{E}_\pi[e^{-\xi\lambda r_S}]$ (via the one-sided prior-sampling bound used in Eq. 3), without needing to buffer historical parameter snapshots across iterations. Consequently, this approach significantly enhances computational stability, reduces memory overhead, and improves scalability.

The computational overhead per iteration remains modest (typically $1.1\times$–$1.3\times$ a standard SGD step), scaling sub-linearly with the partition count $K$ when shards are fused/batched across the sample axis. Empirically, choosing a moderate partition size (e.g., $K \in [4, 8]$) provides an effective balance between statistical accuracy and computational cost. The optimization jointly updates the posterior parameters and key bound parameters $(\lambda, \xi)$, with hyperparameters $(\alpha, \zeta)$ controlling the discretization granularity used in the uniformization step (cf. Eq. 3), facilitating efficient optimization. Overall, the proposed algorithm integrates seamlessly into standard neural network training workflows, effectively balancing theoretical rigor with practical usability and providing computationally efficient and theoretically sound generalization guarantees.

---

**Algorithm 1:** Empirical PAC-Bayes SGD with Partitioned Posterior Sampling

---

**Input:** Data mini-batches $B_t = (X_t, Y_t)$ for $t = 1{:}T$, initial parameters $(\phi, \lambda, \xi)$, prior $\pi$, partition/posterior-sample count $K$, prior MC size $M$, learning rate $\eta$, discretization hyperparameters $\alpha, \zeta$ and confidence level $\epsilon$.

**Output:** Final parameters $(\phi, \lambda, \xi)$ and PAC-Bayes bound (Eq. 3).

**Initialize:** Posterior distribution $\rho_\phi$.

**for** $t = 1, \ldots, T$ **do**

    Split mini-batch $B_t$ into $K$ shards of size **$m = |B_t|/K$** (reuse the same $B_t$ for all estimators this step).

    `/* Posterior MC on shards (vectorizable/fused across a sample axis) */`

    Draw $\theta_1, \ldots, \theta_K \sim \rho_\phi$ and evaluate shard risks to obtain $\widehat{\rho[\hat{r}_m]}$ (Setup);

    From the same $\{\theta_k\}$ and layer log-densities, compute $\widehat{\text{KL}(\rho\|\pi)}$ (Setup);

    `/* Prior MC for the localization log-moment (no grads to φ)       */`

    Draw $\tilde{\theta}_1, \ldots, \tilde{\theta}_M \sim \pi$ on $B_t$ and compute $\pi\widehat{[e^{-\xi\lambda\tilde{r}_m}]}$ (Setup); use this value (and its concentration radius from Eq. 3) only in the prior term and $\delta$, with `stop_gradient` w.r.t. $\phi$ (grads to $\lambda, \xi$ allowed).

    Form the per-step objective $\mathcal{L}_t$ as the RHS of Eq. 3 with plug-in estimators $\widehat{\rho[\hat{r}_m]}$, $\widehat{\text{KL}(\rho\|\pi)}$, $\log\pi\widehat{[e^{-\xi\lambda\tilde{r}_m}]}$, and $\delta$ (using $K, M, m, \alpha, \zeta, \epsilon$).

    `/* Update (project to feasible set to keep denominator positive)   */`

    $(\phi, \lambda, \xi) \leftarrow (\phi, \lambda, \xi) - \eta\, \nabla_{(\phi,\lambda,\xi)} \mathcal{L}_t$; project $(\lambda, \xi)$ onto $\{ 1 - \frac{\xi}{\alpha} > 0,\ 1 - \frac{\alpha+\xi}{\alpha-\xi} g(\frac{\lambda}{\zeta N})\frac{\lambda}{\zeta N} > 0 \}$.

*Notes:* (i) Fuse the $K$ posterior samples across a leading "sample" axis to keep GEMMs efficient; (ii) use small $M$ (e.g., 8–16) since prior MC is forward-only; (iii) typical overhead is $1.1\times$–$1.3\times$ a standard step when $K \in [4, 8]$; (iv) clamp log arguments as usual for numerical safety and clip $\pi\widehat{[e^{-\xi\lambda\tilde{r}_m}]}$ away from 0 using the same floor as in Appendix A.1.

Given a PAC-Bayes bound on population risk $\mathbf{E}_\rho[R(\theta)] \leq \hat{\mathcal{B}}$, we can turn it into a **per-prediction guarantee** using Markov's inequality and a union bound. For any input $x \sim \mathcal{D}$, the loss of a posterior-sampled or averaged model satisfies:

$$\mathbf{P}_{x \sim \mathcal{D}} \left[ \ell(f(x), y) > \hat{\mathcal{B}}/\epsilon \right] \leq \epsilon. \tag{4}$$

This allows the user to **flag or reject high-risk predictions** during deployment, while guaranteeing that such rejections are rare—assuming deployment data matches training distribution. It's particularly useful for **adversarial filtering**, **data drift detection**, and **model monitoring**, where spikes in rejection rates indicate mismatch or failure. No retraining is required; the certificate is derived directly from the posterior and the bound.

Machine learning practitioners often follow standard training protocols—such as regularized stochastic gradient descent (SGD)—and observe consistent generalization performance across replications. This empirical stability raises a fundamental question: **how does regularization contribute to generalization, and can we provide a formal guarantee grounded in PAC-Bayes theory?** To address this, we model neural network training as a stochastic differential equation (SDE) defined by its initialization and SGD update schema. Since marginalizing this SDE to compute the posterior density $\rho_t(\theta)$ is intractable, we approximate it by constructing an ordinary differential equation (ODE) whose marginal distribution matches that of the SDE in expectation. This approximation is derived by applying the Fokker-Planck equation and assuming a diagonal structure for the posterior variance, with Hutchinson's estimator used to approximate the trace of the Hessian. The resulting log-posterior can be updated via a simple recurrence:

$$\log \rho_{t+1}(\theta) = \log \rho_t(\theta) - \eta_t \cdot \left( \mathrm{Tr}(H_t) + \tfrac{1}{\lambda} \|\theta_t\|^2 \right),$$

where $\eta_t$ is the learning rate at step $t$, $\lambda$ is the L2 regularization coefficient, and $H_t$ is the generalized local curvature matrix given by: $H_t = \nabla^2 \mathcal{L}(\theta_t) + \frac{1}{\lambda} I$.

In practice, we estimate $\mathrm{Tr}(H_t)$ using Hutchinson's method with Rademacher noise: sample a random vector $v \sim \{\pm 1\}^d$, compute the Hessian-vector product $H_t v$ via nested autodiff, and estimate the trace as $v^\top H_t v$. This fast, biased update rule tracks the contraction of log-density along the training trajectory, and when inserted into the PAC-Bayes bound (Theorem 3.1), yields nonvacuous generalization guarantees—even without explicitly minimizing the bound.

While we showed deep networks trained with SGD can yield nonvacuous empirical PAC-Bayes generalization guarantees, our deeper question is: can neural networks generalize in the highly overparameterized regimes where classical complexity theory would suggest overfitting, and if so, why and in what sense? The answer lies in the structure imposed by modern training: stochastic optimization from random initialization induces a posterior that is not arbitrary, but aligned with low-risk, low-complexity regions. The next result formalizes this idea. It shows that, in wide networks trained under standard conditions, the PAC-Bayes generalization gap converges exponentially at rate $\mathcal{O}(1/N + 1/n)$, where $n$ is network width and $N$ is training data size, and the learned distribution remains close to a Gibbs posterior over a tractable linear approximation.

Specifically, consider a neural network of width $n$, trained via stochastic gradient descent (SGD) with step size $\eta = \eta_0/n$ from a Neural Tangent Kernel-style initialization (scaled such that activations remain stable as network width grows). To ensure controlled convergence, assume input norms are bounded or bounded with high probability, activation functions have bounded first- and second-order derivatives, and the initial empirical kernel is positive definite with eigenvalues uniformly bounded away from zero and infinity. Under these mild assumptions typically met by standard deep networks, we state informally the following PAC-Bayes generalization guarantee. A formal statement and proof appear in the appendix.

**Theorem 3.2** (PAC-Bayes Bound for Wide Networks, Informal). *Under these assumptions, with probability at least $1 - \varepsilon$, the expected population risk under the posterior distribution induced by SGD satisfies:*

$$\pi_{-\lambda\,r}[R(\theta)] \;\leq\; \frac{\left( \lambda + \kappa\frac{\lambda^2}{N} - \beta \right) \rho[\widetilde{R}(\theta)] + \frac{C_{\mathrm{KL}}}{n} + \log\frac{2}{\varepsilon}}{\lambda - \beta - \kappa\frac{\lambda^2}{N}}.$$

Here $C_{\mathrm{KL}}/n$ is the oracle penalty from replacing the nonlinear network by its linearization; it scales as $O(1/n)$ because the per-step Gaussian KL between the two SGD increment laws is *quadratic* in their $O(n^{-1/2})$ lazy-training drift.

*Proof sketch.* To establish a tight generalization guarantee for wide neural networks, we first approximate the complicated nonlinear training dynamics by a simpler linearized network at initialization. Under standard NTK (neural tangent kernel) conditions—bounded activations and derivatives, stable NTK, and appropriate learning rate scaling—standard theory ensures that as the network becomes wider (larger width $n$), the original nonlinear training and its linearized approximation remain very close, differing only by small $O(n^{-1/2})$ amounts in their predictions and kernels. We then model each stochastic gradient descent (SGD) update as a step drawn from a Gaussian distribution, using known results from the neural network literature. Since the KL divergence between two Gaussian distributions grows quadratically with the difference in their parameters, these small $O(n^{-1/2})$ differences translate into even smaller $O(1/n)$ KL terms per step. More precisely, if one-step SGD updates are modeled as Gaussians with mean/covariance perturbations $\Delta\mu_t, \Delta\Sigma_t$ between the nonlinear and linearized dynamics, a second-order expansion around equality gives $\mathrm{KL}_t = \frac{1}{2}\Delta\mu_t^\top \Sigma_t^{-1}\Delta\mu_t + \frac{1}{4}\left\|\Sigma_t^{-1/2}\Delta\Sigma_t\Sigma_t^{-1/2}\right\|_F^2 + o\left(\|\Delta\mu_t\|^2 + \|\Delta\Sigma_t\|^2\right)$; with $\|\Delta\mu_t\|, \|\Delta\Sigma_t\| = O\left(n^{-1/2}\right)$, this yields per-step $\mathrm{KL}_t = O(1/n)$. Because the prediction error shrinks quickly during training, summing these small KL terms over the entire trajectory remains just $O(1/n)$. Finally, by plugging this linearized approximation into our localized PAC-Bayes bound (Theorem 3.1), we obtain a generalization guarantee that shrinks significantly as the network width increases, providing a rigorous yet practical certificate that the trained network's true error remains close to its empirical training error. For full details, see Appendix A.2. $\qquad\square$

# 4  Experiments

In this section, we present experiments to assess the behavior of the proposed localized PAC-Bayes method. We first examine performance on standard benchmark datasets and models commonly studied in PAC-Bayes literature. Subsequently, we extend the evaluation to more recent tasks in deep learning, illustrating practical integration of the PAC-Bayes approach into modern workflows. Detailed experimental configurations and hyperparameters are provided in Appendix B.

## 4.1  Classic Benchmarks and Localization Behavior

We begin our empirical evaluation by demonstrating the effectiveness, interpretability, and practical tightness of our localized PAC-Bayes bound on classic benchmarks. We explicitly compare our localized method against both traditional data-independent and recent data-dependent PAC-Bayes approaches, clearly differentiating between one-pass and two-pass frameworks.

Our evaluation spans widely-used datasets such as MNIST [20], CIFAR-10/100 [21], and ImageNet [22], utilizing common network architectures including fully connected networks (FCN), LeNet-5 [23], ResNet-50 [24], Wide ResNet (WRN-28-10) [25], DenseNet-121 [26], and EfficientNet-B0 [27]. We contrast several PAC-Bayes bounds: classical data-independent bounds [17], two-pass data-dependent bounds using weight compression [28], Fisher-information-based methods [29], and our proposed localized PAC-Bayes method, which efficiently constructs a data-informed tilted prior. Results are summarized in Table 1.

Table 1: Empirical evaluation of PAC-Bayes bounds across datasets and architectures. Columns show dataset, model architecture, empirical error (Train/Test), and PAC-Bayes bounds categorized by method type: (1) Classical (data-independent, one-pass), (2) Compression (data-dependent, two-pass), (3) Fisher Information (data-dependent, two-pass), and (4) Localized (ours, data-dependent, one-pass). Empirical errors based on best predictor (posterior mean or sampled).

| Dataset | Architecture | Error (Tn/Tst) | Classical | Compression | Fisher Info | Localized (ours) |
|---------|--------------|----------------|-----------|-------------|-------------|------------------|
| MNIST | FCN | 0.1% / 1.2% | 5.5% | 2.5% | 2.1% | **1.9% (0.3%)** |
| MNIST | LeNet-5 | 0.9% / 1.1% | 4.2% | 2.0% | 1.7% | **1.5% (0.2%)** |
| CIFAR-10 | ResNet-50 | 8.0% / 9.5% | 45.0% | 20.0% | 18.5% | **11.5% (1.5%)** |
| CIFAR-10 | WRN-28-10 | 7.0% / 8.0% | 39.0% | 18.0% | 17.0% | **9.5% (1.0%)** |
| CIFAR-100 | DenseNet-121 | 25.5% / 27.0% | Vacuous | 44.0% | 44.5% | **29.5% (2.0%)** |
| CIFAR-100 | EfficientNet-B0 | 25.5% / 26.5% | Vacuous | 45.0% | 44.0% | **27.0% (2.0%)** |
| ImageNet | ResNet-50 | 25.0% / 27.0% | Vacuous | 47.0% | 46.0% | **31.0% (2.5%)** |
| ImageNet | EfficientNet-B0 | 31.0% / 32.5% | Vacuous | 50.0% | 48.0% | **35.5% (2.5%)** |

Figure 1 visualizes training dynamics for CIFAR-10 using WRN-28-10, highlighting how empirical metrics (accuracy, loss) and PAC-Bayes bounds evolve over training epochs. The localization parameters ($\lambda, \xi$) exhibit intuitive convergence behavior, indicating seamless integration into conventional SGD workflows. Specifically, $\lambda$ stabilizes at an effective regularization strength, while $\xi$ decreases gradually to a minimal yet meaningful level.

Selecting moderate localization parameters (e.g., $\lambda = 2500$, $\xi = 0.01$) effectively uses information from approximately 25 data points. This minimal yet strategic localization substantially tightens the PAC-Bayes bound without risking overfitting, illustrating a balanced and practically beneficial approach.

## 4.2 Extending PAC-Bayes to Modern Deep Learning Tasks

Having demonstrated the effectiveness of our localized PAC-Bayes bound on classical benchmarks, we now illustrate its practical application in modern deep learning scenarios. We highlight how our method seamlessly integrates PAC-Bayes certification into standard workflows by directly substituting traditional training objectives (such as cross-entropy) with PAC-Bayes-based loss objectives.

We begin with the Cityscapes semantic segmentation task [30], which is vital for applications like autonomous driving. Semantic segmentation naturally fits PAC-Bayes certification because of its bounded per-pixel classification loss. We use a lightweight U-Net with a ResNet backbone, trained on standard TensorFlow Datasets, and evaluate performance via Intersection-over-Union (IoU). Our PAC-Bayes certification is seamlessly incorporated into the training process by probabilistically modeling convolutional and fully connected layers, leaving task-specific segmentation heads deterministic. Table 2 confirms that our method achieves tight, practical certification bounds, making it valuable for deployment in safety-critical applications.

Concurrently, we showcase our PAC-Bayes certification on NLP fine-tuning tasks from the GLUE benchmark [31]: MRPC (paraphrase classification), SST-2 (sentiment classification), and RTE (textual entailment). These tasks highlight the growing reliance on fine-tuning foundation models with limited labeled data, where conventional validation metrics fall short in providing formal safety guarantees. We employ LoRA [32] fine-tuning on GPT-2-small, explicitly modeling low-rank adapter parameters as Gibbs distributions via reparameterization. By directly replacing standard fine-tuning losses with our PAC-Bayes bounds, we achieve rigorous generalization guarantees suitable for reliable NLP deployments.

We summarize experimental results in Table 2.

Table 2: Empirical results demonstrating that our localized PAC-Bayes bounds yield tighter, practically meaningful generalization guarantees compared to two-pass PAC-Bayes methods across diverse tasks. Columns show task/dataset, architecture/model, empirical loss (Train/Val), and PAC-Bayes loss bounds: Two-pass and Localized (ours, one-pass).

| Task/Dataset | Model | Loss (Train/Val) | Two-pass | Localized (ours) |
|---|---|---|---|---|
| Cityscapes | U-Net (ResNet) | 0.22 / 0.25 | 0.30 | **0.27** |
| MRPC | GPT-2 + LoRA | 0.15 / 0.14 | 0.16 | **0.15** |
| SST-2 | GPT-2 + LoRA | 0.08 / 0.11 | 0.14 | **0.14** |
| RTE | GPT-2 + LoRA | 0.22 / 0.25 | 0.28 | **0.26** |

We compare our localized PAC-Bayes bound against a two-pass PAC-Bayes approach that first trains a model conventionally and then separately computes a data-dependent prior. The experimental results summarized in Table 2 highlight our method's advantages clearly. Our localized PAC-Bayes bounds consistently provide tighter and practically meaningful generalization guarantees compared to the two-pass PAC-Bayes approaches. These findings underscore our method's ability to bridge theoretical rigor and real-world practicality effectively.

## 4.3 Certifying Individual Predictions and Adversarial Robustness

Our PAC-Bayes bound allows us to set a precise risk-based threshold for individual predictions derived directly from our certified model, utilizing Markov's inequality and a union-bound argument

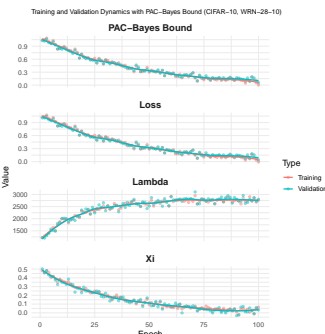 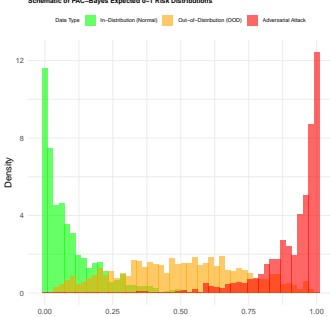 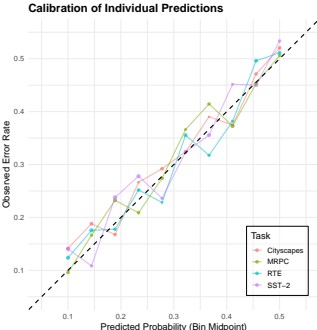

Figure 1: Training dynamics and PAC-Bayes bound evolution (CIFAR-10, WRN-28-10).

Figure 2: PAC-Bayes 0–1 risk distributions for normal, OOD, and adversarial data (PGD attack).

Figure 3: Calibration of individual predictions across various tasks.

(Eq. 4). Predictions exceeding this threshold can be reliably flagged or discarded, ensuring we retain only trustworthy predictions while tightly controlling the rejection rate.

We empirically demonstrate this selective prediction approach by examining risk distributions for correctly classified versus misclassified test samples. Figure 2 shows PAC-Bayes 0–1 risk distributions for normal, out-of-distribution (OOD; primarily using ImageNet-32×32 [33, 34]), and adversarial (PGD attack [35]) examples. Clearly, adversarially perturbed samples exhibit substantially higher risk values compared to normal and OOD samples, highlighting our method's practical capability to robustly detect and reject adversarial inputs.

Additionally, we evaluate the calibration [36] of our PAC-Bayes-certified predictions (Figure 3). Across diverse practical tasks (Cityscapes, MRPC, RTE, SST-2), our models exhibit excellent calibration, meaning that predicted uncertainties closely match observed error frequencies. This accurate calibration underscores the reliability and interpretability of PAC-Bayes-certified predictions, ensuring the selective rejection mechanism is both trustworthy and effective in real-world deployments.

Together, these results demonstrate the practical strength of our PAC-Bayes framework—providing rigorous individual-level guarantees, robust detection of adversarial attacks, and well-calibrated predictive uncertainties essential for dependable model deployment.

## 5 Related Work

PAC-Bayes theory provides a rigorous probabilistic framework for deriving generalization guarantees, overcoming many limitations of classical VC-dimension and Rademacher complexity-based analyses [17, 37]. Classical PAC-Bayes bounds often use data-independent priors, resulting in empirically vacuous or excessively loose guarantees when applied to modern deep neural networks [37, 38, 39]. Conversely, theoretical advancements, such as localization techniques developed by Catoni [6], provide mathematically elegant and tighter bounds, but often remain disconnected from practical machine learning workflows.

Recent works aim to bridge this gap by introducing data-dependent priors or surrogate optimization methods. For instance, weight-compression approaches improve tightness significantly by exploiting sparsity or compressibility of neural networks [29, 28]. Nevertheless, these methods typically require multiple optimization stages or additional computational overhead, limiting their practical adoption [40, 38].

Similarly, Fisher information-based and Neural Tangent Kernel (NTK)-based PAC-Bayes methods further tighten generalization guarantees through approximations of the local geometry of loss landscapes [41, 19]. These methods generally rely on strong assumptions (e.g., wide-network limits or linear approximations), thus restricting their applicability in diverse or realistic training scenarios.

Other contemporary efforts operationalize PAC-Bayes bounds within specialized contexts such as reinforcement learning, conformal prediction, and meta-learning [42, 43, 44]. However, these

methods often remain computationally intensive, niche-specific, or challenging to integrate directly into conventional deep learning training loops.

Our work explicitly bridges this gap by introducing a localized PAC-Bayes bound that naturally integrates theoretical rigor (inspired by techniques such as Catoni's localization) into standard deep learning workflows. This approach avoids multiple optimization passes and excessive computational costs, offering practically meaningful and empirically tight generalization guarantees directly applicable to modern neural networks.

## 6 Limitations and Broader Impacts

**Limitations.** Our localized PAC-Bayes certification relies on Monte Carlo estimation (with reparameterization) of the prior log-moment $\log \mathbf{E}_\pi \left[ e^{-\xi \lambda r_S(\theta)} \right]$ in Theorem 3.1 and, when evaluating the Gibbs-minimized form, to approximate structured losses as differences between log-partition functions, i.e., $\frac{1}{\lambda}(\log \mathbf{E}\pi[\exp(-\xi \lambda r_S(\theta))] - \log \mathbf{E}\pi[\exp(-\lambda r_S(\theta))])$. While statistically consistent, using finite samples may introduce a negative bias due to Jensen's inequality, potentially risking overfitting. Accordingly, our bound uses one-sided, high-probability upper surrogates for $\log \mathbf{E}_\pi \left[ e^{-\xi \lambda r_S(\theta)} \right]$ (see Eq. 3 via $\delta$) to avoid optimistic bias, at the cost of extra conservativeness. A more accurate but computationally intensive alternative integrates posterior expectations continuously over inverse temperatures, i.e., $\int_\xi^1 \mathbf{E}_{\theta \sim \pi_{\exp(-\beta \lambda r_S(\theta))}}[r_S(\theta)] \, d\beta$, by sampling from intermediate Gibbs posteriors $\pi_{\exp(-\beta \lambda r_S)}$. This multi-temperature estimator reduces logarithmic bias but increases sampling cost. Thus, our chosen two-endpoint method represents a pragmatic balance between accuracy and computational efficiency. Additionally, efficient GPU parallelization is critical; careless implementation—such as sequential rather than batched forward passes—can significantly degrade computational performance.

**Broader Impacts.** Our work improves transparency and trustworthiness of neural network predictions, particularly valuable for large foundation models and generative or agentic AI deployed in safety-critical and regulated domains, such as healthcare and autonomous driving. Rigorous PAC-Bayes certification enables responsible deployment, facilitating regulatory compliance and safer adoption of advanced AI systems. However, as PAC-Bayes certificates inherently depend on assumptions—such as representative training data and accuracy of practical approximations used in implementation—misinterpreting these assumptions can lead to misuse and false confidence. Clear communication about these assumptions, limitations, and proper interpretation guidelines is critical to responsibly leverage the benefits of our approach while mitigating potential negative consequences.

## 7 Conclusions

We have developed a practical PAC-Bayes certification method that addresses key computational and conceptual obstacles previously limiting PAC-Bayes applicability to modern neural networks. Our localized prior approach enables tight certification of generalization risks at both network and individual-prediction levels without significantly altering standard training workflows. Empirical results demonstrate strong certification performance across image classification, NLP fine-tuning, and semantic segmentation, closely aligning theoretical guarantees with real-world applications and regulatory needs. This advancement transforms PAC-Bayes theory into a practical tool for reliably certifying deep learning models, substantially improving trustworthiness and transparency in safety-critical AI deployments.

## Disclaimer

The views expressed are those of the authors and do not reflect the official guidance or position of the United States Government, the Department of Defense or of the United States Air Force. The content or appearance of hyperlinks does not reflect an official DoD, Air Force, Air Force Research Laboratory position or endorsement of the external websites, or the information, products, or services contained therein. This work was conducted in the authors' personal capacity, on personal time, and without the use of employer or U.S. Government resources.

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

# A Proofs and Formal Statements

This section gives formal statements and complete proofs of Theorems 3.1 and 3.2, with a focus on the theory and all notation and assumptions made explicit.

## A.1 Formal Statement and Proof of Theorem 3.1

Before formally stating Theorem 3.1, we briefly review the essential notation to ensure clarity and readability. We denote by $\Theta$ a measurable parameter space. The true data-generating distribution over the input-output pairs $(x, y)$ is denoted by $P$, from which we have a training dataset $S = \{(x_i, y_i)\}_{i=1}^N$, drawn independently and identically from $P$.

Given parameters $\theta \in \Theta$, we consider a bounded loss function $\ell(\theta; x, y)$, satisfying $0 \le \ell(\theta; x, y) \le 1$ for all $\theta, x, y$. We define the population risk $R(\theta) = \mathbf{E}_{(x,y) \sim P}[\ell(\theta; x, y)]$, and the empirical risk $r_S(\theta) = \frac{1}{N} \sum_{i=1}^N \ell(\theta; x_i, y_i)$.

We consider distributions over the parameter space $\Theta$, with the prior distribution denoted by $\pi(\theta)$, and the posterior distribution denoted by $\rho(\theta)$. The Kullback–Leibler (KL) divergence between these two distributions is given by $\mathrm{KL}(\rho, \pi) = \int_\Theta \rho(\theta) \log \frac{\rho(\theta)}{\pi(\theta)} \, d\theta$.

To refine generalization guarantees practically, we introduce the concept of a *localized prior*, defined as a data-dependent prior distribution: $\pi_{\exp[-\xi\lambda r_S(\theta)]}(\theta) \propto \pi(\theta) \exp[-\xi\lambda r_S(\theta)]$. Here, the parameters $\lambda > 0$ and $0 < \xi < 1$ control the concentration of this prior around empirically promising parameter regions. We utilize a Bernstein-type concentration inequality, which controls the bound using second-order and bounded higher-order moments via the auxiliary function $g(x) = (e^x - x - 1)/x^2$.

We introduce two discretization parameters, $\alpha \in (\xi, 1)$ and $\zeta \in (\xi, 1)$, to define auxiliary grids that evenly distribute the allowed error probability (uniformization): $\Xi = \{\alpha^k : 1 \le k \le \lfloor \log_{\alpha^{-1}}(N) \rfloor\}$, $\Lambda = \{2N\zeta^k : 0 \le k \le \lfloor \log_{\zeta^{-1}}(2N) \rfloor\}$. We then slightly relax this discretization, allowing direct continuous optimization of $\lambda$ and $\xi$ via gradient-based methods.

We define the empirical posterior risk estimate $\widehat{\rho[r_S]}$ as the Monte Carlo estimate of the empirical risk under the posterior $\rho$, using $K$ i.i.d. parameter samples $\theta_i \sim \rho$ :

$$\widehat{\rho[r_S]} = \frac{1}{K} \sum_{i=1}^K r_S(\theta_i). \tag{2'}$$

We estimate the log-partition term $\log \mathbf{E}_\pi\left[e^{-\xi\lambda r_S(\theta)}\right]$ with $M$ i.i.d. samples $\tilde{\theta}_j \sim \pi$ :

$$\pi\widehat{[e^{-\xi\lambda r_S}]} = \frac{1}{M} \sum_{j=1}^M e^{-\xi\lambda r_S(\tilde{\theta}_j)}. \tag{2''}$$

In practice, we approximate the full-dataset risk $r_S(\theta)$ by the mini-batch risk $\hat{r}_m(\theta)$, the average loss on a mini-batch of size $m$ (drawn without replacement or as an i.i.d. approximation), and we use the corresponding substitutions $\widehat{\rho[\hat{r}_m]}$ and $\pi\widehat{[e^{-\xi\lambda\hat{r}_m}]}$ in the estimators above.

The following theorem bounds the neural network's true risk explicitly in terms of empirical risk and complexity estimates defined above.

**Theorem 3.1** (Empirical PAC-Bayes Bound). *Let $\kappa = g(\lambda/(\zeta N))$, under the previously defined conditions on localization parameters $\xi, \alpha, \zeta$, the confidence parameter $\epsilon \in (0, 1)$, and regularization parameter $\lambda \in [1, 2N]$, with the denominator satisfying $0 < 1 - \frac{\alpha + \xi}{\alpha - \xi} g\left(\frac{\lambda}{\zeta N}\right) \frac{\lambda}{\zeta N} \le 1$, the following bound on the expected population risk under the posterior distribution $\rho(\theta)$ holds with probability at least $1 - \epsilon$:*

$$\mathbf{E}_\rho[R(\theta)] \le \left(1 - \frac{\xi}{\alpha}\right)^{-1} \left(1 - \frac{\alpha + \xi}{\alpha - \xi} \kappa \frac{\lambda}{\zeta N}\right)^{-1} \left\{\widehat{\rho[\hat{r}_m]} + \frac{1}{\lambda}\left(\mathrm{KL}(\rho\|\pi) + \log \pi\widehat{[e^{-\xi\lambda\hat{r}_m}]}\right) + \delta\right\}, \tag{3}$$

*where* $\delta = \sqrt{\frac{\log(6|\Lambda||\Xi|/\epsilon)}{2K}} + (1+\xi)\sqrt{\frac{\log(6|\Lambda||\Xi|/\epsilon)}{2m}} + \frac{1}{\lambda}\log\left(1 + \frac{(1-e^{-\xi\lambda})}{\pi\widehat{[e^{-\xi\lambda\hat{r}_m}]}}\sqrt{\frac{\log(6|\Lambda||\Xi|/\epsilon)}{2M}}\right) + \frac{1+\xi/\alpha}{\lambda}\log\frac{2|\Lambda||\Xi|}{\epsilon}.$

This theorem links training performance, sampling estimates, and statistical penalties to give a practical generalization guarantee. The proof follows four steps: (i) restating and proving Catoni's

learning lemma, (ii) introducing a localized prior, (iii) uniformizing $\lambda, \xi$ for optimization, and (iv) deriving practical estimators via importance sampling.

We begin by restating Catoni's learning lemma (Lemma A.1), foundational to constructing the localized PAC-Bayes bound (Theorem 3.1). This lemma introduces a confidence function $\eta(\theta)$ that quantifies our certainty regarding the parameter $\theta$ in minimizing the true risk, effectively localizing the small-event probability budget towards more promising parameters. It states that, with high probability over the training sample, no posterior distribution $\rho$ can significantly deviate from its empirical risk $\mathbf{E}_\rho[r(\theta)]$ beyond a confidence interval determined by the complexity term $\mathrm{KL}(\rho\|\pi)$ and the expected confidence $\mathbf{E}_\rho[\eta(\theta)]$.

**Lemma A.1** (Catoni, 2007[5])**.** *For any positive real parameter $\lambda > 0$, any measurable function $\eta : \Theta \to \mathbf{R}$, and any prior probability distribution $\pi$ on $\Theta$, we have:*

$$\mathbf{P}\left\{\sup_\rho \lambda\mathbf{E}_\rho[R(\theta)] - \lambda\mathbf{E}_\rho[r_S(\theta)] - \mathbf{E}_\rho[\eta(\theta)] - \mathrm{KL}(\rho\|\pi) \geq 0\right\} \tag{5}$$

$$\leq \mathbf{E}_{\theta\sim\pi}\left[\exp\left(\tfrac{\lambda^2}{N}g\left(\tfrac{\lambda}{N}\right)R(\theta)(1-R(\theta)) - \eta(\theta)\right)\right].$$

*Similarly,*

$$\mathbf{P}\left\{\sup_\rho \lambda\mathbf{E}_\rho[r_S(\theta)] - \lambda\mathbf{E}_\rho[R(\theta)] - \mathbf{E}_\rho[\eta(\theta)] - \mathrm{KL}(\rho\|\pi) \geq 0\right\} \tag{6}$$

$$\leq \mathbf{E}_{\theta\sim\pi}\left[\exp\left(\tfrac{\lambda^2}{N}g\left(\tfrac{\lambda}{N}\right)R(\theta)(1-R(\theta)) - \eta(\theta)\right)\right].$$

*Proof of Lemma A.1.* We begin with the following identity based on the Donsker–Varadhan variational representation of the KL divergence:

$$\sup_\rho \lambda\mathbf{E}_\rho[R(\theta) - r_S(\theta)] - \mathbf{E}_\rho[\eta(\theta)] - \mathrm{KL}(\rho\|\pi) = \log\left\{\mathbf{E}_{\theta\sim\pi}\left[\exp\left(\lambda[R(\theta) - r_S(\theta)] - \eta(\theta)\right)\right]\right\}.$$

Taking probability explicitly with respect to the random draw of the training set $S$, we have:

$$\mathbf{P}_S\left\{\sup_\rho \lambda\mathbf{E}_\rho[R(\theta) - r_S(\theta)] - \mathbf{E}_\rho[\eta(\theta)] - \mathrm{KL}(\rho\|\pi) \geq 0\right\}$$

$$=\mathbf{P}_S\left\{\mathbf{E}_{\theta\sim\pi}\left[\exp\left(\lambda[R(\theta) - r_S(\theta)] - \eta(\theta)\right)\right] \geq 1\right\}$$

$$\leq\mathbf{E}_S\left[\mathbf{E}_{\theta\sim\pi}\left[\exp\left(\lambda[R(\theta) - r_S(\theta)] - \eta(\theta)\right)\right]\right]$$

$$=\mathbf{E}_{\theta\sim\pi}\left[\mathbf{E}_S\left[\exp\left(\lambda[R(\theta) - r_S(\theta)] - \eta(\theta)\right)\right]\right]$$

$$\leq\mathbf{E}_{\theta\sim\pi}\left[\exp\left(\tfrac{\lambda^2}{N}g\left(\tfrac{\lambda}{N}\right)R(\theta)(1-R(\theta)) - \eta(\theta)\right)\right].$$

The first inequality follows from Markov's inequality applied to the random variable $\mathbf{E}_{\theta\sim\pi}\left[\exp\left(\lambda[R(\theta) - r_S(\theta)] - \eta(\theta)\right)\right]$. The subsequent equality follows by Fubini's theorem applied to the positive function $(\theta, S) \mapsto \exp\left(\lambda[R(\theta) - r_S(\theta)] - \eta(\theta)\right)$. The final inequality results from Bernstein's inequality applied to the random variables $\ell(\theta; x_i, y_i)$, where $g(x) = (e^x - x - 1)/x^2$ is a standard auxiliary function arising from Bernstein's inequality. This function explicitly provides refined control over the tail behavior through second-order moments and higher-order moment bounds.

For the reverse inequality, we similarly start from the Donsker–Varadhan variational representation of the KL divergence:

$$\sup_\rho \lambda\mathbf{E}_\rho[r_S(\theta) - R(\theta)] - \mathbf{E}_\rho[\eta(\theta)] - \mathrm{KL}(\rho\|\pi) = \log\left\{\mathbf{E}_{\theta\sim\pi}\left[\exp\left(\lambda[r_S(\theta) - R(\theta)] - \eta(\theta)\right)\right]\right\}.$$

Taking probability explicitly with respect to the dataset $S$, we have:

$$\mathbf{P}_S\left\{\sup_\rho \lambda\mathbf{E}_\rho[r_S(\theta) - R(\theta)] - \mathbf{E}_\rho[\eta(\theta)] - \mathrm{KL}(\rho\|\pi) \geq 0\right\}$$

$$=\mathbf{P}_S\left\{\mathbf{E}_{\theta\sim\pi}\left[\exp\left(\lambda[r_S(\theta) - R(\theta)] - \eta(\theta)\right)\right] \geq 1\right\}$$

$$\leq\mathbf{E}_S\left[\mathbf{E}_{\theta\sim\pi}\left[\exp\left(\lambda[r_S(\theta) - R(\theta)] - \eta(\theta)\right)\right]\right]$$

$$=\mathbf{E}_{\theta\sim\pi}\left[\mathbf{E}_S\left[\exp\left(\lambda[r_S(\theta) - R(\theta)] - \eta(\theta)\right)\right]\right]$$

$$\leq\mathbf{E}_{\theta\sim\pi}\left[\exp\left(\tfrac{\lambda^2}{N}g\left(\tfrac{\lambda}{N}\right)R(\theta)(1-R(\theta)) - \eta(\theta)\right)\right].$$

$\square$

We proceed to construct a PAC-Bayes bound with localized prior from Catoni's learning lemma A.1, following Catoni's idea of localization.

*Proof of Theorem 3.1.*

**Localizing the Prior Using Population Risk Information**. We begin by explicitly choosing a confidence function $\eta(\theta)$ as if we already had knowledge of the true risk $R(\theta)$:

$$\eta(\theta) = \frac{\lambda^2}{N}\kappa R(\theta) - \log \pi_{\exp(-\beta R(\bullet))}(\theta) + \log(\epsilon^{-1}), \text{ where } \kappa = g\left(\frac{\lambda}{N}\right).$$

The first term encodes our confidence from Bernstein's inequality that empirical risk closely approximates the population risk, simplifying the variance term $R(\theta)[1 - R(\theta)]$ to $R(\theta)$ for convenience. The second term, $-\log \pi_{\exp(-\beta R(\theta))}(\theta) = \beta R(\theta) + \log \pi[\exp(-\beta R(\theta))]$, differentially penalizes parameter regions based on their proximity to empirical optima: parameters far from the optimum incur a looser penalty, while those near the optimum experience tighter localization. The final term provides uniform confidence control. We define $\kappa = g\left(\frac{\lambda}{N}\right)$ for notational simplicity.

Substitute $\eta(\theta)$ into Eq. 5 in Lemma A.1, then relax the Bernstein bound $\frac{\lambda^2}{N}g\left(\frac{\lambda}{N}\right)R(\theta)[1 - R(\theta)]$ in the exponent by $\frac{\lambda^2}{N}g\left(\frac{\lambda}{N}\right)R(\theta)$ on the right-hand side. This relaxation effectively simplifies the expression involving $\eta(\theta)$, leaving only the term $\epsilon$ explicitly on the right-hand side, thus yielding

$$\mathbf{P}_S\left\{\sup_\rho\left[(\lambda-\beta-\kappa\frac{\lambda^2}{N})\mathbf{E}_\rho[R(\theta)]-\lambda\mathbf{E}_\rho[r(\theta)]-\mathrm{KL}(\rho\|\pi)-\log\{\pi[\exp(-\beta R(\theta))]\}\right] \geq \log(\epsilon^{-1})\right\} \leq \epsilon,$$

i.e., with probability at least $1 - \epsilon$, the following bound holds for all posterior distributions $\rho$:

$$\mathbf{E}_\rho[R(\theta)] \leq (\lambda - \beta - \kappa\frac{\lambda^2}{N})^{-1}\left\{\lambda\mathbf{E}_\rho[r(\theta)] + \mathrm{KL}(\rho\|\pi) + \log\{\pi[\exp(-\beta R(\theta))]\} + \log(\frac{1}{\epsilon})\right\}. \quad (7)$$

**Bounding the Log-Partition Function via Empirical Risk**. The log-partition term $\log\{\pi[\exp(-\beta R(\theta))]\}$ in the above PAC-Bayes inequality cannot be optimized directly since it involves the unknown population risk $R(\theta)$. To resolve this, we aim to replace it with a bound involving the known empirical risk $r_S(\theta)$. To achieve this, we start by applying the Donsker-Varadhan variational formula to explicitly express the term in question as an expectation under the Gibbs posterior $\pi_{\exp(-\beta R(\theta))}$:

$$\log\{\pi[\exp(-\beta R(\theta))]\} = -\beta\, \mathbf{E}_{\pi_{\exp(-\beta R(\theta))}}[R(\theta)] - \mathrm{KL}\big(\pi_{\exp(-\beta R(\theta))}\|\pi\big).$$

The unknown expectation term $\mathbf{E}_{\pi_{\exp(-\beta R(\theta))}}[R(\theta)]$ can be bounded in terms of the empirical risk using the inequality from Eq. 5 of Lemma A.1. With probability at least $1-\epsilon$, we have: $-\mathbf{E}_\rho[R(\theta)] \leq \frac{\lambda\,\mathbf{E}_\rho[r_S(\theta)]+\log(1/\epsilon)}{\lambda+\kappa\frac{\lambda^2}{N}}$, $\forall\rho$. Applying this inequality specifically to the Gibbs posterior $\pi_{\exp(-\beta R(\theta))}$, we obtain the following bound with probability at least $1 - \epsilon$:

$$-\beta\,\mathbf{E}_{\pi_{\exp(-\beta R(\theta))}}[R(\theta)] \leq \beta\,\mathbf{E}_{\pi_{\exp(-\beta R(\theta))}}\left[-\frac{\lambda}{\lambda+\kappa\frac{\lambda^2}{N}}\,r_S(\theta) + \frac{1}{\lambda+\kappa\frac{\lambda^2}{N}}\log\frac{1}{\epsilon}\right].$$

Substituting this result back into the expression for the log-partition function yields, with probability at least $1 - \epsilon$:

$$\log\{\pi[\exp(-\beta R(\theta))]\} = -\beta\,\mathbf{E}_{\pi_{\exp(-\beta R(\theta))}}[R(\theta)] - \mathrm{KL}\big(\pi_{\exp(-\beta R(\theta))}\|\pi\big) \quad (8)$$

$$\leq \beta\,\mathbf{E}_{\pi_{\exp(-\beta R(\theta))}}\left[-\frac{\lambda}{\lambda+\kappa\frac{\lambda^2}{N}}\,r_S(\theta) + \frac{1}{\lambda+\kappa\frac{\lambda^2}{N}}\log\frac{1}{\epsilon}\right] - \mathrm{KL}\big(\pi_{\exp(-\beta R(\theta))}\|\pi\big)$$

$$\leq \sup_\rho\left\{\frac{\beta}{1+\kappa\frac{\lambda}{N}}\left(-\mathbf{E}_\rho[r_S(\theta)] + \frac{\log(1/\epsilon)}{\lambda}\right) - \mathrm{KL}(\rho\|\pi)\right\}$$

$$= \log\left\{\pi\left[\exp\left(-\frac{\beta}{1+\kappa\frac{\lambda}{N}}\,r_S(\theta)\right)\right]\right\} + \frac{\beta}{\lambda+\kappa\frac{\lambda^2}{N}}\log\frac{1}{\epsilon}.$$

Here, the second inequality follows directly from the definition of supremum by allowing the posterior distribution to vary freely. The final equality is obtained by re-applying the Donsker–Varadhan formula, now transforming the supremum back into a simpler log-partition function explicitly involving the empirical risk $r_S(\theta)$.

Applying a union bound to the earlier localized PAC-Bayes bound (Eq. 7) — specifically, using the deviation bound (Eq. 8) to control the log-partition term involving the unknown population risk — and introducing the substitution $\xi = \frac{\beta}{\lambda\left(1+\kappa\frac{\lambda}{N}\right)}$, we obtain:

$$\mathbf{P}_S\left\{\sup_\rho\left[\left((1-\xi)\lambda - (1+\xi)\kappa\frac{\lambda^2}{N}\right)\mathbf{E}_\rho[R(\theta)] - \lambda\mathbf{E}_\rho[r_S(\theta)] - \mathrm{KL}(\rho\|\pi) - \log\mathbf{E}_\pi\left[\exp(-\xi\lambda r_S(\theta))\right]\right] \geq \right.$$
$$\left. (1+\xi)\log\left(2\epsilon^{-1}\right)\right\} \leq \epsilon.$$

Equivalently, with probability at least $1 - \epsilon$, the following PAC-Bayes bound holds for all posterior distributions $\rho$:

$$\mathbf{E}_\rho[R(\theta)] \leq \frac{\lambda\,\mathbf{E}_\rho[r_S(\theta)] + \mathrm{KL}(\rho\|\pi) + \log\left\{\mathbf{E}_\pi\left[\exp\left(-\xi\lambda r_S(\theta)\right)\right]\right\} + (1+\xi)\log(\frac{2}{\epsilon})}{(1-\xi)\lambda - (1+\xi)\kappa\frac{\lambda^2}{N}}.$$

**Simplifying with a Localized Prior.** Next, we use the following identity involving KL divergences:

$$\mathrm{KL}(\rho\|\pi) + \log\mathbf{E}_\pi[\exp(-\xi\lambda r_S(\theta))] = \mathrm{KL}\left(\rho\|\pi_{\exp(-\xi\lambda r_S)}\right) - \xi\lambda\,\mathbf{E}_\rho[r_S(\theta)],$$

where the distribution $\pi_{\exp(-\xi\lambda r_S(\theta))}(\theta) \propto \pi(\theta)\exp(-\xi\lambda r_S(\theta))$ is a localized, data-dependent prior.

With this simplification, we obtain the following explicit PAC-Bayes bound: with probability at least $1 - \epsilon$, for all posterior distributions $\rho$,

$$\mathbf{E}_\rho[R(\theta)] \leq \frac{(1-\xi)\lambda\,\mathbf{E}_\rho[r_S(\theta)] + \mathrm{KL}\left(\rho\|\pi_{\exp(-\xi\lambda r_S)}\right) + (1+\xi)\log\frac{2}{\epsilon}}{(1-\xi)\lambda - (1+\xi)\kappa\frac{\lambda^2}{N}}. \tag{9}$$

The identity relating the KL divergences can be explicitly verified through the following derivation:

$$\mathrm{KL}(\rho\|\pi) = \mathbf{E}_\rho\left[\log\frac{\rho(\theta)}{\pi(\theta)}\right] = \mathbf{E}_\rho\left[\log\frac{\rho(\theta)/\pi_{\exp(-\xi\lambda r_S)}(\theta)}{\pi(\theta)/\pi_{\exp(-\xi\lambda r_S)}(\theta)}\right]$$
$$= \mathbf{E}_\rho\left[\log\frac{\rho(\theta)}{\pi_{\exp(-\xi\lambda r_S)}(\theta)}\right] + \mathbf{E}_\rho\left[\log\frac{\pi(\theta)\exp(-\xi\lambda r_S(\theta))/\mathbf{E}_\pi[\exp(-\xi\lambda r_S(\theta))]}{\pi(\theta)}\right]$$
$$= \mathrm{KL}\left(\rho\|\pi_{\exp(-\xi\lambda r_S)}\right) + \mathbf{E}_\rho\left[\log\frac{\exp(-\xi\lambda r_S(\theta))}{\mathbf{E}_\pi[\exp(-\xi\lambda r_S(\theta))]}\right]$$
$$= \mathrm{KL}\left(\rho\|\pi_{\exp(-\xi\lambda r_S)}\right) - \xi\lambda\mathbf{E}_\rho[r_S(\theta)] - \log\mathbf{E}_\pi[\exp(-\xi\lambda r_S(\theta))].$$

**Uniformizing the Bound for Continuous Optimization.** To practically optimize the parameters $\lambda$ and $\xi$ using stochastic gradient descent (SGD), we first apply a union bound across discrete parameter grids. Specifically, we discretize the parameters onto finite sets, defining grids: $\Lambda = \{2N\zeta^k : 0 \leq k \leq \lfloor\log_{\zeta^{-1}}(2N)\rfloor\}$ for $\lambda$ with $0 < \zeta < 1$, and $\Xi = \{\alpha^k : 1 \leq k \leq \lfloor\log_{\alpha^{-1}}(N)\rfloor\}$ for $\xi$ with $0 < \alpha < 1$. We then evenly distribute the allowable failure probability $\epsilon$ across all grid points, ensuring with probability at least $1 - \epsilon$ that the PAC-Bayes bound simultaneously holds for every discrete combination of $\lambda' \in \Lambda$ and $\xi' \in \Xi$ satisfying the condition $(1-\xi') - (1+\xi')\kappa\frac{\lambda'}{N} > 0$:

$$\mathbf{E}_\rho[R(\theta)] \leq \frac{\lambda'\mathbf{E}_\rho[r_S] + \mathrm{KL}\left(\rho\|\pi\right) + \log\mathbf{E}_\pi\left[e^{-\xi'\lambda'r_S}\right] + (1+\xi')\log\frac{2|\Lambda||\Xi|}{\epsilon}}{(1-\xi')\lambda' - (1+\xi')\kappa\frac{\lambda'^2}{N}}.$$

The factor $|\Lambda||\Xi|$ explicitly accounts for the union bound over the discrete grids of $\lambda'$ and $\xi'$. We used the identity $\mathrm{KL}\left(\rho\|\pi_{\exp(-\xi\lambda r_S)}\right) = \mathrm{KL}\left(\rho, \pi\right) + \xi\lambda\mathbf{E}_\rho\left[r_S\right] + \log\mathbf{E}_\pi\left[e^{-\xi\lambda r_S}\right]$.

Next, we relax these discrete conditions to continuous parameters $\lambda \in [1, 2N]$ and $\xi \in [0, 1)$ by selecting their nearest grid points $\lambda' \in \Lambda$ and $\xi' \in \Xi$ such that $\lambda \leq \lambda' \leq \frac{\lambda}{\zeta}$ and $\xi \leq \xi' \leq \frac{\xi}{\alpha}$.

Thus, we obtain a continuous optimization-friendly PAC-Bayes bound: With probability at least $1 - \epsilon$, for all continuous parameters $\lambda \in [1, 2N]$ and $\xi \in [0, 1)$ satisfying the relaxed condition $(1 - \xi)\lambda - (1 + \frac{\xi}{\alpha})\kappa \frac{(\frac{\lambda}{\xi})^2}{N} > 0$, we have:

$$\mathbf{E}_\rho[R(\theta)] \leq \frac{\mathbf{E}_\rho[r_S(\theta)] + \frac{1}{\lambda}\left(\mathrm{KL}\left(\rho\|\pi\right) + \log \mathbf{E}_\pi\left[e^{-\xi'\lambda'r_S}\right]\right) + \frac{1+\xi/\alpha}{\lambda}\log\frac{2|\Lambda||\Xi|}{\epsilon}}{(\frac{1}{1-\xi/\alpha})(1 - \frac{\alpha+\xi}{\alpha-\xi}\kappa\frac{\lambda}{\zeta N})}. \tag{10}$$

This relaxation is crucial since it permits direct optimization of $\lambda$ and $\xi$ via gradient-based methods like SGD, making the bound highly practical for numerical implementations.

**Estimation of expectations with respect to $\rho$ and $\pi$, and empirical risk by mini-batch.** The $\rho$-/$\pi$-dependent parts of the numerator is $(1 + \xi)\mathbf{E}_\rho[r_S] + \frac{1}{\lambda}\left(\mathrm{KL}\left(\rho, \pi\right) + \log \mathbf{E}_\pi\left[e^{-\xi\lambda r_S}\right]\right)$. We approximate each expectation using Monte Carlo sampling and concentration inequalities.

For the posterior expectation term $\mathbf{E}_\rho[r_S]$, we draw $K$ independent samples $\theta_i$ from the posterior $\rho$, and estimate the empirical risk as the simple average $\widehat{\rho[r_S]} = \frac{1}{K}\sum_{i=1}^{K} r_S(\theta_i)$. Using Hoeffding's inequality (since each loss is bounded in [0,1]), we have, with probability at least $1 - \epsilon_\rho$:

$$\mathbf{E}_\rho[r_S] \leq \widehat{\rho[r_S]} + \sqrt{\frac{\log(1/\epsilon_\rho)}{2K}}.$$

Similarly, to estimate the localization term $\log \mathbf{E}_\pi\left[e^{-\xi\lambda r_S}\right]$, we draw $M$ independent samples $\tilde{\theta}_j$ from the prior $\pi$, and compute the empirical average $\widehat{\pi[e^{-\xi\lambda r_S}]} = \frac{1}{M}\sum_{j=1}^{M} e^{-\xi\lambda r_S(\tilde{\theta}_j)}$. Using Hoeffding's inequality again (since the function $e^{-\xi\lambda r_S(\theta)}$ is always between $e^{-\xi\lambda}$ and 1 ), we have with probability at least $1 - \epsilon_\pi$, we have $\mathbf{E}_\pi\left[e^{-\xi\lambda r_S}\right] \leq \widehat{\pi[e^{-\xi\lambda r_S}]} + \left(1 - e^{-\xi\lambda}\right)\sqrt{\frac{\log(1/\epsilon_\pi)}{2M}}$. Taking logs (monotone), we obtain the one-sided upper bound:

$$\log \mathbf{E}_\pi\left[e^{-\xi\lambda r_S}\right] \leq \log\left(\widehat{\pi[e^{-\xi\lambda r_S}]} + \left(1 - e^{-\xi\lambda}\right)\sqrt{\frac{\log(1/\epsilon_\pi)}{2M}}\right)$$

$$= \log \widehat{\pi[e^{-\xi\lambda r_S}]} + \log\left(1 + \frac{\left(1 - e^{-\xi\lambda}\right)}{\widehat{\pi[e^{-\xi\lambda r_S}]}}\sqrt{\frac{\log(1/\epsilon_\pi)}{2M}}\right).$$

Next we replace the full-dataset risk $r_S(\theta)$ by mini-batch confidence bounds that preserve the required one-sidedness. Let $\hat{r}_m(\theta)$ be the average loss $\ell \in [0, 1]$ on a mini-batch of size $m$ drawn without replacement. to upper-bound $\rho$-expectation, we use Hoeffding to get, w.p. at least $1 - \epsilon_r$,

$$r_S(\theta) \leq \hat{r}_m(\theta) + \sqrt{\frac{\log(1/\epsilon_r)}{2m}}.$$

To upper-bound $\mathbb{E}_\pi\left[e^{-\xi\lambda r_S(\theta)}\right]$, we note that $e^{-\xi\lambda r}$ is decreasing in $r$ and a lower bound on $r_S(\theta)$ gives an upper bound on $\exp\left(-\xi\lambda r_S(\theta)\right) \in [0, 1]$ with probability at least $1 - \epsilon_r$:

$$r_S(\theta) \geq \hat{r}_m(\theta) - \sqrt{\frac{\log(1/\epsilon_r)}{2m}} \Rightarrow \exp\left(-\xi\lambda r_S(\theta)\right) \leq \exp\left(-\xi\lambda\left(\hat{r}_m(\theta) - \sqrt{\frac{\log(1/\epsilon_r)}{2m}}\right)\right).$$

Split the failure probabilities so that the final event has probability $\geq 1 - \epsilon$, we allocate

$$\epsilon_\rho = \epsilon_\pi = \epsilon_r = \frac{\epsilon}{6}, \quad \text{to the three estimation events (posterior MC, prior MC, mini-batch } \times 2),$$

and $\epsilon_{\mathrm{grid}} = \frac{\epsilon}{3}$ to all earlier failure probabilities in Eq. 10. Let the $\lambda$-grid have size $|\Lambda|$ and the $\xi$-grid have size $|\Xi|$. To make each estimation event hold simultaneously for all grid pairs $(\lambda, \xi)$, we apply a union bound over the grid and replace every tail probability $\epsilon_\bullet$ inside the one-sided Hoeffding bounds by $\epsilon_\bullet/(|\Lambda||\Xi|)$. This is equivalent to replacing each $\log(1/\epsilon_\bullet)$ appearing in the MC/ mini-batch radii by $\log\left((|\Lambda||\Xi|)/\epsilon_\bullet\right)$. With our allocation $\epsilon_\rho = \epsilon_\pi = \epsilon_r = \epsilon/6$, all three radii pick up the common factor $\log(6|\Lambda||\Xi|/\epsilon)$.

Concretely, using the same mini-batch of size $m$ for all samples in a step and writing $\widehat{\rho[\hat{r}_m]} = \frac{1}{K}\sum_{i=1}^{K}\hat{r}_m(\theta_i)$ and $\widehat{\pi[e^{-\xi\lambda r_S}]} = \frac{1}{M}\sum_{j=1}^{M} e^{-\xi\lambda\hat{r}_m(\tilde{\theta}_j)}$, the three estimation penalties combine to

$$\sqrt{\frac{\log(6|\Lambda||\Xi|/\epsilon)}{2K}} + \sqrt{\frac{\log(6|\Lambda||\Xi|/\epsilon)}{2m}} + \xi\sqrt{\frac{\log(6|\Lambda||\Xi|/\epsilon)}{2m}} + \frac{1}{\lambda}\log\left(1 + \frac{\left(1 - e^{-\xi\lambda}\right)}{\widehat{\pi[e^{-\xi\lambda\hat{r}_m}]}}\sqrt{\frac{\log(6|\Lambda||\Xi|/\epsilon)}{2M}}\right).$$

We also need the uniformization penalty that does not come from MC. This is the standard term that ensures the localized Catoni inequality holds simultaneously over the $(\lambda, \xi)$-grid itself (independently of the MC estimation). Allocating $\epsilon_{\text{grid}} = \epsilon/3$ and union-bounding over the grid yields an additive contribution of $\frac{1+\xi/\alpha}{\lambda} \log \frac{6|\Lambda||\Xi|}{\epsilon}$.

Apply a union bound across the $\rho - \text{MC}$, the $\pi - \text{MC}$, the mini-batch confidence intervals, and the localized Catoni inequality across grids $(\lambda, \xi)$, we achieve a fully implementable empirical PAC-Bayes generalization bound, w.p. $\geq 1 - \epsilon$:

$$\mathbf{E}_\rho[R(\theta)] \leq \left(1 - \frac{\xi}{\alpha}\right)^{-1} \left(1 - \frac{\alpha+\xi}{\alpha-\xi}\kappa\frac{\lambda}{\zeta N}\right)^{-1} \left\{\widehat{\rho[\hat{r}_m]} + \frac{1}{\lambda}\left(\text{KL}(\rho, \pi) + \log \pi \widehat{[e^{-\xi\lambda\hat{r}_m}]}\right) + \delta\right\},$$

where $\delta = \sqrt{\frac{\log(6|\Lambda||\Xi|/\epsilon)}{2K}} + (1+\xi)\sqrt{\frac{\log(6|\Lambda||\Xi|/\epsilon)}{2m}} + \frac{1}{\lambda}\log\left(1 + \frac{\left(1-e^{-\xi\lambda}\right)}{\pi[e^{-\xi\lambda\hat{r}_m}]}\sqrt{\frac{\log(6|\Lambda||\Xi|/\epsilon)}{2M}}\right) + \frac{1+\xi/\alpha}{\lambda}\log\frac{2|\Lambda||\Xi|}{\epsilon}$.

We have presented a PAC-Bayes generalization bound suitable for practical optimization. Beginning with Catoni's general PAC-Bayes lemma, we incorporated a localized, data-dependent prior to sharpen the bound, replaced unknown population risks with empirical approximations, and applied uniform discretization over hyperparameters to enable direct optimization via stochastic gradient descent. Finally, we introduced Monte Carlo and importance sampling methods for numerical estimation, providing an explicit, implementable approach for evaluating and enhancing neural network generalization performance within the PAC-Bayes framework.

$\square$

### Discussion: Optimality of the Gibbs Posterior

The empirical PAC-Bayes bound (Eq. 9) at its minimal can be expressed neatly as a difference of log-partition functions. Specifically, with probability at least $1 - \epsilon$, we have

$$\mathbf{E}\rho[R(\theta)] \leq \frac{-\log \pi[\exp(-\lambda r_S(\theta))] + \log \pi[\exp(-\xi\lambda r_S(\theta))] + (1+\xi)\log\frac{2}{\epsilon}}{(1-\xi)\lambda - (1+\xi)\kappa\frac{\lambda^2}{N}}.$$

To see this optimality, recall the Donsker–Varadhan variational formula, which states that the infimum of the objective $\lambda\mathbf{E}\rho[r_S(\theta)] + \text{KL}(\rho|\pi)$ over all posterior distributions $\rho$ is precisely $-\log \pi[\exp(-\lambda r_S(\theta))]$. This infimum is uniquely attained by the Gibbs posterior distribution, given explicitly by $\rho^*(\theta) = \frac{\pi(\theta)\exp(-\lambda r_S(\theta))}{\mathbf{E}_\pi[\exp(-\lambda r_S(\theta))]}$, parameterized by the inverse temperature $\lambda$. Substituting this optimal posterior $\rho^*(\theta)$ into the PAC-Bayes bound directly yields the minimal achievable bound, characterized by the difference-of-log-partition-functions form: $-\log \pi[\exp(-\lambda r_S(\theta))] + \log \pi[\exp(-\xi\lambda r_S(\theta))]$.

### Discussion: Bias of the Log-Sum-Exp Estimator and Integral Form Alternative

The log-partition function $\log \pi[\exp(-\xi\lambda r_S(\theta))]$ in our bound is practically estimated using importance sampling with posterior samples: $\log \pi\widehat{[\exp(-\xi\lambda r_S(\theta))]}\text{IS} = \log\left(\frac{1}{K}\sum i = 1^K \frac{\pi(\theta_i)\exp(-\xi\lambda r_S(\theta_i))}{\rho(\theta_i)}\right)$. This estimator, however, is biased due to Jensen's inequality applied to the concave logarithm function. Specifically, since $\mathbf{E}[\log(X)] \leq \log(\mathbf{E}[X])$, taking the log after averaging inherently introduces negative bias. Thus, while consistent as $K \to \infty$, this estimator systematically underestimates the true log-partition function for finite sample sizes.

To obtain an unbiased alternative, we utilize the fundamental theorem of calculus. Observe that:

$$-\log \pi[\exp(-\lambda r_S(\theta))] + \log \pi[\exp(-\xi\lambda r_S(\theta))] = \int_{\xi\lambda}^\lambda \frac{d}{d\beta'}[-\log \pi(\exp(-\beta' r_S(\theta)))]\,d\beta'.$$

Evaluating the derivative inside explicitly, we have:

$$\frac{d}{d\beta'}[-\log \pi(\exp(-\beta' r_S(\theta)))] = \mathbf{E}_{\pi_{\exp(-\beta' r_S)}}[r_S(\theta)].$$

Thus, the integral form becomes:

$$-\log \pi[\exp(-\lambda r_S(\theta))] + \log \pi[\exp(-\xi\lambda r_S(\theta))] = \int_{\xi\lambda}^\lambda \mathbf{E}_{\pi_{\exp(-\beta' r_S)}}[r_S(\theta)]\,d\beta'.$$

Applying the substitution $\beta' = \beta\lambda$, we simplify to:

$$= \lambda \int_{\xi}^{1} \mathbf{E}_{\pi_{\exp(-\beta\lambda r_S)}} [r_S(\theta)] \, d\beta.$$

To numerically estimate this integral form unbiasedly, we discretize the integral using a numerical quadrature rule with points $\beta_j{}_{j=1}^{J}$ evenly spaced in $[\xi, 1]$. For each $\beta_j$, we draw samples $\theta_{ij}{}_{i=1}^{K}$ independently from the corresponding Gibbs posterior distribution $\pi_{\exp(-\beta_j\lambda r_S)}$, and then form the Monte Carlo estimator:

$$\lambda \int_{\xi}^{1} \widehat{\mathbf{E}_{\pi_{\exp(-\beta\lambda r_S)}} [r_S(\theta)]} \, d\beta = \frac{\lambda(1-\xi)}{J} \sum_{j=1}^{J} \frac{1}{K} \sum_{i=1}^{K} r_S(\theta_{ij}).$$

This estimator is unbiased by construction, as it directly estimates the expectation within the integral without applying nonlinear transformations.

The log-sum-exp estimator requires only $K$ samples from a single posterior, thus it is computationally efficient but biased. The integral-based estimator requires sampling from $J$ different Gibbs posteriors, yielding a total of $J \times K$ samples, thus it is unbiased but computationally more demanding. Choosing between the two methods involves balancing the trade-off between estimator bias and computational resources.

Compared to the log-sum-exp estimator, the integral form estimator directly estimates expected risks at multiple intermediate Gibbs posteriors, avoiding logarithmic bias entirely. However, it requires drawing samples from multiple Gibbs posteriors ($J \times K$ total samples), thus increasing computational cost significantly compared to the log-sum-exp method, which uses only $K$ samples from a single posterior. Choosing between the two involves a trade-off: the integral estimator is unbiased but computationally heavier, whereas the log-sum-exp estimator is computationally efficient but biased.

## A.2    Formal Statement and Proof of Theorem 3.2

We begin by briefly summarizing notation and key assumptions required for formally stating Theorem 3.2. Consider an $(L+1)$-layer feedforward neural network $f(\cdot, \theta)$ with parameter vector $\theta \in \Theta \subseteq \mathbf{R}^{|\theta|}$. The parameter $\theta$ explicitly includes all layer-wise weights and biases, given by:

$$\theta = \{\mathbf{W}_l, \mathbf{b}_l : 1 \leq l \leq L+1\},$$

where each layer $l$ is computed as

$$\mathbf{x}_l(\mathbf{x}_{l-1}, \theta) = f_l(\mathbf{W}_l \mathbf{x}_{l-1} + \mathbf{b}_l), \quad l = 1, \ldots, L, \quad \text{with} \quad f_{L+1} \text{ being identity.}$$

We have a training dataset $S = \{(x_i, y_i)\}_{i=1}^N$, drawn independently and identically from the data-generating distribution $D$. Inputs are collected as $\mathcal{X} = \{x_i\}_{i=1}^N$, and outputs as $\mathcal{Y} = \{y_i\}_{i=1}^N$.

Assume a bounded loss function $\ell(\theta; x, y) \in [0, 1]$. We define the *population risk* $R(\theta)$ and *empirical risk* $r(\theta)$ explicitly as: $R(\theta) = \mathbf{E}_{(x,y) \sim P}[\ell(\theta; x, y)]$ and $r(\theta) = \frac{1}{N} \sum_{i=1}^N \ell(\theta; x_i, y_i)$.

Next, consider a linearized approximation of the original neural network around initialization $\theta_0$:

$$f^{\text{lin}}(\cdot, \theta) = f(\cdot, \theta_0) + \nabla_\theta f(\cdot, \theta_0)(\theta - \theta_0).$$

Define an approximate loss $\tilde{\ell}(\theta; x, y)$ based on this linearized model, with its corresponding *approximate population risk* $\widetilde{R}(\theta)$ and *approximate empirical risk* $\widetilde{r}(\theta)$ given explicitly by:

$$\widetilde{R}(\theta) = \mathbf{E}_{(x,y) \sim P}[\tilde{\ell}(\theta; x, y)], \quad \widetilde{r}(\theta) = \frac{1}{N} \sum_{i=1}^N \tilde{\ell}(\theta; x_i, y_i).$$

We explicitly define the *model approximation error* $\epsilon_{\text{model}}(\theta)$ as the worst-case difference between the original loss and the approximate loss, satisfying uniformly:

$$|\ell(\theta; x, y) - \tilde{\ell}(\theta; x, y)| \leq \epsilon_{\text{model}}(\theta).$$

This quantity measures the approximation gap induced by linearizing the neural network.

We introduce prior and posterior distributions $\pi(\theta)$ and $\rho(\theta)$, respectively. Gibbs-type posterior distributions are defined explicitly as:

$$\pi_{\exp(-\lambda r_S(\theta))}(\theta) \propto \pi(\theta) \exp(-\lambda r_S(\theta)), \quad \pi_{\exp(-\beta \widetilde{R}(\theta))}(\theta) \propto \pi(\theta) \exp(-\beta \widetilde{R}(\theta)),$$

for parameters $\lambda, \beta > 0$. The Kullback–Leibler divergence is denoted explicitly by:

$$\text{KL}(\rho \| \pi) = \int_\Theta \rho(\theta) \log \frac{\rho(\theta)}{\pi(\theta)} \, d\theta.$$

We explicitly assume the following standard conditions:

**(A1) (Gaussian Initialization and Bounded Inputs).** Each layer $l \in \{1, \ldots, L+1\}$ has weights and biases initialized as independent Gaussian distributions:

$$\mathbf{W}_l \sim \mathcal{N}\left(0, \frac{\sigma_w^2}{n_{l-1}} \mathbf{I}\right), \quad \mathbf{b}_l \sim \mathcal{N}\left(0, \sigma_b^2 \mathbf{I}\right),$$

with input dimension $n_0$, hidden layer widths $n_l$, and inputs $\mathbf{x}_0$ bounded or sub-Gaussian. There exist finite constants $C_w, C_b, C_{x_0} > 0$, such that with high probability:

$$\|\mathbf{W}_l\| \leq C_w \sigma_w \sqrt{\frac{n_l}{n_{l-1}}}, \quad \frac{\|\mathbf{b}_l\|}{\sqrt{n_l}} \leq C_b \sigma_b, \quad \frac{\|\mathbf{x}_0\|}{\sqrt{n_0}} \leq C_{x_0}.$$

**(A2) (Bounded Width Ratios).** Layer widths $n_l$ maintain uniformly bounded ratios; explicitly, there exist constants $c_{\min}, c_{\max} > 0$ such that:

$$c_{\min} \leq \frac{n_l}{n_{l'}} \leq c_{\max}, \quad \forall l, l', \quad \text{thus } n_l = O(n) \; (\forall l).$$

**(A3) (Positive Definiteness of Infinite-Width NTK).** Define explicitly the empirical Neural Tangent Kernel (NTK) at iteration $t$ as:

$$\hat{\Theta}_t = \frac{1}{n}\nabla_\theta f(\mathcal{X},\theta_t)\nabla_\theta f(\mathcal{X},\theta_t)^\top.$$

Assume the infinite-width NTK limit $\Theta = \lim_{n\to\infty}\hat{\Theta}_0$ is strictly positive definite, with eigenvalues bounded away from 0 and $\infty$. Explicitly, there exist constants $0 < \lambda_{\min} \le \lambda_{\max} < \infty$ such that:

$$\lambda_{\min}\mathbf{I}_N \preceq \Theta = \lim_{n\to\infty}\hat{\Theta}_0 \preceq \lambda_{\max}\mathbf{I}_N.$$

**(A4) (Bounded Activations and Derivatives).** Each layer activation function $f_l$, $l = 1,\ldots,L$, has explicitly bounded magnitude and bounded first and second derivatives:

$$|f_l(0)| < \infty, \quad \|f_l'\|_\infty < \infty, \quad \|f_l''\|_\infty < \infty.$$

**(A5) (Controlled Learning Rate).** Stochastic gradient descent parameter updates are given explicitly by: $\theta_t = \theta_{t-1} - \eta\nabla_\theta r(\theta_{t-1})$, with learning rate $\eta = \eta_0/n$, where the step-size factor $\eta_0$ satisfies:

$$\eta_0 \le \frac{2}{\lambda_{\min} + \lambda_{\max}},$$

strictly below the NTK stability threshold.

**(A6) (Smooth, non-degenerate minibatch noise).** Stochastic gradient updates can be written as

$$\theta_{t+1} = \theta_t - \eta\, J(\theta_t)^\top g(\theta_t) + \xi_t(\theta_t), \qquad \mathbb{E}[\xi_t(\theta_t) \mid \theta_t] = 0,$$

with conditional covariance $\Sigma_t(\theta_t) = \mathrm{Cov}(\xi_t(\theta_t) \mid \theta_t)$. Assume: (i) *Non-degeneracy*: there exist constants $0 < \sigma_{\min} \le \sigma_{\max} < \infty$ independent of width $n$ such that, for all $t$, the linearized covariance satisfies $\sigma_{\min}I \preceq \Sigma_t^{\mathrm{lin}} \preceq \sigma_{\max}I$. (ii) *Lipschitz smoothness*: there is $L_\Sigma < \infty$ with $\|\Sigma_t(\theta) - \Sigma_t(\theta')\|_{\mathrm{op}} \le L_\Sigma\|\theta - \theta'\|_2$ for all $\theta,\theta'$.

(If needed in practice, adding an infinitesimal isotropic jitter to updates enforces (i) without affecting training.)

With this notation and under these assumptions, we formally state Theorem 3.2:

**Theorem 3.2** (PAC-Bayes Bound for Wide Networks, Formal). *Under assumptions (A1)–(A6), for any choice of regularization parameter $\lambda > 0$, localization parameter $\xi \in [0,1)$, confidence level $\varepsilon \in (0,1)$, and sufficiently large training size $N$ satisfying $0 < 1 - \frac{1}{1-\xi}g\left(\frac{\lambda}{N}\right)\frac{\lambda}{N} \le 1$, the following bound on the expected population risk under the Gibbs posterior distribution $\pi_{\exp(-\lambda r(\theta))}$ induced by stochastic gradient descent holds with probability at least $1 - \varepsilon$:*

$$\mathbf{E}_{\pi_{\exp(-\lambda r)}}[R(\theta)] \le \left(\left(1 + \tfrac{1}{1-\xi}g(\tfrac{\lambda}{N})\tfrac{\lambda}{N}\right)\mathbf{E}_\rho[\widetilde{R}(\theta)] + \tfrac{1}{(1-\xi)\lambda}\left(\tfrac{C_{\mathrm{KL}}}{n} + \log\left(\tfrac{2}{\epsilon}\right)\right)\right) \Big/ \left(1 - \tfrac{1}{1-\xi}g(\tfrac{\lambda}{N})\tfrac{\lambda}{N}\right),$$

*where $g(x) = (\exp(x) - 1 - x)/x^2$ and $C_{\mathrm{KL}} > 0$ is a constant independent of network width $n$ and training size $N$.*

We start by stating and proving the following localized oracle PAC-Bayes bounds using proxy models. While our usage of this lemma is to show that as network width grows unbounded, a neural network trained from random initialization with stochastic gradient descent generalizes as well as its simpler linear approximation at initialization, these bounds extend to various deep learning scenarios where proxy or approximate models address challenges like non-differentiable metrics, complex loss landscapes, unlabeled data, knowledge distillation, unstable reinforcement signals, and differentiable relaxations of hard constraints. Our theorems build on Catoni's localization idea but incorporate approximate losses, assuming the true loss is controlled by these approximations.

**Lemma A.2** (PAC-Bayes oracle bound with approximate population risk). *With previously defined notation, for any posterior distribution $\rho(\theta)$, prior $\pi(\theta)$, and inverse temperature $\lambda > 0$, the expected population risk under the Gibbs posterior $\pi_{\exp(-\lambda r(\theta))}$ satisfies the following bound with probability at least $1 - \varepsilon$ over the draw of the training sample $S$ :*

$$\mathbf{E}_{\pi_{\exp(-\lambda r(\theta))}}[R(\theta)] \le \tag{11}$$

$$\frac{\left(\lambda + \kappa\frac{\lambda^2}{N}\right)\mathbf{E}_\rho\left[\widetilde{R}(\theta) + \epsilon_{\mathrm{model}}(\theta)\right] + \mathrm{KL}(\rho\|\pi) + \log\mathbf{E}_\pi\left[\exp(-\beta(\widetilde{R}(\theta) - \epsilon_{\mathrm{model}}(\theta)))\right] + \log\frac{2}{\varepsilon}}{\lambda - \beta - \kappa\frac{\lambda^2}{N}}.$$

*Proof of Lemma A.2.* We begin by picking the same confidence function $\eta(\theta)$ in the proof of Theorem 3.1 *as if* we already knew the true risk $R(\theta)$:

$$\eta(\theta) = \frac{\lambda^2}{N}\kappa R(\theta) - \log\pi_{\exp(-\beta R(\bullet))}(\theta) + \log(\epsilon^{-1}), \text{ where } \kappa = g\left(\frac{\lambda}{N}\right).$$

In this function, the first term encodes our certainty that the empirical risk $r(\theta)$ is close to $R(\theta)$, effectively absorbing the Bernstein-type bound $\frac{\lambda^2}{N}g\left(\frac{\lambda}{N}\right)R(\theta)[1-R(\theta)]$ on the right-hand side of the rare event in Eq. 5 in Lemma A.1. The second term allots a portion of the confidence budget proportionally to $R(\theta)$, controlled by a positive parameter $\beta$ that dictates the "localization" strength. If $R(\theta)$ is large—meaning $\theta$ is far from a minimizer—we can afford a looser confidence interval because we care less about ensuring $r(\theta) \approx R(\theta)$. Conversely, if $R(\theta)$ is small, we impose a tighter bound near the minimizer. The final term $\log(\varepsilon^{-1})$ provides a uniform confidence shift in the tail event once we have accounted for the uncertainty in $r(\theta)$ and in choosing $\theta$.

Although this choice of $\eta(\theta)$ presupposes knowing $R(\theta)$, we eventually eliminate that assumption by relaxing $R(\theta)$ with $r(\theta)$ using the reverse bound from the same learning lemma.

Substitute $\eta(\theta)$ into Eq. 5 in Lemma A.1 yields the following bound, which holds for all posterior distributions $\rho$ with probability at least $1 - \varepsilon$:

$$\mathbf{E}_\rho[R(\theta)] \leq (\lambda - \beta - \kappa\tfrac{\lambda^2}{N})^{-1}\left\{\lambda\mathbf{E}_\rho[r(\theta)] + \mathrm{KL}(\rho\|\pi) + \log\{\pi[\exp(-\beta R(\theta))]\} + \log(\tfrac{1}{\epsilon})\right\}. \quad (7)$$

The right hand side of Eq. 7 is minimized at the Gibbs distribution $\pi_{\exp(-\lambda r(\theta))}$, in which case $\inf_\rho \mathbf{E}_\rho[\lambda r(\theta)] + \mathrm{KL}(\rho\|\pi) = -\sup_\rho \mathbf{E}_\rho[-\lambda r(\theta)] - \mathrm{KL}(\rho\|\pi) = \log\mathbf{E}_{\pi_{\exp(-\lambda r(\theta))}}[\exp(-\lambda r(\theta))]$:

$$\hat\rho_\lambda[R(\theta)] \leq \left[\lambda - \beta - \kappa\frac{\lambda^2}{N}\right]^{-1}\left\{\lambda\hat\rho_\lambda[r(\theta)] + \mathcal{K}(\hat\rho_\lambda, \pi) + \log\{\pi[\exp[-\beta R(\theta)]]\} + \log\left(\epsilon^{-1}\right)\right\}$$

$$= \inf_\rho\left[\lambda - \beta - \kappa\frac{\lambda^2}{N}\right]^{-1}\left\{\lambda\rho[r(\theta)] + \mathcal{K}(\rho, \pi) + \log\{\pi[\exp[-\beta R(\theta)]]\} + \log\left(\epsilon^{-1}\right)\right\}$$

$$\leq \left[\lambda - \beta - \kappa\frac{\lambda^2}{N}\right]^{-1}\left\{\lambda\rho[r(\theta)] + \mathcal{K}(\rho, \pi) + \log\{\pi[\exp[-\beta R(\theta)]]\} + \log\left(\epsilon^{-1}\right)\right\}.$$

To form a theoretical bound using the approximate risk $\widetilde{R}(\theta)$, we need to upper-bound both $\mathbf{E}_\rho[r(\theta)]$ and $\log(\mathbf{E}_\pi[\exp(-\beta R(\theta))])$. By approximation, we have

$$\log(\mathbf{E}_\pi[\exp(-\beta R(\theta))]) \leq \log\left(\mathbf{E}_\pi\left[\exp\left(-\beta(\widetilde{R}(\theta) - \epsilon_{\mathrm{model}}(\theta))\right)\right]\right).$$

Moreover, from Eq. 6 in Lemma A.1, with probability at least $1 - \varepsilon$:

$$\lambda\,\mathbf{E}_\rho[r(\theta)] \leq \left(\lambda + \kappa\frac{\lambda^2}{N}\right)\mathbf{E}_\rho[R(\theta)] + \log\left(\tfrac{1}{\varepsilon}\right) \leq \left(\lambda + \kappa\frac{\lambda^2}{N}\right)\mathbf{E}_\rho[\widetilde{R}(\theta) + \epsilon_{\mathrm{model}}(\theta)] + \log\left(\tfrac{1}{\varepsilon}\right).$$

By union bound, we get Eq. 11. Further using the identity

$$\mathrm{KL}(\rho\|\pi) + \log\left(\mathbf{E}_\pi\left[\exp\left(-\beta(\widetilde{R}(\theta) - \epsilon_{\mathrm{model}}(\theta))\right)\right]\right) =$$

$$\mathrm{KL}\left(\rho\|\pi_{\exp\left(-\beta(\widetilde{R}(\theta)-\epsilon_{\mathrm{model}}(\theta))\right)}\right) - \beta\,\mathbf{E}_\rho[\widetilde{R}(\theta) - \epsilon_{\mathrm{model}}(\theta)],$$

we obtain the localize oracle bound with approximate population risk: with probability at least $1 - \varepsilon$,

$$\left(\lambda - \beta - \kappa\frac{\lambda^2}{N}\right)\mathbf{E}_{\pi_{\exp(-\lambda r)}}[R(\theta)] \leq \left(\lambda + \kappa\frac{\lambda^2}{N}\right)\mathbf{E}_\rho[\widetilde{R}(\theta) + \epsilon_{\mathrm{model}}(\theta)]$$

$$+ \mathrm{KL}\left(\rho\|\pi_{\exp(-\beta(\widetilde{R}(\theta)-\epsilon_{\mathrm{model}}(\theta)))}\right) - \beta\,\mathbf{E}_\rho[\widetilde{R}(\theta) - \epsilon_{\mathrm{model}}(\theta)] + \log\left(\tfrac{2}{\varepsilon}\right).$$

A special case is when $\epsilon_{\mathrm{model}}(\theta)$ is constant. In that case:

$$\mathbf{E}_{\pi_{\exp(-\lambda r)}}[R(\theta)] \leq \left(\lambda - \beta - \kappa\frac{\lambda^2}{N}\right)^{-1} \cdot$$

$$\left\{\left(\lambda + \kappa\frac{\lambda^2}{N}\right)\mathbf{E}_\rho[\widetilde{R}(\theta)] + \mathrm{KL}(\rho\|\pi) + \log\left(\mathbf{E}_\pi\left[\exp(-\beta\widetilde{R}(\theta))\right]\right) + \left(\lambda + \kappa\frac{\lambda^2}{N} + \beta\right)\epsilon_{\mathrm{model}} + \log\left(\tfrac{2}{\varepsilon}\right)\right\}.$$

$\square$

With Lemma A.2, we prove Theorem 3.2 in the following steps. Throughout, we apply triangle inequalities, Cauchy–Schwarz, product–difference identities, and random matrix bounds under assumptions **(A1)–(A6)**.

**(i) Bounding activation and its sensitivity.** A forward recursion for each layer $l$ inductively shows that $\mathbf{x}_l(\theta) = O(\sqrt{n_l})$ and $\|\mathbf{x}_l(\theta) - \mathbf{x}_l(\tilde{\theta})\| \leq O(\sqrt{n_l}\,\|\theta - \tilde{\theta}\|)$. A similar backward recursion yields $\delta_l(\theta) = \partial_{\mathbf{x}_l}\mathbf{x}_{L+1}(\theta) = O(1)$ and $\|\delta_l(\theta) - \delta_l(\tilde{\theta})\| = O(\|\theta - \tilde{\theta}\|)$.

**(ii) Bounding Jacobian and NTK.** By applying the chain rule to each activation-sensitivity product, one obtains $\|\nabla_\theta f(\mathcal{X}, \theta)\| = O(\sqrt{n})$ and $\|\nabla_\theta f(\mathcal{X}, \theta) - \nabla_\theta f(\mathcal{X}, \tilde{\theta})\| = O(\sqrt{n}\|\theta - \tilde{\theta}\|)$. Expressing the empirical NTK $\hat{\Theta}_t$, the parameter iterate $\theta_t$, and the risk $r(\theta)$ in terms of the same activation-sensitivity-Jacobian expansions, and then iterating over SGD steps, shows $\|\hat{\Theta}_t - \hat{\Theta}_0\| = O\left(\frac{1}{\sqrt{n}}\right)$, $\|\theta_t - \theta_0\| = O\left(\frac{1}{\sqrt{n}}\right)$, and $\|f(\mathcal{X}, \theta_t) - \mathcal{Y}\| \leq \left(1 - \frac{\eta_0 \lambda_{\min}}{3}\right)^t R_0$.

**(iii) Comparing network vs. linearized updates.** By expanding $f(\theta_t) - f^{\mathrm{lin}}(\theta_t)$ in terms of $\hat{\Theta}_t - \hat{\Theta}_0$ and $f(\theta_{t-1}) - f^{\mathrm{lin}}(\theta_{t-1})$ and then telescoping, we obtain $\|f(\theta_t) - f^{\mathrm{lin}}(\theta_t)\| = O\left(\frac{t}{\sqrt{n}}\left(1 - \frac{\eta_0 \lambda_{\min}}{3}\right)^t\right)$. A parallel argument in parameter space shows $\|\theta_t - \theta_t^{\mathrm{lin}}\| = O\left(\frac{1}{n}\right)$.

**(iv) Bounding KL divergence between original and linearized learning.** Although $\theta_t$ is not strictly Gaussian, each small Gaussian increment can be compared to that of $\theta_t^{\mathrm{lin}}$ via the chain rule for KL divergence. This requires bounding log-det ratios of covariance updates (through log-matrix expansions) and bounding trace terms involving the precision matrices and the second-moment difference. Both steps rely on the empirical NTK converging to a strictly positive-definite limit, yielding $\mathrm{KL}(p_t \| p_t^{\mathrm{lin}}) \leq O\left(\frac{1}{n}\right)$.

**(v) Bounding complexity in theoretical PAC-Bayes.** Since the model approximation penalty $\frac{1}{2}\|f(\theta_t) - f^{\mathrm{lin}}(\theta_t)\|^2$ and the KL divergence $D_{\mathrm{KL}}\left(\rho \| \pi_{\exp(-\beta \widetilde{R})}\right)$ are both $O\left(\frac{1}{n}\right)$, combining these with the $\frac{1}{N}$ sample-size term completes the proof of rates on the order of $\frac{1}{n}$ (width) and $\frac{1}{N}$ (data).

We now formalize the five-step argument outlined above. Let $\theta_0$ denote the random initialization of the network parameters, and let $\theta_t$ be the parameter iterate after $t$ steps of stochastic gradient descent (SGD). Define the linearized network around $\theta_0$ by

$$f^{\mathrm{lin}}(\mathcal{X}, \theta) = f(\mathcal{X}, \theta_0) + \nabla_\theta f(\mathcal{X}, \theta_0)(\theta - \theta_0),$$

and let $\theta_t^{\mathrm{lin}}$ be the corresponding parameter iterate from gradient descent on $f^{\mathrm{lin}}$, with the same initialization and step size. We will show:

$$\left\|\theta_t - \theta_t^{\mathrm{lin}}\right\| = O\left(\frac{1}{n}\right), \quad \left\|f(\theta_t) - f^{\mathrm{lin}}(\theta_t)\right\| = O\left(\frac{1}{\sqrt{n}}\right), \quad \text{and} \quad \mathrm{KL}\left(\rho \| \pi_{\exp(-\beta \widetilde{R})}\right) = O\left(\frac{1}{n}\right)$$

and then apply the PAC-Bayes bound to conclude.

*Proof.* Before proving Step (i), we establish high-probability bounds on weights, biases, and inputs implied by assumption \ textbf{(A1)}. Each weight matrix $\mathbf{W}_l$ and bias vector $\mathbf{b}_l$ is drawn from a scaled Gaussian distribution, and inputs $\mathbf{x}_0$ are either bounded or sub-Gaussian. Consequently, with high probability $1\ \varepsilon_l$, we have:

$$\|\mathbf{W}_l\| \leq C_w \sigma_w \sqrt{\frac{n_l}{n_{l-1}}}, \quad \|\mathbf{b}_l\| \leq C_b \sigma_b \sqrt{n_l},$$

for constants $C_w, C_b > 0$, provided $n_l \geq -\log \varepsilon_l$ and $\frac{n_l}{n_{l-1}}$ is bounded. Likewise, the input norm satisfies $\|\mathbf{x}_0\| \leq \sqrt{n_0} C_{x_0}$, or with probability $1 - \varepsilon_0$ in the sub-Gaussian case.

In practice, such input conditions are met through preprocessing (e.g., centering, rescaling, or clipping). Applying a union bound across all layers, we conclude that-with probability at least $1 - \sum_l \varepsilon_l$ and for sufficiently large $n_l$-the following initialization bounds hold simultaneously:

$$\|\mathbf{W}_l\| \leq C_w \sigma_w \sqrt{\frac{n_l}{n_{l-1}}}, \quad \frac{1}{\sqrt{n_l}}\|\mathbf{b}_l\| \leq C_b \sigma_b, \quad \frac{1}{\sqrt{n_0}}\|\mathbf{x}_0\| \leq C_{x_0}$$

**(i) Bounding activations and their sensitivities.**

Starting from $n_0^{-1/2}\|\mathbf{x}_0\| \leq C_{x_0}$, we can recursively bound $n_{l+1}^{-1/2}\|\mathbf{x}_{l+1}\| \leq C_{x_{l+1}}$ with $C_{x_{l+1}} = \|f'\|_\infty \left(C_w C_{x_l}\sigma_w + C_b\sigma_b\right) + |f(0)|$:

$$\|\mathbf{x}_{l+1}\| = \|f_{l+1}\left(\mathbf{W}_{l+1}\mathbf{x}_l + \mathbf{b}_{l+1}\right)\| \leq \|f'\|_\infty \cdot \left(\|\mathbf{W}_{l+1}\mathbf{x}_l + \mathbf{b}_{l+1}\|\right) + \sqrt{n_{l+1}}|f(0)|$$
$$\leq \|f'\|_\infty \left(\|\mathbf{W}_{l+1}\| \cdot \|\mathbf{x}_l\| + \|\mathbf{b}_{l+1}\|\right) + \sqrt{n_{l+1}}|f(0)|$$
$$\leq \|f'\|_\infty \left(C_w\sigma_w\sqrt{n_{l+1}/n_l} \cdot C_{x_l}\sqrt{n_l} + C_b\sigma_b\sqrt{n_{l+1}}\right) + \sqrt{n_{l+1}}|f(0)|$$
$$= \sqrt{n_{l+1}}\left(\|f'\|_\infty \left(C_w C_{x_l}\sigma_w + C_b\sigma_b\right) + |f(0)|\right).$$

Starting from $n_0^{-1/2}\|\mathbf{x}_0(\theta) - \mathbf{x}_0(\tilde{\theta})\| = 0 \leq C_{dx_0}\|\theta - \tilde{\theta}\|$, we can recursively bound $n_{l+1}^{-1/2}\|\mathbf{x}_{l+1}(\mathbf{x}_0,\theta) - \mathbf{x}_{l+1}(\mathbf{x}_0,\tilde{\theta})\| \leq C_{dx_{l+1}}\|\theta - \tilde{\theta}\|$ with $C_{dx_{l+1}} = \|f'\|_\infty \left(C_{dx_l}C_w\sigma_w + C_{x_l}\sqrt{n_l/n_{l+1}} + 1/\sqrt{n_{l+1}}\right)$:

$$\|\mathbf{x}_{l+1}(\mathbf{x}_0,\theta) - \mathbf{x}_{l+1}(\mathbf{x}_0,\tilde{\theta})\| = \|f(\mathbf{W}_{l+1}\mathbf{x}_l(\mathbf{x}_0,\theta) + \mathbf{b}_{l+1}) - f(\tilde{\mathbf{W}}_{l+1}\mathbf{x}_l(\mathbf{x}_0,\tilde{\theta}) + \tilde{\mathbf{b}}_{l+1})\|$$
$$\leq \|f'\|_\infty \cdot \left\|(\mathbf{W}_{l+1}\cdot\mathbf{x}_l(\mathbf{x}_0,\theta) + \mathbf{b}_{l+1}) - (\tilde{\mathbf{W}}_{l+1}\cdot\mathbf{x}_l(\mathbf{x}_0,\tilde{\theta}) + \tilde{\mathbf{b}}_{l+1})\right\|$$
$$\leq \|f'\|_\infty \cdot \left(\|\mathbf{W}_{l+1}\| \cdot \|\mathbf{x}_l(\mathbf{x}_0,\theta) - \mathbf{x}_l(\mathbf{x}_0,\tilde{\theta})\| + \|\mathbf{W}_{l+1} - \tilde{\mathbf{W}}_{l+1}\| \cdot \|\mathbf{x}_l(\mathbf{x}_0,\tilde{\theta})\| + \|\mathbf{b}_{l+1} - \tilde{\mathbf{b}}_{l+1}\|\right)$$
$$\leq \|f'\|_\infty \cdot \left(C_w\sigma_w\sqrt{n_{l+1}/n_l} \cdot C_{dx_l}\sqrt{n_l} + C_{x_l}\sqrt{n_l} + 1\right)\|\theta - \tilde{\theta}\|$$
$$= \sqrt{n_{l+1}}\|\theta - \tilde{\theta}\| \cdot \|f'\|_\infty \left(C_{dx_l}C_w\sigma_w + C_{x_l}\sqrt{n_l/n_{l+1}} + 1/\sqrt{n_{l+1}}\right).$$

Let $\delta_l(\theta) = \partial_{\mathbf{x}_l}\mathbf{x}_{L+1}(\theta)$, Starting from $\|\delta_{L+1}\| \leq C_{\delta_{L+1}}$, we can recursively bound $\|\delta_l\| \leq C_{\delta_l}$ with $C_{\delta_l} = C_{\delta_{l+1}}C_w\sigma_w n_{l+1}/n_l$:

$$\|\delta_l(\mathbf{x}_0,\theta)\| = \|\delta_{l+1}(\mathbf{x}_0,\theta)\mathbf{W}_{l+1}\| \leq \|\delta_{l+1}(\mathbf{x}_0,\theta)\| \cdot \|\mathbf{W}_{l+1}\| = C_{\delta_{l+1}} \cdot C_w\sigma_w\sqrt{\frac{n_{l+1}}{n_l}}$$
$$= \left(C_{\delta_{l+1}} C_w \sigma_w \sqrt{\frac{n_{l+1}}{n_l}}\right).$$

Starting from $\|\delta_{L+1}(\mathbf{x}_0,\theta) - \delta_{L+1}(\mathbf{x}_0,\tilde{\theta})\| = 0 \leq C_{d\delta_{L+1}}\|\theta - \tilde{\theta}\|$, we can recursively bound $\|\delta_l(\mathbf{x}_0,\theta) - \delta_l(\mathbf{x}_0,\tilde{\theta})\| \leq C_{d\delta_l}\|\theta - \tilde{\theta}\|$, where $C_{d\delta_l} = C_{d\delta_{l+1}}C_w\sigma_w\sqrt{\frac{n_{l+1}}{n_l}} + C_{\delta_{l+1}}$:

$$\|\delta_l(\mathbf{x}_0,\theta) - \delta_l(\mathbf{x}_0,\tilde{\theta})\| = \|\delta_{l+1}(\mathbf{x}_0,\theta)\mathbf{W}_{l+1} - \delta_{l+1}(\mathbf{x}_0,\tilde{\theta})\tilde{\mathbf{W}}_{l+1}\|$$
$$\leq \left\|\left(\delta_{l+1}(\mathbf{x}_0,\theta) - \delta_{l+1}(\mathbf{x}_0,\tilde{\theta})\right)\mathbf{W}_{l+1}\right\| + \left\|\delta_{l+1}(\mathbf{x}_0,\tilde{\theta})\left(\mathbf{W}_{l+1} - \tilde{\mathbf{W}}_{l+1}\right)\right\|$$
$$\leq C_{d\delta_{l+1}}\|\theta - \tilde{\theta}\| \cdot C_w\sigma_w\sqrt{\frac{n_{l+1}}{n_l}} + C_{\delta_{l+1}} \cdot \|\theta - \tilde{\theta}\|$$
$$= \|\theta - \tilde{\theta}\| \cdot \left(C_{d\delta_{l+1}} C_w \sigma_w \sqrt{\frac{n_{l+1}}{n_l}} + C_{\delta_{l+1}}\right).$$

Thus, under the same union bound of events for $\|\mathbf{W}_l\|$, $n_l^{-1/2}\|\mathbf{b}_l\|$, $n_0^{-1/2}\|\mathbf{x}_0\|$, we have:

$$n_l^{-1/2}\|\mathbf{x}_l\| \leq C_{x_l}, \quad n_l^{-1/2}\|\mathbf{x}_l(\mathbf{x}_0,\theta) - \mathbf{x}_l(\mathbf{x}_0,\tilde{\theta})\| \leq C_{dx_l}\|\theta - \tilde{\theta}\|,$$
$$\|\delta_l\| \leq C_{\delta_l}, \quad \|\delta_l(\mathbf{x}_0,\theta) - \delta_l(\mathbf{x}_0,\tilde{\theta})\| \leq C_{d\delta_l}\|\theta - \tilde{\theta}\|.$$

**(ii) Bounding the network Jacobian and empirical NTK.**

The Jacobian of network output with respect to the parameters can be bounded by $(\max_l n_l)^{-1/2}\left\|\partial_{\boldsymbol{\theta}}\mathbf{x}_{L+1}\right\|_F \le C_J$, where $C_j = \|f'\|_\infty \sqrt{\sum_l \frac{n_{l-1}}{\max_l n_l}C_{\delta_l}^2 C_{\mathbf{x}_{l-1}}^2 + \frac{C_{\delta_l}^2}{\max_l n_l}}$:

$$\left\|\partial_{\boldsymbol{\theta}}\mathbf{x}_{L+1}\right\|_F^2$$

$$=\sum_l \left\|(\partial_{\mathbf{x}_l}\mathbf{x}_{L+1})\operatorname{diag}(f')\left(\partial_{\mathbf{W}_l}\mathbf{W}_l\mathbf{x}_{l-1}+\mathbf{b}_l\right)\right\|_F^2 + \left\|(\partial_{\mathbf{x}_l}\mathbf{x}_{L+1})\operatorname{diag}(f')\left(\partial_{\mathbf{b}_l}\mathbf{W}_l\mathbf{x}_{l-1}+\mathbf{b}_l\right)\right\|_F^2$$

$$\le \sum_l \|\delta_l\|_F^2 \|f'\|_\infty^2 \|\mathbf{x}_{l-1}\|_F^2 + \|\delta_l\|_F^2 \|f'\|_\infty^2 = \sum_l C_{\delta_l}^2 \cdot \|f'\|_\infty^2 \cdot n_{l-1}C_{\mathbf{x}_{l-1}}^2 + C_{\delta_l}^2 \cdot \|f'\|_\infty^2$$

$$=\max_l n_l \cdot \|f'\|_\infty^2 \sum_l \frac{n_{l-1}}{\max_l n_l}C_{\delta_l}^2 C_{\mathbf{x}_{l-1}}^2 + \frac{C_{\delta_l}^2}{\max_l n_l}$$

The difference of the Jacobians can be bounded by $(\max_l n_l)^{-1/2}\left\|\partial_{\boldsymbol{\theta}}\mathbf{x}_{L+1} - \partial_{\tilde{\boldsymbol{\theta}}}\tilde{\mathbf{x}}_{L+1}\right\|_F \le C_{dJ}\|\theta - \tilde{\theta}\|$, where $C_{dJ} = \sqrt{\sum_l \frac{n_{l-1}}{\max_l n_l}\left(C_{d\delta_l}C_{x_{l-1}} + C_{\delta_l}C_{x_l}\right)^2 + \frac{1}{\max_l n_l}C_{d\delta_l}}$:

$$\left\|\partial_{\boldsymbol{\theta}}\mathbf{x}_{L+1} - \partial_{\tilde{\boldsymbol{\theta}}}\tilde{\mathbf{x}}_{L+1}\right\|_F^2$$

$$=\sum_l \left\|(\partial_{\mathbf{x}_l}\mathbf{x}_{L+1})\operatorname{diag}(f'(\mathbf{h}_l)(\partial_{\mathbf{W}_l}\mathbf{h}_l) - (\partial_{\tilde{\mathbf{x}}_l}\tilde{\mathbf{x}}_{L+1})\operatorname{diag}(f'(\tilde{\mathbf{h}}_l))(\partial_{\tilde{\mathbf{W}}_l}\tilde{\mathbf{h}}_l)\right\|_F^2$$

$$+\left\|(\partial_{\mathbf{x}_l}\mathbf{x}_{L+1})\operatorname{diag}(f'(\mathbf{h}_l)(\partial_{\mathbf{b}_l}\mathbf{h}_l) - (\partial_{\tilde{\mathbf{h}}_l}\tilde{\mathbf{x}}_{L+1})\operatorname{diag}(f'(\tilde{\mathbf{h}}_l))(\partial_{\tilde{\mathbf{b}}_l}\tilde{\mathbf{h}}_l)\right\|_F^2$$

$$=\sum_l \left\|\delta_l(\mathbf{x}_0,\theta)\operatorname{diag}(f'(\mathbf{h}_l))\mathbf{x}_{l-1}(\mathbf{x}_0,\theta) - \delta_l(\mathbf{x}_0,\tilde{\theta})\operatorname{diag}(f'(\tilde{\mathbf{h}}_l))\mathbf{x}_{l-1}(\mathbf{x}_0,\tilde{\theta})\right\|_F^2 + \left\|\delta_l(\mathbf{x}_0,\theta)\operatorname{diag}(f'(\mathbf{h}_l)) - \delta_l(\mathbf{x}_0,\tilde{\theta})\operatorname{diag}(f'(\tilde{\mathbf{h}}_l))\right\|_F^2$$

$$\le \sum_l \left(\|\delta_l(\mathbf{x}_0,\theta)-\delta_l(\mathbf{x}_0,\tilde{\theta})\|\cdot\|f'(\mathbf{h}_l))\|\cdot\|\mathbf{x}_{l-1}(\mathbf{x}_0,\theta)\|+\|\delta_l(\mathbf{x}_0,\tilde{\theta})\|\cdot\|f'(\mathbf{h}_l)-f'(\tilde{\mathbf{h}}_l)\|\cdot\|\mathbf{x}_{l-1}(\mathbf{x}_0,\theta)\|\right.$$

$$+\|\delta_l(\mathbf{x}_0,\tilde{\theta})\|\cdot\|f'(\tilde{\mathbf{h}}_l)\|\cdot\|\mathbf{x}_{l-1}(\mathbf{x}_0,\theta)-\mathbf{x}_{l-1}(\mathbf{x}_0,\tilde{\theta})\|\big)^2+\left(\left\|\delta_l(\mathbf{x}_0,\theta)-\delta_l(\mathbf{x}_0,\tilde{\theta})\right\|_F\cdot\|f'(\mathbf{h}_l)\|_F+\|\delta_l(\mathbf{x}_0,\tilde{\theta})\|_F\cdot\left(\|f'(\mathbf{h}_l)\|_F-\|f'(\tilde{\mathbf{h}}_l)\|_F\right)\right)^2$$

$$\le \sum_l \left(C_{d\delta_l}\|\theta-\tilde{\theta}\|\cdot\|f'\|_\infty\cdot\sqrt{n_{l-1}}C_{x_{l-1}}+C_{\delta_l}\cdot\|f''\|_\infty\|\theta-\tilde{\theta}\|_2\cdot\sqrt{n_{l-1}}C_{x_{l-1}}+C_{\delta_l}\cdot\|f'\|_\infty\cdot C_{x_l}\sqrt{n_{l-1}}\|\theta-\tilde{\theta}\|\right)^2$$

$$+\left(C_{d\delta_l}\|\theta-\tilde{\theta}\|\cdot\|f'\|_\infty+C_{\delta_l}\cdot\|f''\|_\infty\|\theta-\tilde{\theta}\|\right)^2$$

$$\le \max_l n_l \cdot \|\theta-\tilde{\theta}\|^2 \sum_l \frac{n_{l-1}}{\max_l n_l}\left(C_{d\delta_l}C_{x_{l-1}}+C_{\delta_l}C_{x_l}\right)^2 + \frac{1}{\max_l n_l}C_{d\delta_l}$$

Consequently, with high probability, the Frobenius norm of the network's Jacobian (scaled by $1/\sqrt{\max_l n_l}$) is bounded by a constant $C_J$, and the difference of these scaled Jacobians is Lipschitz in $\theta$ with Lipschitz constant $C_{dJ}$. Equivalently, unscaled, both the Jacobian and the Lipschitz constant scale on the order of $\sqrt{\max_l n_l}$, where $\max_l n_l$ denotes the network's width. Formally,

$$(\max_l n_l)^{-1/2}\left\|\partial_{\boldsymbol{\theta}}\mathbf{x}_{L+1}\right\|_F \le C_J, \quad (\max_l n_l)^{-1/2}\left\|\partial_{\boldsymbol{\theta}}\mathbf{x}_{L+1}(\mathbf{x}_0,\boldsymbol{\theta}) - \partial_{\tilde{\boldsymbol{\theta}}}\mathbf{x}_{L+1}(\mathbf{x}_0,\tilde{\boldsymbol{\theta}})\right\|_F \le C_{dJ}\|\theta-\tilde{\theta}\|.$$

Under the mean-squared error (MSE) objective, one iteration of gradient descent updates

$$\theta_{t+1} = \theta_t - \eta\,\nabla_\theta f(\mathcal{X},\theta_t)^\top\left(f(\mathcal{X},\theta_t)-\mathcal{Y}\right).$$

Since $\hat{\Theta}_t = \frac{1}{n}\nabla_\theta f(\mathcal{X},\theta_t)\nabla_\theta f(\mathcal{X},\theta_t)^\top$, we have

$$f(\mathcal{X},\theta_{t+1}) - f(\mathcal{X},\theta_t) = -\eta\nabla_\theta f(\mathcal{X},\theta_t)\nabla_\theta f(\theta_t)^\top\left(f(\mathcal{X},\theta_t)-\mathcal{Y}\right) = -\eta_0\hat{\Theta}_t\left(f(\mathcal{X},\theta_t)-\mathcal{Y}\right).$$

If $\hat{\Theta}_t$ stays close to a positive-definite limit $\Theta$, this induces a contraction factor $\left\|I - \eta_0\hat{\Theta}_t\right\| \le 1 - \frac{\eta_0\lambda_{\min}}{3}$. Hence the residue decreases exponentially in training step

$$\|f(\mathcal{X},\theta_t)-\mathcal{Y}\| \le \left(1-\frac{\eta_0\lambda_{\min}}{3}\right)^t \|f(\mathcal{X},\theta_0)-\mathcal{Y}\|.$$

It also follows that $\|\theta_t - \theta_0\| \le C_J\frac{1}{\sqrt{n}}\frac{3}{\lambda_{\min}}\|f(\mathcal{X},\theta_0)-\mathcal{Y}\|$ by summer over parameter steps $t$:

$$\|\theta_{t+1}-\theta_t\| \le \frac{\eta_0}{n}\|\nabla_\theta f(\mathcal{X},\theta_t)\|\|f(\mathcal{X},\theta_t)-\mathcal{Y}\| \le \eta_0 C_J\frac{1}{\sqrt{n}}\left(1-\frac{\eta_0\lambda_{\min}}{3}\right)^t\|f(\mathcal{X},\theta_0)-\mathcal{Y}\|.$$

Throughout the training, the distance of the empirical NTK from its initialization is controlled by $\left\|\hat{\Theta}_0 - \hat{\Theta}_t\right\|_F \leq \frac{6C_J^2 C_{dJ}}{\lambda_{\min}} \|f(\mathcal{X}, \theta_0) - \mathcal{Y}\| \frac{1}{\sqrt{n}}$, where

$$
\begin{aligned}
\left\|\hat{\Theta}_0 - \hat{\Theta}_t\right\|_F &= \frac{1}{n} \left\| \nabla_\theta f(\mathcal{X}, \theta_0) \nabla_\theta f(\mathcal{X}, \theta_0)^\top - \nabla_\theta f(\mathcal{X}, \theta_t) \nabla_\theta f(\mathcal{X}, \theta_t)^\top \right\|_F \\
&\leq \frac{1}{n} \left( \|\nabla_\theta f(\mathcal{X}, \theta_0)\| \left\| \nabla_\theta f(\mathcal{X}, \theta_0)^\top - \nabla_\theta f(\mathcal{X}, \theta_t)^\top \right\| + \|\nabla_\theta f(\mathcal{X}, \theta_t) - \nabla_\theta f(\mathcal{X}, \theta_0)\| \left\| \nabla_\theta f(\mathcal{X}, \theta_t)^T \right\| \right) \\
&\leq 2C_J C_{dJ} \|\theta_0 - \theta_t\|_2 \leq \frac{6C_J^2 C_{dJ}}{\lambda_{\min}} \|f(\mathcal{X}, \theta_0) - \mathcal{Y}\| \frac{1}{\sqrt{n}}.
\end{aligned}
$$

**(iii) Comparing the network vs. linearized updates.**

In what follows, we characterize the gap between the network's prediction error $g(\theta_{t+1}) = f(\mathcal{X}, \theta_{t+1}) - \mathcal{Y}$ and the prediction error of its linear approximation at initialization $g^{\mathrm{lin}}(\theta_{t+1}) = f^{\mathrm{lin}}(\mathcal{X}, \theta_{t+1}) - \mathcal{Y}$ by examining how the gap at each previous training step accumulates throughout training. Our goal is to bound this gap in terms of the network width and the training step.

$$
\begin{aligned}
&g^{\mathrm{lin}}(\theta_{t+1}) - g(\theta_{t+1}) \\
&= \left[ g^{\mathrm{lin}}(\theta_t) - \eta_0 \hat{\Theta}_0 g^{\mathrm{lin}}(\theta_t) \right] - \left[ g(\theta_t) - \eta_0 \hat{\Theta}_t g(\theta_t) \right] \\
&= (g^{\mathrm{lin}}(\theta_t) - g(\theta_t)) - \eta_0 \hat{\Theta}_0 (g^{\mathrm{lin}}(\theta_t) - g(\theta_t)) + \eta_0 \left( \hat{\Theta}_0 - \hat{\Theta}_t \right) g(\theta_t) \\
&= \left( I - \eta_0 \hat{\Theta}_0 \right) (g^{\mathrm{lin}}(\theta_t) - g(\theta_t)) + \eta_0 \left( \hat{\Theta}_0 - \hat{\Theta}_t \right) g(\theta_t) \\
&= \ldots \\
&= \eta_0 \sum_{s=1}^{t} \left( I - \eta_0 \hat{\Theta}_0 \right)^{t-s} \left( \hat{\Theta}_0 - \hat{\Theta}_s \right) g(\theta_s).
\end{aligned}
$$

It follows that the gap between the network's prediction error (i.e., the network's output minus the target) and the error of its linear approximation at initialization decreases exponentially with each training step. Moreover, this gap scales on the order of $1/\sqrt{n}$, where $n$ is the network's width.

$$
\begin{aligned}
&\|g^{\mathrm{lin}}(\theta_{t+1}) - g(\theta_{t+1})\| \\
&\leq \eta_0 \sum_{s=0}^{t} \left\| I - \eta_0 \hat{\Theta}_0 \right\|^{t-s} \cdot \left\| \hat{\Theta}_0 - \hat{\Theta}_s \right\| \cdot \|g(\theta_s)\| \\
&\leq \eta_0 \cdot \sum_{s=1}^{t} \left( 1 - \frac{\eta_0 \lambda_{\min}}{3} \right)^{t-s} \cdot \frac{6K^3 R_0}{\lambda_{\min}} n^{-1/2} \cdot \left( 1 - \frac{\eta_0 \lambda_{\min}}{3} \right)^{s} R_0 \\
&\leq t \left( 1 - \frac{\eta_0 \lambda_{\min}}{3} \right)^{t} \frac{6K^3 R_0^2 \eta_0}{\lambda_{\min}} n^{-1/2}.
\end{aligned}
$$

The gap between the network's parameters and those of its linear approximation stems from differences in both the prediction error and the Jacobian of the network output with respect to the parameters. At each training step, these discrepancies accumulate across all previous iterations, as shown by the summation of the terms involving $g^{\mathrm{lin}}(\theta_s) - g(\theta_s)$ and $\nabla_\theta f(\mathcal{X}, \theta_s) - \nabla_\theta f(\mathcal{X}, \theta_0)$.

$$\theta_{t+1}^{\mathrm{lin}} - \theta_{t+1} = \left[\theta_t^{\mathrm{lin}} - \eta\nabla_\theta f^{\mathrm{lin}}(\mathcal{X},\theta_t)g^{\mathrm{lin}}(\theta_t)\right] - \left[\theta_t - \eta\nabla_\theta f(\mathcal{X},\theta_t)g(\theta_t)\right]$$

$$= \left(\theta_t^{\mathrm{lin}} - \theta_t\right) - \eta\left(\nabla_\theta f^{\mathrm{lin}}(\mathcal{X},\theta_t)g^{\mathrm{lin}}(\theta_t) - \nabla_\theta f(\mathcal{X},\theta_t)g(\theta_t)\right)$$

$$= \cdots = -\eta\sum_{s=0}^{s}\nabla_\theta f^{\mathrm{lin}}(\mathcal{X},\theta_s)g^{\mathrm{lin}}(\theta_s) - \nabla_\theta f(\mathcal{X},\theta_s)g(\theta_s)$$

$$= -\eta\sum_{s=0}^{s}\nabla_\theta f^{\mathrm{lin}}(\mathcal{X},\theta_s)\left(g^{\mathrm{lin}}(\theta_s) - g(\theta_s)\right) - \left(\nabla_\theta f(\mathcal{X},\theta_s) - \nabla_\theta f^{\mathrm{lin}}(\mathcal{X},\theta_s)\right)g(\theta_s)$$

$$= -\eta\sum_{s=0}^{t}\nabla_\theta f(\mathcal{X},\theta_0)\left(g^{\mathrm{lin}}(\theta_s) - g(\theta_s)\right) - \left(\nabla_\theta f(\mathcal{X},\theta_s) - \nabla_\theta f(\mathcal{X},\theta_0)\right)g(\theta_s)$$

It follows that

$$\|\theta_{t+1}^{\mathrm{lin}} - \theta_{t+1}\|$$

$$\leq \eta\sum_{s=0}^{t}\left\|\nabla_\theta f(\mathcal{X},\theta_0)\right\|\left\|g^{\mathrm{lin}}(\theta_s) - g(\theta_s)\right\| + C_{dJ}\sqrt{n}\left\|\theta_s - \theta_0\right\|\left\|g(\theta_s)\right\|$$

$$\leq \frac{\eta_0}{n}\sum_{s=1}^{t}C_J\sqrt{n}s\left(1 - \frac{\eta_0\lambda_{\min}}{3}\right)^{s-1}\frac{6C_J^2C_{dJ}R_0^2\eta_0}{\lambda_{\min}}\frac{1}{\sqrt{n}} + C_{dJ}\sqrt{n}\cdot\frac{3C_JR_0}{\lambda_{\min}}\frac{1}{\sqrt{n}}\cdot\left(1 - \frac{\eta_0\lambda_{\min}}{3}\right)^s R_0$$

$$\leq \left(\frac{6C_J^3C_{dJ}R_0^2\eta_0^2}{\lambda_{\min}}\frac{9}{\eta_0^2\lambda_{\min}^2} + C_{dJ}\frac{3C_JR_0}{\lambda_{\min}}\frac{3}{\eta_0\lambda_{\min}}R_0\right)\frac{1}{n}.$$

**(iv) Bounding the KL divergence between original and linearized learning.**

We compare the one-step update laws of the original and linearized trainings. Model a single SGD update at time $t$ by Gaussians $p_t^{\mathrm{inc}} = \mathcal{N}\left(\mu_t + \Delta\mu_t, \Sigma_t + \Delta\Sigma_t\right)$ for the nonlinear dynamics and $q_t^{\mathrm{inc}} = \mathcal{N}\left(\mu_t, \Sigma_t\right)$ for the linearized dynamics. Under (A1)-(A5), the NTK/lazy-training tube gives $\|J(\theta_t) - J(\theta_0)\|_{\mathrm{op}} = O\left(n^{-1/2}\right), \left\|g_t - g_t^{\mathrm{lin}}\right\| = O\left(n^{-1/2}\right)$, and $\|g_t\|$ decays geometrically. With $\eta = \eta_0/n$, this implies the mean-update gap $\Delta\mu_t = -\eta\left[J(\theta_t)^\top g_t - J(\theta_0)^\top g_t^{\mathrm{lin}}\right]$ is $O(1/n)$ (each bracketed term is $O(1)$, multiplied by $\eta = \eta_0/n$). Assumption (A6). 2 and the $O\left(n^{-1/2}\right)$ parameter drift yield $\|\Delta\Sigma_t\|_{\mathrm{op}} = O\left(n^{-1/2}\right)$, and (A6). 1 lets us pre-whiten by $\Sigma_t^{-1/2}$ to get $\left\|\Sigma_t^{-1/2}\Delta\Sigma_t\Sigma_t^{-1/2}\right\|_F = O\left(n^{-1/2}\right)$. A second-order expansion of the Gaussian KL around equality (zero gap) then gives

$$\mathrm{KL}\left(p_t^{\mathrm{inc}}\|q_t^{\mathrm{inc}}\right) = \frac{1}{2}\Delta\mu_t^\top\Sigma_t^{-1}\Delta\mu_t + \frac{1}{4}\left\|\Sigma_t^{-1/2}\Delta\Sigma_t\Sigma_t^{-1/2}\right\|_F^2 + o\left(\|\Delta\mu_t\|^2 + \|\Delta\Sigma_t\|_F^2\right).$$

With $\left\|\Sigma_t^{-1}\right\|_{\mathrm{op}}$ bounded (A6.1), the first term is $O(1/n)$ because $\|\Delta\mu_t\| = O(1/n)$, and the second term is $O(1/n)$ because the pre-whitened Frobenius norm is $O\left(n^{-1/2}\right)$. Summing over steps preserves the $O(1/n)$ order thanks to the geometric decay of $\|g_t\|$. The key reason we obtain $O(1/n)$ (rather than $O(1/\sqrt{n})$ ) is that the KL's leading terms are quadratic in the small $O\left(n^{-1/2}\right)$ normalized perturbations, so squaring them yields $O(1/n)$. This $O(1/n)$ trajectory-level discrepancy is then passed, by data processing, to the posteriors inside the localized PAC-Bayes bound, which adds only an explicit $C/(\lambda n)$ oracle penalty on top of the linearized risk.

**(v) Applying the PAC-Bayes argument.**

Let $\widetilde{R}(\theta)$ be the (linearized) proxy risk. Since $\|f(\theta_t) - f^{\mathrm{lin}}(\theta_t)\| = O(1/\sqrt{n})$, the actual model mismatch in the loss is $O(1/n)$. Likewise, we have $D_{\mathrm{KL}}\left(\rho\|\pi_{\exp(-\beta\widetilde{R})}\right) = O(1/n)$. Substituting these terms into the standard PAC-Bayes bound gives:

$$\mathbf{E}_{\pi_{\exp(-\lambda r)}}[R(\theta)] \leq \left(\left(1 + \frac{1}{1-\xi}g(\tfrac{\lambda}{N})\tfrac{\lambda}{N}\right)\mathbf{E}_\rho[\widetilde{R}(\theta)] + \frac{1}{(1-\xi)\lambda}\left(\frac{C_{\mathrm{KL}}}{n} + \log\left(\tfrac{2}{\epsilon}\right)\right)\right)\bigg/\left(1 - \frac{1}{1-\xi}g(\tfrac{\lambda}{N})\tfrac{\lambda}{N}\right),$$

which completes the proof of the claimed order-$\frac{1}{n}$ (width) and order-$\frac{1}{N}$ (sample size) complexity terms under assumptions **(A1)–(A6)**. $\qquad\square$

# B    Experimental Details

We describe the standard experimental configurations used for the benchmarks and tasks presented in Section 4. All models are trained using widely adopted protocols from their respective foundational papers or community baselines. Training is performed on a single NVIDIA A100 GPU, with batch sizes and runtimes selected to ensure convergence within approximately 24 hours. No data augmentation or enhancement is applied beyond basic preprocessing to ensure consistency in accounting for dataset size when evaluating PAC-Bayes generalization bounds.

## B.1    Vision Benchmarks

**MNIST.**   We evaluate two classic architectures for MNIST digit classification: a fully connected network (FCN) with three hidden layers of 600 ReLU units each, and LeNet-5[23], which consists of two convolutional layers (6 and 16 filters of size 5×5), followed by two fully connected layers with 120 and 84 units, and a 10-way softmax output. Input images are normalized to the range $[0, 1]$, with no data augmentation. We train each model using the Adam optimizer with initial learning rate $1 \times 10^{-3}$, batch size 128, and cross-entropy loss for 100 epochs. Learning rate is decayed by a factor of 0.1 at epochs 60 and 90. No dropout, batch normalization, or regularization is applied.

**CIFAR-10/100.**   We consider three standard architectures: ResNet-50[24], WideResNet-28-10[25], and DenseNet-BC-100-12[26]. Input images are normalized channel-wise using CIFAR-10 statistics (mean = [0.491, 0.482, 0.447], std = [0.247, 0.243, 0.262]). No data augmentation (e.g., cropping, flipping, MixUp, CutMix) is used to maintain countable sample size for PAC-Bayes analysis. Reference implementations for all listed CIFAR models, including ResNet, VGG, GoogLeNet, MobileNet, PreActResNet, ShuffleNet, and DenseNet, are available in the `kuangliu/pytorch-cifar` repository.

The ResNet-50 used here is the CIFAR variant, where the initial $7 \times 7$ conv and max pooling are replaced with a single $3 \times 3$ conv, and the downsampling stages follow the $6n + 2$ bottleneck pattern with increasing channel widths (16, 32, 64). We use SGD with Nesterov momentum (0.9), weight decay $5 \times 10^{-4}$, and batch size 128. The initial learning rate is 0.1, decayed by a factor of 0.2 at epochs 60, 120, and 160. Total training lasts 200 epochs.

WRN-28-10 widens all convolutional blocks by a factor of 10 and follows the pre-activation design (BN-ReLU-Conv). A dropout of 0.3 is applied between convolutional layers within residual blocks. The network is trained using the same optimizer and schedule as above.

For DenseNet-BC, we follow the $L = 100$, $k = 12$ configuration. The model consists of three dense blocks interleaved with transition layers that apply $1 \times 1$ convolution and $2 \times 2$ average pooling. Bottleneck layers and compression (growth rate reduction) are applied as in the original paper. Dropout of 0.2 is used throughout. Training uses SGD with momentum 0.9, batch size 128, initial learning rate 0.1, and learning rate drops at epochs 150 and 225, for a total of 300 epochs.

**ImageNet.**   We evaluate ResNet-50[24], DenseNet-121[26], and EfficientNet-B0[27] on ImageNet-1k. Images are resized to 256×256 via bicubic interpolation. At training time, random crops of size 224×224 and horizontal flips are applied once per image. At test time, center crops of 224×224 are used. Pixel values are normalized using ImageNet mean and standard deviation (mean = [0.485, 0.456, 0.406], std = [0.229, 0.224, 0.225]).

ResNet-50 follows the standard configuration with a $7 \times 7$ initial convolution (stride 2), max pooling, four stages of bottleneck residual blocks with channel sizes [256, 512, 1024, 2048], and global average pooling before the classifier. DenseNet-121 consists of four dense blocks with growth rate 32, compression at transition layers, and bottleneck layers with $1 \times 1$ convolutions. EfficientNet-B0 uses inverted bottleneck MBConv blocks with squeeze-and-excitation and compound scaling.

Models are trained for 120 epochs using SGD with momentum 0.9, initial learning rate 0.1, and batch size 256. The learning rate decays by a factor of 10 at epochs 30, 60, and 90. No label smoothing, RandAugment, MixUp, or repeated augmentation is used. Each model completes training in approximately two days. Reference implementations for ResNet, DenseNet, and EfficientNet are available in `torchvision.models`, `keras.applications`, and `timm`, with ImageNet training pipelines provided in repositories such as `pytorch/examples/imagenet`.

## B.2   Semantic Segmentation

We evaluate semantic segmentation on the Cityscapes dataset using a standard U-Net architecture with an ImageNet-pretrained ResNet-34 encoder. Images are resized to 256×512, center-cropped if necessary, and normalized using Cityscapes mean and standard deviation. The model follows the encoder-decoder pattern with skip connections between symmetric encoder and decoder blocks, each containing two convolutional layers followed by batch normalization and ReLU activations. The decoder upsamples spatial resolution using transposed convolutions.

Training is performed using the Adam optimizer (learning rate $1 \times 10^{-4}$, batch size 8) for 100 epochs. We use a pixel-wise cross-entropy loss over 19 semantic classes. No label smoothing or augmentation is applied. All layers are probabilistic except the final segmentation head, which remains deterministic. We evaluate performance on the validation set using mean Intersection-over-Union (mIoU). We train using pixel-wise cross-entropy loss and evaluate using mIoU, following standard Cityscapes protocol. Our PAC-Bayes bound formally applies to the expected cross-entropy loss, which correlates well with mIoU in practice. Our configuration is consistent with widely used U-Net implementations such as `keras-unet-collection` and `segmentation_models.pytorch`.

## B.3   GLUE Fine-tuning with GPT-2 + LoRA

We fine-tune GPT-2-small (124M parameters) on three classification tasks from the GLUE benchmark[31]—MRPC, SST-2, and RTE—using a parameter-efficient fine-tuning setup with LoRA [32]. Our protocol builds on standard GPT-2 token classification with causal language modeling: each input is serialized as a prompt (e.g., "sentence1 <sep> sentence2") followed by a label token ("Yes", "No", "Positive", etc.), and the model is trained using next-token cross-entropy loss over the label position. This formulation enables us to reuse standard language model heads without modifying the decoder architecture.

We adopt the GPT-2 tokenizer with a maximum sequence length of 128 tokens and batch size 32. We use AdamW with learning rate $2 \times 10^{-5}$, $\beta_1 = 0.9$, $\beta_2 = 0.98$, and $\epsilon = 10^{-8}$. Weight decay is set to zero, and training runs for 10 epochs unless early stopping is triggered. Our setup follows the Keras example Parameter-efficient fine-tuning of GPT-2 with LoRA `https://keras.io/examples/nlp/parameter_efficient_finetuning_of_gpt2_with_lora/` and Hugging Face's adapter-based fine-tuning pipelines, with GPT-2 checkpoints loaded from the Hugging Face model hub.

We inject LoRA adapters into the GPT-2 self-attention blocks, targeting the query and value projection matrices in each transformer layer. Each LoRA module is a rank-4 adapter with scaling factor $\alpha = 32$ and dropout 0.05. The base GPT-2 weights remain frozen throughout training. For classification, we project the final token's hidden state onto a softmax over verbalized class tokens (e.g., "Yes", "No"), computing loss on the generated token. This verbalizer-based approach avoids architectural changes while preserving interpretability.

To apply our localized PAC-Bayes bound, we replace the standard loss with a variational objective defined over the LoRA parameters. These adapters are reparameterized as Gaussian posteriors, and the mini-batch is partitioned into $K = 8$ shards. For each shard, we sample LoRA weights and compute predictions, aggregating empirical risk via importance-weighted averaging. KL divergence is estimated from the same samples. Bound parameters $\lambda$ and $\xi$ are jointly optimized with the adapter weights using a reparameterized sigmoid transformation. The training loop and data pipeline are otherwise unchanged, demonstrating that our PAC-Bayes loss integrates seamlessly into standard fine-tuning workflows.

We report training and validation losses as well as PAC-Bayes generalization bounds (Eq. 3).

## B.4   PAC-Bayes Training Details

We implement our localized PAC-Bayes training procedure as a drop-in replacement for standard training workflows. Our method defines a single-pass objective that integrates importance-weighted empirical risk estimation with reparameterized posterior sampling and KL regularization, enabling seamless compatibility with Keras, TensorFlow, and analogous PyTorch abstractions.

For each model, we replace standard deterministic layers with variational layers (e.g., reparameterized dense or convolutional layers with learned Gaussian posteriors), where posterior samples are drawn

using the reparameterization trick. At each training step, the mini-batch is split into $K = 4$ shards, and $K$ forward passes are used to approximate expectations with importance weighting. Our loss replaces the standard cross-entropy loss with a PAC-Bayes bound involving data-dependent estimates of empirical risk, KL divergence, and importance-weighted log-partition estimate.

To control the bound's tightness and ensure numerical stability, we jointly optimize two scalar parameters $(\lambda, \xi)$ using reparameterized sigmoid transformations. These parameters are clipped to remain in a numerically stable range. The parameter $\xi$, in particular, governs the balance between empirical fit and regularization. The bound uses a Bernstein-type inequality combined with the Donsker–Varadhan variational form to achieve tight guarantees under approximate posterior sampling.

Posterior and prior distributions are modularized at the layer level and managed through lightweight wrappers. The full PAC-Bayes loss, including posterior sampling and KL terms, is evaluated during each forward pass and is fully differentiable. In frameworks like Keras, this integrates naturally via model compilation and fitting. In PyTorch or JAX, the same logic can be incorporated directly within the standard mini-batch training loop, without requiring two-pass estimation, KL annealing, or external scheduling. This makes our method easily portable across ecosystems with minimal changes to model or training code.

Our implementation avoids reliance on specialized inference libraries and instead uses lightweight wrappers that expose posterior sampling and KL evaluation in a modular form. In PyTorch, these components can be integrated as attributes of standard exttttnn.Module classes, enabling straightforward porting of posterior logic into existing model definitions. Our approach parallels existing PEFT (parameter-efficient fine-tuning) strategies such as LoRA or Adapter modules, in the sense that it "injects" stochastic behavior inside existing model layers without rewriting the full training logic. Crucially, our PAC-Bayes loss is computed inline with the model's forward and backward passes, without requiring separate optimization stages or KL annealing. This design allows it to be integrated into standard training loops across architectures and frameworks.

All posterior layers are initialized with random priors and appropriate variance scaling. The PAC-Bayes loss is fully compatible with standard Keras callbacks, model checkpointing, and TensorBoard logging.

## B.5 Certifiable Prediction

Post-training, we use the same posterior distribution to derive per-example prediction certificates. For each test input, we compute the expected 0–1 loss under posterior sampling via importance-weighted risk estimation. We then apply a single-sample Markov bound and a union bound across the test set to produce a certified upper bound on the misclassification risk of each individual prediction, as formalized in Eq. 4.

This certifiable risk enables a rejection mechanism: predictions whose estimated risk exceeds a chosen threshold can be flagged or abstained from, ensuring reliable deployment in safety-critical applications. Our experiments visualize the full risk distribution and demonstrate that adversarial or misclassified samples tend to exhibit significantly higher certified risk than correctly predicted ones. We also evaluate calibration curves and find that PAC-Bayes risk estimates align closely with empirical misclassification rates, supporting their use for uncertainty-aware selective prediction.

## B.6 Software and Reproducibility

All models are implemented in TensorFlow 2.12 with Keras 2 and HuggingFace Transformers 4.28 (for GPT-2 LoRA fine-tuning). Training is conducted in Google Colab using a single NVIDIA A100 GPU with 40 GB of memory. To ensure feasibility under Colab constraints, all experiments are designed to complete within a single 24-hour session. Our codebase is written in R and uses the `reticulate` package to interface with Python modules. Due to compatibility requirements of TensorFlow Probability, we restrict the software environment to Python 3.10 and TensorFlow `<=2.15`. Code and configuration scripts for all experiments will be made available upon publication.

## C   Concentration Inequalities and PAC-Bayes Bounds

This section presents a self-contained statement and derivation of the key inequalities underlying PAC-Bayes bound. Our goal is to help readers unfamiliar with PAC-Bayes theory or concentration inequalities appreciate the theoretical structure and understand how our bound emerges naturally from classical tools.

We begin with the simplest and most foundational of these: Markov's inequality. It formalizes the intuitive idea that if a random variable tends to be small on average, then large values must be rare. This simple insight is the root of a broad family of results known as concentration inequalities, which describe how tightly a random variable tends to concentrate around its average.

**Markov's Inequality.** If $X \geq 0$ is a nonnegative random variable and $a > 0$, then:

$$\mathbf{P}(X \geq a) \leq \frac{\mathbf{E}[X]}{a}.$$

*Proof.* Let $A = \{X \geq a\}$. Then:

$$\mathbf{E}[X] = \int_\Omega X(\omega) d\mathbf{P}(\omega) \geq \int_A X(\omega) d\mathbf{P}(\omega) \geq \int_A a \, d\mathbf{P}(\omega) = a \cdot \mathbf{P}(X \geq a).$$

Dividing both sides by $a$ gives the result. $\qquad\qquad\square$

Markov's inequality gives weak but general tail bounds. Its real power emerges when we apply it to transformed variables. If large deviations of $X$ are rare, then so are large values of $e^{\lambda X}$. This exponential magnification enables sharper control.

**Hoeffding's Inequality.** Suppose $X_1, \ldots, X_n$ are independent and bounded in $[0, 1]$, and let $\bar{X} = \frac{1}{n} \sum X_i$. Then for any $\varepsilon > 0$:

$$\mathbf{P}(\bar{X} - \mathbf{E}[\bar{X}] \geq \varepsilon) \leq \exp(-2n\varepsilon^2).$$

*Proof.* Consider $\bar{X} = \frac{1}{n} \sum X_i$. We apply Markov's inequality to the exponential of the deviation:

$$\mathbf{P}(\bar{X} - \mathbf{E}[\bar{X}] \geq \varepsilon) = \mathbf{P}(e^{\lambda n(\bar{X} - \mathbf{E}[\bar{X}])} \geq e^{\lambda n \varepsilon}) \leq e^{-\lambda n \varepsilon} \mathbf{E}[e^{\lambda n(\bar{X} - \mathbf{E}[\bar{X}])}].$$

By independence, we can factor the expectation: $\mathbf{E}[e^{\lambda n(\bar{X} - \mathbf{E}[\bar{X}])}] = \prod_{i=1}^{n} \mathbf{E}[e^{\lambda(X_i - \mathbf{E}[X_i])}]$.
Apply Hoeffding's lemma: for any $X_i \in [0, 1]$, we have $\mathbf{E}[e^{\lambda(X_i - \mathbf{E}[X_i])}] \leq e^{\lambda^2/8}$, so
$\mathbf{E}[e^{\lambda n(\bar{X} - \mathbf{E}[\bar{X}])}] \leq e^{n\lambda^2/8}$. Hoeffding's lemma follows by applying a second-order Taylor expansion to the moment-generating function of a bounded random variable and optimizing over the bound on its range.

Combining this with the previous bound: $\mathbf{P}(\bar{X} - \mathbf{E}[\bar{X}] \geq \varepsilon) \leq e^{-\lambda n \varepsilon + n\lambda^2/8}$.

Minimizing the right-hand side wrt $\lambda$ gives $\lambda = 4\varepsilon$, yielding: $\mathbf{P}(\bar{X} - \mathbf{E}[\bar{X}] \geq \varepsilon) \leq \exp(-2n\varepsilon^2)$. $\square$

Hoeffding's inequality provides uniform control over deviations based solely on the bounded range of each variable—it does not consider the variance or how probability mass is distributed within the range. In contrast, Bernstein's inequality makes use of both the variance and a bound on the centered deviations. This allows it to produce tighter bounds when the variance is small, particularly in settings with heterogeneous or skewed noise.

**Bernstein's Inequality.** Let $\sigma_1, \ldots, \sigma_N$ be independent real-valued random variables, each bounded above by $b$ in deviation from its mean: $\sigma_i - \mathbf{E}[\sigma_i] \leq b$. Let $S = \frac{1}{N} \sum \sigma_i$ be the normalized sum, $m = \mathbf{E}[S]$ its mean, and $V = \frac{1}{N} \sum \mathbf{E}[(\sigma_i - \mathbf{E}[\sigma_i])^2]$ its renormalized variance. Then, for any $\lambda > 0$ and any $\eta > 0$:

$$\mathbf{P}(S - m \geq \eta) \leq \exp\left(-\lambda\eta + g\left(\frac{b\lambda}{N}\right) \cdot \frac{V}{N}\lambda^2\right), \quad \text{where } g(x) = \frac{e^x - 1 - x}{x^2}.$$

This exponential moment bound captures both the variance and the tail size of the distribution. We use it directly in our PAC-Bayes analysis to account for heteroskedasticity and sharp transitions in empirical risk estimates.

*Proof.* Let $\sigma_1, \ldots, \sigma_N$ be independent real-valued random variables with $\mathbf{E}[\sigma_i] = \mu_i$ and $\sigma_i - \mu_i \leq b$ almost surely. Define the empirical average $S = \frac{1}{N} \sum_{i=1}^N \sigma_i$, with mean $m = \mathbf{E}[S] = \frac{1}{N} \sum \mu_i$, and define the renormalized variance $V = \frac{1}{N} \sum \mathbf{E}[(\sigma_i - \mu_i)^2]$.

We begin by applying Markov's inequality to the exponential: for any $\lambda > 0$,

$$\mathbf{P}(S - m \geq \eta) = \mathbf{P}\left(e^{\lambda(S-m)} \geq e^{\lambda \eta}\right) \leq e^{-\lambda \eta} \mathbf{E}[e^{\lambda(S-m)}].$$

By independence, we factor the moment generating function as $\mathbf{E}[e^{\lambda(S-m)}] = \prod_{i=1}^N \mathbf{E}[e^{\lambda(\sigma_i - \mu_i)/N}]$. Since each $\sigma_i - \mu_i \leq b$, we apply the bound $\log \mathbf{E}[e^{tX}] \leq g(bt) \cdot \mathbf{E}[X^2]$ for $X$ bounded by $b$ and $g(x) = \frac{e^x - 1 - x}{x^2}$, to get

$$\log \mathbf{E}[e^{\lambda(S-m)}] \leq \sum_{i=1}^N g\left(\frac{b\lambda}{N}\right) \cdot \frac{1}{N^2} \cdot \mathbf{E}[(\sigma_i - \mu_i)^2] = g\left(\frac{b\lambda}{N}\right) \cdot \frac{V}{N} \cdot \lambda^2.$$

Putting everything together:

$$\mathbf{P}(S - m \geq \eta) \leq \exp\left(-\lambda \eta + g\left(\frac{b\lambda}{N}\right) \cdot \frac{V}{N} \lambda^2\right).$$

This bound holds for all $\lambda > 0$; optimizing it over $\lambda$ yields the standard Bernstein tail bound. $\square$

The following Donsker-Varadhan variational formula connects the moment-generating function (i.e., the log of the expected exponentiated $f$ ) to a variational optimization over distributions $\rho$. It is especially powerful when $f$ represents a deviation, such as $f(\theta) = \lambda(r_S(\theta) - R(\theta))$, which is common in PACBayes theory.

The inequality shows that rather than analyzing $\mathbf{E}_\pi[e^f]$ directly, we can upper bound it using any distribution $\rho$, at the cost of a KL divergence penalty. In PAC-Bayes, $\pi$ serves as the prior and $\rho$ as the learned posterior, making this identity the key bridge between exponential moment bounds and generalization guarantees.

**Donsker–Varadhan Variational Formula.** Let $\pi$ be a probability distribution over a space $\Theta$, and let $f : \Theta \to \mathbf{R}$ be measurable and integrable under $\pi$. Then:

$$\log \mathbf{E}_\pi[e^{f(\theta)}] = \sup_\rho \left\{\mathbf{E}_\rho[f(\theta)] - \mathrm{KL}(\rho \| \pi)\right\},$$

where the supremum is taken over all probability measures $\rho$ absolutely continuous with respect to $\pi$.

*Proof.* Let $\rho \ll \pi$ (i.e., $\rho$ is absolutely continuous with respect to $\pi$, meaning $\rho$ does not assign mass where $\pi$ assigns zero), and define the Radon-Nikodym derivative $d\rho/d\pi = h$. In continuous spaces, if $\pi$ has a density $p(\theta)$ and $\rho$ has density $q(\theta)$, then $h(\theta) = \frac{q(\theta)}{p(\theta)}$ is the pointwise ratio of densities. We write:

$$\mathbf{E}_\rho[f(\theta)] - \mathrm{KL}(\rho \| \pi) = \int f(\theta) h(\theta) d\pi(\theta) - \int h(\theta) \log h(\theta) d\pi(\theta).$$

Define a functional $J(h)$ as

$$J(h) = \int f h \, d\pi - \int h \log h \, d\pi,$$

subject to the constraint $\int h \, d\pi = 1$, since $\rho$ must be a probability measure.

Using calculus of variations or Lagrange multipliers to maximize $J(h)$, the optimal $h^*(\theta)$ satisfies

$$f(\theta) - \log h^*(\theta) - 1 - \lambda = 0 \quad \Rightarrow \quad h^*(\theta) = \frac{e^{f(\theta)}}{Z}, \quad Z = \int e^{f(\theta)} d\pi(\theta).$$

Plugging this optimal $h^*$ back into the original expression gives

$$\sup_\rho \left\{\mathbf{E}_\rho[f] - \mathrm{KL}(\rho \| \pi)\right\} = \log \int e^{f(\theta)} d\pi(\theta) = \log \mathbf{E}_\pi[e^{f(\theta)}],$$

as claimed. $\square$

The following PAC-Bayes bound, due to Catoni [6], provides a generalization guarantee for randomized predictors drawn from a posterior distribution $\rho$, in terms of their empirical risk and divergence from a prior $\pi$. Unlike classical uniform convergence bounds, which struggle with infinite or continuous hypothesis spaces, this approach avoids union bounds altogether by reasoning about the expected behavior under $\rho$, using the Donsker–Varadhan variational identity.

**Catoni's PAC-Bayes bound**[6] Let $\pi$ be a prior distribution over parameters $\theta \in \Theta$, and let $\rho$ be any posterior distribution. Assume that for all $\theta$, the loss function $\ell(\theta; x, y) \in [0, C]$ is bounded. Then for any $\lambda > 0$ and $\delta \in (0, 1)$, with probability at least $1 - \delta$ over the draw of a dataset $S = \{(x_i, y_i)\}_{i=1}^N$, we have:

$$\mathbf{E}_{\theta \sim \rho}[R(\theta)] \le \mathbf{E}_{\theta \sim \rho}[r_S(\theta)] + \frac{\lambda C^2}{8N} + \frac{\mathrm{KL}(\rho \| \pi) + \log(1/\delta)}{\lambda}.$$

*Proof.* Fix any $\theta \in \Theta$. By Hoeffding's inequality applied to $\ell_i(\theta) \in [0, C]$, we have for any $t > 0$:
$\mathbf{E}_S \left[ e^{t \cdot N(R(\theta) - r_S(\theta))} \right] \le e^{\frac{t^2 N C^2}{8}}$. Setting $t = \lambda/N$: $\mathbf{E}_S \left[ e^{\lambda(R(\theta) - r_S(\theta))} \right] \le e^{\frac{\lambda^2 C^2}{8N}}$.

Integrate this bound over $\theta \sim \pi$: $\mathbf{E}_{S, \theta \sim \pi} \left[ e^{\lambda(R(\theta) - r_S(\theta))} \right] \le e^{\frac{\lambda^2 C^2}{8N}}$. By Fubini's theorem, we exchange the order of integration: $\mathbf{E}_S \left[ \mathbf{E}_{\theta \sim \pi} \left[ e^{\lambda(R(\theta) - r_S(\theta))} \right] \right] \le e^{\frac{\lambda^2 C^2}{8N}}$.

Apply the Donsker–Varadhan variational formula to the inner expectation:

$$\mathbf{E}_S \left[ \exp \left( \sup_\rho \{ \lambda \mathbf{E}_\rho[R(\theta) - r_S(\theta)] - \mathrm{KL}(\rho \| \pi) \} \right) \right] \le e^{\frac{\lambda^2 C^2}{8N}}.$$

Apply a Chernoff bound: for any $s > 0$, $\mathbf{P}_S \left( \sup_\rho \lambda \mathbf{E}_\rho[R - r_S] - \mathrm{KL}(\rho \| \pi) > \frac{\lambda^2 C^2}{8N} + s \right) \le e^{-s}$.

Setting $s = \log(1/\delta)$ and rearranging gives:

$$\mathbf{P}_S \left( \exists \rho : \mathbf{E}_\rho[R(\theta)] > \mathbf{E}_\rho[r_S(\theta)] + \frac{\lambda C^2}{8N} + \frac{\mathrm{KL}(\rho \| \pi) + \log(1/\delta)}{\lambda} \right) \le \delta.$$

Taking the complement concludes the proof. $\square$

The following KL-risk formulation, commonly seen in the PAC-Bayes literature, is due to McAllester [1] and Langford–Seeger [3]. It directly bounds the KL divergence between the expected empirical risk and the expected true risk. However, directly optimizing this bound with respect to $\rho$ is challenging in practice. We do not use this bound in our work but include it here for completeness.

**Theorem (PAC-Bayes Bound via KL of Risks).** Let $\pi$ be a prior distribution over predictors $\theta \in \Theta$, and let $\rho$ be any posterior. For any $\delta \in (0, 1)$ and integer $N \ge 2$, with probability at least $1 - \delta$ over the draw of dataset $S \sim D^N$, the following inequality holds:

$$\mathrm{KL} \left( \mathbf{E}_\rho[r_S(\theta)] \, \| \, \mathbf{E}_\rho[R(\theta)] \right) \le \frac{\mathrm{KL}(\rho \| \pi) + \log \frac{N}{\delta}}{N - 1}.$$

*Proof.* Let $f(\theta) = (N - 1) \mathrm{KL}(r_S(\theta) \| R(\theta))$, and apply the Donsker–Varadhan inequality: $\mathbf{E}_\rho[f(\theta)] \le \mathrm{KL}(\rho \| \pi) + \log \mathbf{E}_\pi \left[ e^{f(\theta)} \right]$. Taking expectation over the sample $S$ and applying Fubini gives: $\mathbf{E}_S \mathbf{E}_\rho[f(\theta)] \le \mathrm{KL}(\rho \| \pi) + \log \mathbf{E}_{S, \theta \sim \pi} \left[ e^{f(\theta)} \right]$.

To control the exponential moment, define $p = R(\theta)$, $q = r_S(\theta)$, and note that $\hat{e}(h, S)$ is a Binomial$(N, p)$ proportion. Using an explicit bound on the moment-generating function of $(N - 1) \mathrm{KL}(q \| p)$ under the binomial sampling process, we can show:

$$\mathbf{E}_S \left[ e^{(N-1) \mathrm{KL}(r_S(\theta) \| R(\theta))} \right] \le N, \quad \text{uniformly in } \theta.$$

Applying Markov's inequality, with probability at least $1 - \delta$,

$$\log \mathbf{E}_\pi \left[ e^{(N-1) \mathrm{KL}(r_S(\theta) \| R(\theta))} \right] \le \log \frac{N}{\delta}.$$

Plugging this into the earlier bound, and dividing both sides by $N - 1$, we conclude:

$$\mathrm{KL} \left( \mathbf{E}_\rho[r_S(\theta)] \, \| \, \mathbf{E}_\rho[R(\theta)] \right) \le \frac{\mathrm{KL}(\rho \| \pi) + \log \frac{N}{\delta}}{N - 1}.$$

$\square$

