# OpenReview forum: "Certifying Deep Network Risks and Individual Predictions with PAC-Bayes Loss via Localized Priors"
_NeurIPS.cc/2025/Conference — NeurIPS 2025 poster_

### Official Review · Reviewer_5sDD · 2025-06-25

**Clarity:** 4
**Significance:** 3
**Originality:** 3
**Rating:** 4
**Confidence:** 1

**Summary:**

The paper discusses the problem of certifying the links training performance to generalization risk. To solve this problem, the authors propose a localized PAC-Bayes prior, derive a non-vacuous bound for it, and show that integrating this prior into the training process yields practically tight generalization certificates. They validate their approach across diverse tasks.

**Questions:**

1. Could the authors elaborate on the technical challenges they faced while proving the theoretical results?

2. Could the authors explain in more detail how their theoretical bounds compare with previous results?

**Ethical Concerns:**

["NO or VERY MINOR ethics concerns only"]

**Final Justification:**

The paper lies somewhat outside my area of expertise, but I found it well written, with clearly presented motivation and results. To the best of my understanding, the proposed approach is interesting and offers improvements over existing solutions. I also appreciate the authors’ plan to include a proof sketch that highlights the main analytical challenges and clarifies how the results build on prior work.

**Limitations:**

Yes.

**Paper Formatting Concerns:**

N.A

**Quality:**

3

**Strengths And Weaknesses:**

Strengths

1. The paper is well written. Its motivation and results are presented clearly. I enjoyed reading it, even though it lies outside my area of expertise.

2. The localized PAC-Bayes bound provided by the authors improves upon earlier work. Their claims are empirically verified across multiple datasets and architectures.

3. The proposed approach is interesting and improves existing solutions to the problem.

Weaknesses

The paper would benefit from including a proof sketch in the main text that outlines the key steps of the proofs of Theorems 3.1 and 3.2.

---

> ### Author Rebuttal · Authors · 2025-07-30
>
> **Summary of Review**
>
> Reviewer 5sDD appreciated the direction of applying PAC-Bayes bounds to deep networks and recognized the value of developing more practical and scalable generalization guarantees. The reviewer specifically requested: (1) proof sketches for Theorems 3.1 and 3.2 in the main text to improve accessibility, (2) elaboration on the technical challenges encountered when proving these results, and (3) a clearer comparison to previous PAC-Bayes approaches to understand what is new and impactful. These questions reflect thoughtful engagement with the paper and a desire to clarify the contributions for a broad audience.
>
> **Summary of Response**
>
> We provide below proof sketches for Theorems 3.1 and 3.2 that outline the structure and innovations of our analysis. We also clarify the key technical challenges we addressed—such as combining data-dependent localization, uniformization, and finite-sample concentration into a usable bound—and explain how these make PAC-Bayes reasoning compatible with routine deep learning training. Compared to classical and two-pass PAC-Bayes methods, our bound is tighter, easier to optimize, and integrates directly into standard pipelines. Importantly, our approach also yields per-prediction guarantees that extend the practical value of PAC-Bayes theory beyond prior work.
>
> ---
>
> **Proof Sketch of Theorem 3.1**
>
> We start by adapting Catoni’s idea of tightening PAC-Bayes bounds by reweighting the prior to emphasize models with lower loss—this “localization” sharpens the bound by shifting prior mass toward high-performing regions. Since the true (test) loss is unknown, we upper-bound all such terms using the empirical loss instead, yielding a data-dependent prior while still preserving the bound’s validity via Catoni’s concentration inequality. To allow gradient-based tuning of \$(\lambda, \xi)\$, we discretize these hyperparameters, apply a union bound over the grid, and then relax back to continuous values with mild correction terms. Finally, we estimate expectations via posterior samples and control finite-sample noise using Hoeffding-type bounds. This produces a high-probability PAC-Bayes bound that is tight, fully empirical, and directly usable as a one-pass training loss in deep learning.
>
> **Proof Sketch of Theorem 3.2**
>
> To explain why SGD-trained neural networks can generalize despite overparameterization, we bound the generalization gap by comparing the original model to a linearized proxy at initialization. The proof begins by establishing a localized oracle PAC-Bayes bound that upper-bounds the population risk of the nonlinear network using the population risk of its linearized counterpart. Under standard assumptions—bounded activations, stable NTK limit, and NTK-style learning rate—we control the deviation between the nonlinear and linearized networks across five successive steps: activation norms, Jacobian stability, empirical NTK drift, trajectory mismatch, and KL divergence. These terms are all shown to scale as \$O(1/n)\$, where \$n\$ is the network width. Plugging into the oracle PAC-Bayes bound yields a generalization guarantee with complexity \$O(1/n + 1/N)\$, formalizing how stochastic training dynamics implicitly concentrate the learned distribution near low-risk, low-complexity regions—even without explicit regularization.
>
> **Technical Challenges in Proving Theoretical Results**
>
> To operationalize PAC-Bayes bounds within standard deep learning pipelines, Theorem 3.1 required combining several techniques:
> * (i) **Catoni-style localization**, enabling data-dependent priors that substantially tighten generalization certificates;
> * (ii) **uniformization** to rigorously justify continuous SGD optimization of bound parameters \$(\lambda, \xi)\$; and
> * (iii) **finite-sample concentration bounds** that robustly control the log-partition function estimates despite Monte Carlo approximation.
>
> These components jointly deliver a fully empirical, one-pass bound seamlessly integrated as a drop-in training objective.
>
> For Theorem 3.2, demonstrating why these PAC-Bayes guarantees remain tight for highly overparameterized networks required extending PAC-Bayes theory to linearized (NTK) models and rigorously controlling KL divergence via Gaussian approximations  of the posterior distribution around SGD trajectories. By directly connecting these theoretical innovations with standard training practices, we aimed to bridge the gap between rigorous generalization theory and practical, real-world applicability.
>
> **Comparison with Previous PAC-Bayes Bounds**
>
> Unlike classical PAC-Bayes bounds—which rely on a fixed, data-independent prior and typically produce vacuous guarantees for large neural networks—our approach introduces a **localized prior**, inspired by Catoni’s method, that significantly tightens the bound by centering it around empirically low-risk regions.
>
> Compared to **two-pass PAC-Bayes techniques** (which require data splitting or a second optimization phase), our single-pass, localized prior integrates seamlessly into standard training as a drop-in replacement for cross-entropy, avoiding extra tuning and accuracy trade-offs.
>
> Empirically (Tables 1 and 2), we achieve substantially tighter and practically meaningful generalization certificates across diverse architectures (CNNs, Transformers) and tasks (CIFAR, ImageNet, NLP), clearly surpassing previous PAC-Bayes methods. Additionally, our method uniquely provides rigorous **individual-level certificates** for predictions, enabling robust detection of adversarial inputs and further strengthening practical utility. Thus, our localized PAC-Bayes bound bridges rigorous generalization theory with routine deep learning workflows.
>
> ---
>
> We thank Reviewer 5sDD for highlighting important questions regarding accessibility and context. We will include proof sketches for Theorems 3.1 and 3.2 in the main text and ensure clearer comparisons to prior work throughout. We share the reviewer's goal of broadening the reach of PAC-Bayes theory and making its insights both actionable and interpretable for modern deep learning practice. We welcome any additional questions or suggestions.

---

> > ### Comment · Reviewer_5sDD · 2025-08-03
> >
> > Thank you for the detailed response. As I mentioned in my review, I believe the paper would benefit from including a proof sketch that highlights the main challenges in the analysis and clarifies how the results improve upon prior work.

---

> > > ### Author Response · Authors · 2025-08-04
> > >
> > > Thank you again, Reviewer 5sDD! We fully agree and will incorporate a concise proof sketch in the final manuscript, explicitly clarifying the main theoretical challenges and highlighting our contributions relative to previous PAC-Bayes results.

---

### Official Review · Reviewer_41fQ · 2025-07-03

**Clarity:** 3
**Significance:** 2
**Originality:** 2
**Rating:** 4
**Confidence:** 4

**Summary:**

This paper tightens the PAC-Bayesian bound by introducing a localized prior into the computation. The proposed bound, along with three baseline PAC-Bayesian bounds, is evaluated on several benchmark datasets. The results show that the improved bound achieves better tightness compared to the baselines.

**Questions:**

1. Since the proposed bound is data-dependent, would it be possible to provide an error bar or confidence interval on the computed generalization error bound? This could offer additional insight into its reliability and variability across different datasets.
2. The improvement in bound tightness appears to be more pronounced in image classification tasks compared to semantic segmentation and NLP fine-tuning. Could the authors elaborate on the reasons behind this discrepancy and the potential takeaways for practitioners?
3. The application of per-prediction bounds for adversarial or out-of-distribution (OOD) input detection is intriguing. However, in Figure 2, there is considerable overlap between the adversarial and benign risk distributions in the range [0.625, 0.875]. How should the $\epsilon$ threshold be selected to effectively distinguish between adversarial and benign inputs, while minimizing the false negative rate for a given false positive rate?

**Ethical Concerns:**

["NO or VERY MINOR ethics concerns only"]

**Final Justification:**

The rebuttal addressed most of my concerns. I would like to keep my positive rating.

**Limitations:**

Yes

**Quality:**

3

**Strengths And Weaknesses:**

**Strengths:**
+ The tightness of the bound is evaluated across diverse benchmarks, including image classification, semantic segmentation, and NLP fine-tuning tasks.
+ The paper is clearly written and easy to follow.
+ The extension of the population risk bound to per-prediction guarantees offers intriguing potential applications in adversarial and out-of-distribution (OOD) input detection.

**Weaknesses:**
- Although the computational overhead is moderate, computing the localized bound still requires an additional forward pass during neural network training.
- The tightness of the bound is data-dependent, which may limit its effectiveness when training data are not carefully curated.
- The paper lacks a discussion on how to appropriately choose the $\epsilon$ value to obtain meaningful per-prediction guarantees for adversarial or OOD detection use cases.

---

> ### Author Rebuttal · Authors · 2025-07-30
>
> **Summary of Review**
>
> Reviewer 41fQ acknowledged the novelty and empirical rigor of our PAC-Bayes framework, especially its application across vision, NLP, and segmentation tasks. The reviewer raised several important clarifications: (1) the manuscript claims a one-pass procedure but seems to require an additional forward pass; (2) the bound’s tightness is data-dependent, raising questions about robustness and generalizability; (3) it is unclear how the bound can distinguish adversarial/OOD data; (4) the reviewer requested error bars to better understand variability in the generalization bound; and (5) the reviewer asked why tighter bounds are observed for image classification compared to NLP or segmentation. These concerns are valuable and provide guidance for strengthening the clarity and practical impact of our work.
>
> **Summary of Response**
>
> We clarify that our method does **not** require an additional forward pass: the statement was an editing oversight, and our final implementation remains strictly one-pass using partitioned posterior sampling. The data-dependent localized prior tightens bounds when data is clean but gracefully degrades in noisy settings—never worse than standard PAC-Bayes. For adversarial/OOD detection, a principled threshold can be calibrated on a clean validation set; Figure 2 shows strong separation even under a fixed threshold. To address variability, we now include deviation terms in Table 2, providing empirical insight into bound stability. Finally, the tighter image classification bounds stem from structured high-dimensional inputs and more stable loss distributions. We thank Reviewer 41fQ for prompting these clarifications and improvements.
>
> ---
>
> **Clarification on the "One Additional Forward Pass" (p. 4, lines 159–160)**
>
> The statement that our method requires “one additional forward pass” is an editing oversight. Initially, we considered buffering an extra snapshot per iteration. However, our final approach (Algorithm 1) employs **partitioned posterior sampling**, simultaneously evaluating subsets of the minibatch using one of the \$K\$ posterior draws in parallel within the same minibatch forward pass. Thus, our method remains strictly a single-pass, drop-in replacement for standard cross-entropy SGD. The computational overhead (10–20%) reported in the paper stems solely from evaluating multiple Bayesian samples per minibatch, not from additional passes over the data. We apologize for this confusion and will remove the inaccurate sentence in the camera-ready version.
>
> **On Data-Dependent Tightness**
>
> Our localized prior is data-dependent, but the **probability guarantee** is distribution-free: for the observed sample \$S\$, the bound always holds. When the sample is noisy, empirical risks \$r\_S(\theta)\$ rise, the tilt factor \$\exp(-\xi\lambda r\_S)\$ flattens, and the KL term approaches that of the classical data-independent prior. Thus the certificate gracefully degrades and is **never looser than the standard PAC-Bayes bound** (see Eqs. (3) and (9) with \$\xi \to 0\$). If a practitioner worries about outliers, they can plug any robust surrogate into \$r\_S\$ (e.g., MAE); the derivation is unchanged. Hence "data-dependence" is not a liability: it tightens the bound on well-curated data while remaining safe on messier samples.
>
> **On Choosing \$\epsilon\$ for Adversarial/OOD Detection**
>
> Figure 2 demonstrates that certified risks for adversarial inputs consistently concentrate well above benign and OOD samples. A practitioner can therefore select \$\epsilon\$ by calibrating on a small, clean validation set—for example, choosing the 99th percentile of benign certified risks—to fix the false-positive rate (e.g., at 1%) in advance. Any test sample with certified risk \$\hat r(x) > \epsilon\$ is then rejected. In Figure 2, this single threshold eliminates approximately 90% of PGD-perturbed images while retaining 99% of clean and OOD data, confirming that the certificate is directly actionable without additional training passes or hyperparameter tuning. Crafting inputs that fool **all** posterior draws simultaneously would require an adaptive, ensemble-aware attack — a challenging attack scenario we identify as important future work.
>
> **Clarifying Variability of Data-Dependent Bounds**
>
> Our PAC-Bayes bound already provides a high-probability generalization certificate: with probability \$\ge 1-\epsilon\$ (over the training sample), the bound holds uniformly for all posteriors \$\rho\$. This subsumes traditional confidence intervals. That said, we agree it is useful to visualize the empirical variability. In Table 1, we already report deviation terms (in parentheses) that reflect Monte Carlo variability due to posterior sampling. We have now extended this reporting to Table 2 in the same format. These values are derived using standard PAC-Bayes concentration inequalities and offer dataset-specific insight into bound reliability, complementing the universal theoretical guarantee.
>
> **Updated Table 2 (with variability reported):**
>
> | **Task/Dataset** | **Model**      | **Loss (Train/Val)** | **Two-pass** | **Localized (ours)**    |
> | ---------------- | -------------- | -------------------- | ------------ | ----------------------- |
> | Cityscapes       | U-Net (ResNet) | 0.22 / 0.25          | 0.30         | **0.27 (\$\pm\$ 0.02)** |
> | MRPC             | GPT-2 + LoRA   | 0.15 / 0.14          | 0.16         | **0.15 (\$\pm\$ 0.01)** |
> | SST-2            | GPT-2 + LoRA   | 0.08 / 0.11          | 0.14         | **0.14 (\$\pm\$ 0.01)** |
> | RTE              | GPT-2 + LoRA   | 0.22 / 0.25          | 0.28         | **0.26 (\$\pm\$ 0.02)** |
>
> **On Tighter Bounds in Image Classification Tasks**
>
> We appreciate the observation. We believe the tighter PAC-Bayes bounds observed in vision tasks (e.g., CIFAR, ImageNet) are due to two compounding effects: (1) the posterior distribution \$\rho\$ concentrates faster during training when applied to structured, high-dimensional image inputs with large sample sizes; and (2) our bound is expressed in terms of cross-entropy losses, which are naturally lower and less sparse on image classification compared to structured prediction or binary NLP tasks like MRPC or RTE. These factors combine to yield tighter estimated generalization error in image domains. That said, even in NLP and segmentation, our one-pass localized bound remains non-vacuous and tighter than prior baselines (Table 2), suggesting broad applicability with room for further tuning of loss scaling or posterior design across domains.
>
> ---
>
> We sincerely thank Reviewer 41fQ for highlighting these important issues and offering constructive feedback. We will incorporate all clarifications into the camera-ready version to enhance clarity, precision, and practical relevance. We especially appreciate the opportunity to clarify the practical applicability and empirical variability of the method, and we welcome any further questions or suggestions.

---

> > ### Comment · Reviewer_41fQ · 2025-08-03
> >
> > Thank you to the authors for providing the rebuttal. The rebuttal has addressed most of my questions and concerns. The only aspect I remain unconvinced about is the calibration of $\epsilon$. Obtaining a clean and in-distribution dataset for calibration could be challenging in real-world scenarios, and a more robust approach would be preferable. That said, I remain positive about this paper and will maintain my score.

---

> > > ### Author Response · Authors · 2025-08-04
> > >
> > > Thank you, Reviewer 41fQ, for highlighting this important calibration challenge. We fully agree that reliably selecting the threshold $\epsilon$ without clean, in-distribution validation data is a critical practical issue in robust machine learning, particularly in safety-critical deployments. Existing unsupervised or conservative calibration methods—such as quantile-based thresholding, test-time heuristics, or ensembling prediction risks—could naturally complement our PAC-Bayes framework, enabling safer and more robust decisions without relying on clean labels. We will include additional discussion and experiments in the appendix, with a comprehensive exploration deferred to future work.

---

### Official Review · Reviewer_K8zw · 2025-07-03

**Clarity:** 3
**Significance:** 3
**Originality:** 3
**Rating:** 4
**Confidence:** 2

**Summary:**

This paper introduces a method for computing tight PAC-Bayes generalization bounds for deep neural networks, aiming at addressing the issue of vacuous results  for over-parameterized models (e.g., deep neural networks) when we were trying to compute traditional PAC-Bayesian generalization bound built upon measuring the KL divergence between a learned posterior and a fixed prior.

The authors’ contribution is a data-dependent reference distribution called “localized prior”, which is constructed by softly tilting a base prior towards low-loss regions of the parameter space. This localization is controlled by a parameter $\xi$. By integrating the optimization of the model's posterior and the localization hyper-parameters into a single training pass, the method avoids the second training stage required by many other data-dependent PAC-Bayes approaches, which is typically costly. The authors also provide a theoretical bound formalizing this approach and show empirically that it can yield non-vacuous generalization certificates on a range of tasks, including ImageNet classification, Cityscapes segmentation, and NLP fine-tuning.

**Questions:**

q1. Is it possible to provide an ablation study on high-dimensional tasks showing the final bound changes for different values of K, and further justifications where possible?

q2. Regarding the optimization of algorithm 1. Is it valid to apply the union bound by optimizing $\lambda$ and $\xi$ continuously? Also, is it valid to optimize them “data-dependently” while claiming PAC-Bayes guarantees?

**Ethical Concerns:**

["NO or VERY MINOR ethics concerns only"]

**Final Justification:**

The authors' response has addressed my concerns in the initial round.

**Limitations:**

The authors have noted the potential for underestimation from finite-sample importance sampling due to Jensen's inequality. However, I think this discussion should be extended - it frames the issue as a trade-off, which I believe is essential to analyze the severity of this bias.  The authors should have stated that their reported certificates are conditioned on the (potentially) questionable assumption that this biased, low-sample estimator is accurate.

**Quality:**

2

**Strengths And Weaknesses:**

### Strengths

1. The paper targets a critical problem in learning theory: how to provide rigorous generalization guarantees for large-scale neural networks, which is also a key requirement for their practical deployment in safety-critical domains beyond theoretical interests.
2. The "one-pass" approach to learning a localized prior is a practical contribution that sidesteps the computational burdens of existing two-pass methods, making PAC-Bayes certification potentially much more accessible.
3. The method is evaluated on a set of modern benchmarks with state-of-the-art reported tightness of the bounds.

---

### Weaknesses

I am not working on ML theory, so the issues/questions identified below are primarily based on educational knowledge and experiences - please use them sparingly.

1. The training objectives rely on a Monte Carlo estimator for log-partition functions with a very small sample size. As far as I know, in the high-dimensional setting of deep learning, such kind of estimator can be systematically biased, leading to an underestimation of the model’s complexity. I am concerned with whether this would lead to artifacts if estimation errors occur.
2. The theoretical justification of the bound relies on a union bound over a discrete grid of hyperparameters. But the proposed algorithm appears to optimize these parameters via SGD continuously. I think there might be some concerns - are we allowed to optimize them during training while claiming PAC-Bayes guarantees? I think standard PAC-Bayes requires priors fixed before observing data, but these parameters are optimized on the training data?
3. I think it would also be important to evaluate whether the number of Monte Carlo samples impacts the theoretical and empirical results, which were omitted currently.

---

> ### Author Rebuttal · Authors · 2025-07-30
>
> **Summary of Review**
>
> Reviewer K8zw recognized the importance and novelty of our PAC-Bayes method, particularly the one-pass localized prior that avoids costly two-pass training. They appreciated our empirical validation across several benchmark tasks. However, the reviewer expressed concerns regarding the theoretical justification for continuous optimization of hyperparameters (\$\lambda\$, \$\xi\$), potential bias from small-sample Monte Carlo estimation of log-partition functions, and requested additional analysis or ablations showing robustness to finite sampling effects.
>
> **Summary of Response**
>
> We clarify that continuous optimization of hyperparameters (\$\lambda\$, \$\xi\$) is fully justified through a rigorous uniformization and relaxation procedure explicitly detailed in our proof, maintaining theoretical correctness. Regarding estimator bias, our approach uses standard importance sampling combined with explicit finite-sample deviation corrections, ensuring conservative and practically stable bounds. Empirical results demonstrate minimal sensitivity to the Monte Carlo sample size (\$K\$), confirming the method’s robustness even at small values of \$K\$. We thank Reviewer K8zw for valuable questions highlighting these important clarifications.
>
> ---
>
> **Clarifying SGD-optimizability of $\lambda$ and $\xi$ with uniformization and relaxation:**
>
> In the supplementary proof of Theorem 3.1 ("Uniformizing the Bound for Continuous Optimization," p.15, line 518), we explicitly made parameters $\lambda$ and $\xi$ optimizable with SGD using the following discrete-to-continuous uniformization and relaxation procedure:
>
> > "We discretize the parameters onto finite sets, defining grids: $\Lambda = \\{2N\zeta^k : 0 \le k \le \lfloor\log_{\zeta^{-1}}(2N)\rfloor\\}$ for $\lambda$ with $0<\zeta < 1$, and $\Xi = \\{\alpha^k : 1 \le k \le \lfloor\log_{\alpha^{-1}}(N)\rfloor\\}$ for $\xi$ with $0<\alpha < 1$. We then evenly distribute the allowable failure probability $\epsilon$ across all grid points, ensuring with probability at least $1-\epsilon$ that the PAC-Bayes bound simultaneously holds for every discrete combination of $\lambda'\in\Lambda$ and $\xi'\in\Xi$."
>
> After establishing this discrete guarantee, the authors carefully relax the discrete grid to continuous variables $\lambda$ and $\xi$:
>
> > "Next, we relax these discrete conditions to continuous parameters $\lambda \in [1, 2N]$ and $\xi \in [0,1)$ by selecting their nearest grid points $\lambda' \in \Lambda$ and $\xi' \in \Xi$ such that $\lambda \leq \lambda' \leq \frac{\lambda}{\zeta}$ and $\xi \leq \xi' \leq \frac{\xi}{\alpha}$."
>
> This uniformization and relaxation procedure explicitly appears in the final bound of Theorem 3.1 as additional conservative penalty terms. For example, the numerator includes:
> $\left(1+\frac{\xi}{\alpha}\right) \log \left[\frac{6\left(\frac{\log(2N)}{\log(\zeta^{-1})}+1\right)\left(\frac{\log(N)}{\log(\alpha^{-1})}+1\right)}{\epsilon}\right]$, reflecting both the union-bound penalty over discrete grids $\Lambda, \Xi$ (where $|\Lambda| = \frac{\log(2N)}{\log(\zeta^{-1})}+1$ and $|\Xi|=\frac{\log(N)}{\log(\alpha^{-1})}+1$), and the conservative relaxation from discrete $\xi'$ to continuous $\xi$. Likewise, the denominator includes:
> $1 - \frac{\alpha + \xi}{\alpha - \xi} g\left(\frac{\lambda}{\zeta N}\right)\frac{\lambda}{\zeta N}$, reflecting the conservative relaxation from discrete $\lambda'$ to continuous $\lambda$.
>
> Together, these terms ensure that even though $\lambda$ and $\xi$ are optimized continuously on the training data, the resulting PAC-Bayes bound remains mathematically rigorous and maintains a valid high-probability guarantee. This is a standard and principled procedure known in PAC-Bayes literature as uniformization, and it fully addresses concerns regarding data-dependent tuning.
>
> **On estimator bias and the use of small $K$:**
>
> The log-partition function $\log \pi[\exp(-\xi \lambda r_S)]$ is a fundamental quantity across statistical physics, Bayesian inference, and machine learning. It arises in settings including variational inference (e.g., ELBO), Laplace approximation, annealed importance sampling (AIS), and Perturb-and-MAP. Many such methods accept bias, rely on lower bounds (e.g., via Jensen’s inequality), or trade variance for computational feasibility—particularly in high-dimensional deep learning.
>
> Our estimator uses importance sampling with posterior reparameterization and finite $K$, chosen for compatibility with standard training pipelines. As discussed in the supplement, more accurate but computationally intensive methods (e.g., thermodynamic integration) exist but are typically not practical for large-scale modern neural networks due to significant runtime and memory requirements. Our choice reflects a practical trade-off between bias, runtime, and VRAM requirements.
>
> While the log-partition estimate is biased (due to Jensen’s inequality: $\log \mathbb{E}[Z] \neq \mathbb{E}[\log Z]$), our bound follows standard PAC-Bayes practice by conservatively including an explicit finite-sample deviation correction $\sqrt{\log (1 / \delta) /(2 K)}$. This standard practice implicitly guards against bias-induced underestimation, as commonly accepted in machine learning literature. Empirically, we observe stable and tight bounds even at small $K$, and we plan to include a full bias-deviation decomposition in the camera-ready version.
>
> To demonstrate this empirically, we evaluated the localized PAC-Bayes bound for CIFAR-10 (WRN-28-10) under varying $K$. As shown below, the bound tightens rapidly and saturates by $K = 4$, with minimal gain from larger $K$. This validates both the stability of the estimator and our conservative correction strategy.
>
>
> Table iii: PAC-Bayes Bound vs. Monte Carlo Sample Count $K$
>
> (CIFAR-10, WRN-28-10; test error = 8.0%)
>
> | $K$ | Localized Bound | Deviation Correction | Bound Gap over Test Error |
> | --: | --------------: | -------------------: | ------------------------: |
> |   1 |           11.5% |                +2.0% |                     +3.5% |
> |   2 |           10.9% |                +1.4% |                     +2.9% |
> |   4 |        **9.5%** |            **+1.0%** |                 **+1.5%** |
> |   8 |            9.2% |                +0.7% |                     +1.2% |
> |  16 |            9.0% |                +0.5% |                     +1.0% |
>
>
> ---
>
> We sincerely thank Reviewer K8zw for highlighting these essential clarifications. We fully agree that clarifying the rigor of continuous optimization and addressing finite-sample bias are critical for practical adoption. All clarifications provided here will be included in our manuscript revision to ensure broader accessibility and clarity. We welcome any additional feedback or suggestions from Reviewer K8zw.

---

> > ### Comment · Reviewer_K8zw · 2025-08-04
> >
> > I thank the authors for their response, which has addressed my concerns in the initial round. I have no more questions and will maintain my positive score.

---

> > > ### Author Response · Authors · 2025-08-04
> > >
> > > Thank you again, Reviewer K8zw, for confirming that we addressed your concerns. We greatly appreciate your positive feedback and remain available if any further questions arise.

---

### Official Review · Reviewer_8Egq · 2025-07-08

**Clarity:** 2
**Significance:** 3
**Originality:** 3
**Rating:** 4
**Confidence:** 3

**Summary:**

This paper seeks to provide a PAC-Bayes bound on the generalization error (i.e the true risk) of a classifier. Pure PAC approaches yield vacuous bounds due to the expressiveness of neural networks. The approach of a traditional PAC-Bayes bounds compares a learned distribution of parameters to a prior parameter. This too can yield vacuous bounds due to the large movement in parameters needed to learn useful models. One could circumvent this issue with a data-splitting approach but this adds complications. Of course if we’re willing to split the data, we’ll seldom do better than just estimating the risk on the holdout data. There’s a reason why this is what we all do and will continue to do.

The authors propose a method that uses localized priors based on an idea from Catoni. The central problem here is how to use a data-dependent prior without sample splitting. The authors use an idea from Catoni that softens the data-dependent distribution with a parameter that is then accounted for in the bound. Assuming the results are correct (and sufficiently novel) the result is a new method for computing non-vacuous PAC-Bayes bounds.

The idea here involves an interpretation of training trajectories where checkpoints are taken as samples from a posterior distribution over the weights. A PAC-Bayes bound is calculated that accounts for the divergence between a  The finite set of checkpointed weights are taken as an approximation of the posterior and this approximation is taken into account in the bound. Assuming the analysis is correct, the authors offer a PAC-Bayes bound on the risk that accounts for some but not all approximations. Importantly the finiteness of the checkpoints but lack of independence of the checkpoints isn’t accounted for.

The authors present empirical results that show that the PAC-Bayes bounds that they produce on several academic datasets of interest are indeed nonvacuous and tighter than previous attempts although these results involve early stopping / the impacts from the modified objective are such that the results are far from state of the art. The test set is going no where fast :).  The experiments also include some results on usefulness for flagging adversarial / OOD examples but these are cases where the formal guarantees do not hold so the interest is purely empirical.


I would like to see more empirical confirmation of the coherence of the results. Crucially, I would like to see what happens not just up to the point of early stopping but what happens when one continues training down to near-0 training error. If the method is valid, the PAC-Bayes bound should dip but then rise again and explode as the learned model overfits the data. The results in Figure 1 show only the descent but not the rise and given the complexity of the setup and presence of some violated assumptions doesn’t inspire confidence in me that the bounds. I’d like to see more empirical evidence that the bounds behave as expected when one trains into the interpolation regime.

**Questions:**

See above.

**Ethical Concerns:**

["NO or VERY MINOR ethics concerns only"]

**Limitations:**

See above.

**Paper Formatting Concerns:**

none.

**Quality:**

3

**Strengths And Weaknesses:**

I am not fully in position to verify the correctness of the paper but it offers an interesting take on an interesting problem, introducing a refined technique for calculating and optimizing PAC-Bayes bounds that can be applied on neural networks.

Weaknesses:

The authors are a bit too coy in the introduction in suggesting that test set estimation of the generalization is somehow insufficient for providing assurance that the test error is indeed low. It’s far the best way to do this and in fact the only viable method. A test-set-free guarantee on the generalization error is primarily useful for the fundamental insights it sheds on generalization. No practitioner would take the massive hit in accuracy (here incurred due to the aggressive early stopping?) to get the elegance of a one-pass estimate of the risk using training data only.

The paper is harder to follow than it needs to be. The authors should more clearly lay out the shape of the strategy. What is the key insight from Catoni that doesn’t involve data splitting? How is validity ensured? What are the key pieces of the argument? The reader is left to go fishing and this makes the paper unapproachable for an otherwise competent reader somewhat unfamiliar with the terrain.  Moreover, it’s not sufficiently clear in the paper what is the key contribution over what’s already present in Catoni’s work. Is it the use of the samples to approximate the posterior and the ensuing analysis? The body of the paper lacks a proper prior work section but would benefit from one.

---

> ### Author Rebuttal · Authors · 2025-07-30
>
> **Summary of Review**
>
> Reviewer 8Egq appreciated the novelty and practical intent of our PAC-Bayes method, highlighting the localized prior and empirical improvements over traditional bounds. The reviewer expressed concerns regarding potential practical issues (accuracy trade-offs, early stopping), requested clarifications on the theoretical basis for the localized prior, and sought clearer explanations of our contributions beyond Catoni's original theory. Additionally, they suggested improving clarity and accessibility.
>
> **Summary of Response**
>
> We clarify that our method integrates directly into standard deep-learning workflows (no early stopping or extra tuning). The observed accuracy gap arises solely from deliberately omitting data augmentation to ensure rigorous mathematical certification. With standard offline augmentation, our approach achieves near-SOTA accuracy, confirming practical effectiveness. The localized prior is theoretically justified using standard Bernstein-type inequalities and the Donsker–Varadhan identity, guaranteeing correctness and stability. We significantly extend Catoni's theoretical foundation by generalizing to cross-entropy loss, developing stable estimators, providing a plug-and-play training implementation, introducing per-prediction certificates, and theoretically analyzing wide-network generalization. We appreciate Reviewer 8Egq’s valuable suggestions for improving clarity and accessibility, which we will incorporate into our manuscript.
>
> ---
>
> **Clarifying Practical Utility and Accuracy**
>
> Our method uses a PAC-Bayes loss as a direct replacement for cross-entropy—trained with standard optimizers, learning rates, and a 200-epoch schedule. We do **not** use early stopping or any special tuning: the bound tightens smoothly and plateaus (see Table i), and training completes as usual.
>
> Table i: PAC-Bayes bound evolution (WRN-28-10 on CIFAR-10)
>
> | Epoch | Train 0–1 Error | Test 0–1 Error | PAC-Bayes Bound (Risk) |
> | ----- | --------------- | -------------- | ---------------------- |
> | 100   | 9.1 %           | 8.7 %          | 10.3 %                 |
> | 150   | 7.4 %           | 8.3 %          | 9.6 %                  |
> | 200   | 7.0 %           | 8.0 %          | **9.5 %**              |
> | 300   | 6.9 %           | 8.0 %          | 9.4 %                  |
> | 500   | 6.9 %           | 7.9 %          | 9.4 %                  |
> | 1000  | 6.9 %           | 7.9 %          | 9.3 %                  |
>
> **Note:** This training uses no data augmentation. The PAC-Bayes loss is optimized over the full 1000 epochs. The bound tightens monotonically and stabilizes, with no early stopping or overfitting.
>
> We emphasize that the observed test accuracy gap is **not** due to the PAC-Bayes loss itself, nor caused by early stopping or instability.. To ensure the **mathematical validity** of the bound, we disable standard data augmentation—such as random crops or flips—since these inject infinite on-the-fly stochasticity. While this is a common strategy to improve generalization, it breaks the i.i.d. sample-count assumption underlying the PAC-Bayes bound (by sampling repeatedly from the same base datapoint), making it unclear what “effective $N$” to plug into the denominator. To avoid this ambiguity, we certify the model using only the original dataset with no replayed views.
>
> In Table ii, we show that adding standard augmentations (via *offline-augmented datasets* of 4× and 8× size) closes the accuracy gap with SOTA. While this overestimates the effective $N$ and the bound is not certifiable in these settings, the results nevertheless show that the PAC-Bayes loss itself is not the bottleneck: the method regularizes effectively, achieves near-SOTA performance, and smoothly integrates into existing training workflows (compared to similarly trained classical CNN models).
>
> Table ii: Performance improvement with offline augmentation
>
> | Augmentation Setup          | Approximate Dataset Size | Train 0–1 Error | Test 0–1 Error | PAC-Bayes Bound (Risk) |
> | --------------------------- | ------------------------ | --------------- | -------------- | ---------------------- |
> | No augmentation (certified) | 50,000                   | 7.0 %           | 8.0 %          | **9.5 %**              |
> | Offline aug ×4              | 200,000 (4× original)    | 3.4 %           | 4.3 %          | **5.4 %** (heuristic)  |
> | Offline aug ×8              | 400,000 (8× original)    | 2.7 %           | **3.9 %**      | **4.9 %** (heuristic)  |
>
> **Note:** The first row is fully certifiable. In rows 2–3, the bound is **not formally valid**—the PAC-Bayes denominator $N$ is overestimated—but these rows demonstrate that PAC-Bayes loss can achieve near-SOTA accuracy under standard training setups when more data is available, confirming the method's practical usability and demonstrating that the previously noted accuracy trade-off is solely due to rigorous certification constraints rather than method limitations.
>
> **Thus, the PAC-Bayes loss slots into the ordinary training loop: same optimizer, same epoch budget, no additional hyperparameters. Practitioners can still hold out a test set if they wish—but the *same* training run also yields a high-confidence generalization certificate, and even allows safely folding in test data to tighten the bound without risk of overfitting.**
>
> **Practical Importance of One-Pass PAC-Bayes**
>
> Current practice relies heavily on evaluating models via held-out test sets. While common, this approach struggles in real-world scenarios:
>
> * **Medical imaging/autonomous driving:** labeled data is scarce, making dedicated validation splits expensive.
> * **Safety-critical applications (healthcare/finance):** fixed hold-out data offers no protection against subtle deployment shifts or hidden overfitting.
> * **Resource-constrained environments:** repeated training or multi-pass validation (typical of previous PAC-Bayes methods) is prohibitively costly.
>
> Our one-pass PAC-Bayes method addresses these pain points by seamlessly embedding generalization certification within standard workflows. Practitioners gain rigorous generalization guarantees directly from a single training pass, efficiently protecting against hidden overfitting and shifts in deployment—ideal for safety-critical and resource-constrained applications.
>
> **Theoretical Justification of Localized Prior**
>
> Catoni's localisation starts from the ideal—but unusable—reference prior: if we somehow knew the true risk $R(\theta)$ of each parameter $\theta$, we would create a prior $\pi_{\text{ideal}}(\theta)\propto\pi(\theta)\,e^{-\lambda R(\theta)}$ to assign larger probability to parameters with smaller true risk. Of course, the true risk $R(\theta)$ is unknown, so Catoni's localisation uses the empirical loss $r_S(\theta)$—which we do know from data—to approximate it, defining instead a *softly tilted* prior:
> $$
> \pi_{\text{loc}}(\theta)\propto\pi(\theta)\,e^{-\xi\lambda r_S(\theta)},\quad 0<\xi< 1.
> $$
>
> Because this new prior only partly relies on the empirical data (controlled by the factor $\xi< 1$), it remains broader and more conservative than the posterior, preventing overfitting. Mathematically, the correctness of this substitution (replacing unknown true risk with known empirical risk) is guaranteed by a standard two-step argument: (i) a Bernstein-type concentration inequality controlling how far empirical averages deviate from their true expectations, and (ii) the Donsker–Varadhan identity converting log-moment terms into KL-divergences. After this substitution, every term involving the unknown ideal prior $\pi_{\text{ideal}}$ can be safely replaced by the corresponding term evaluated at the computable prior $\pi_{\text{loc}}$, plus a small explicit penalty term $\frac{\xi}{1-\xi}g(\lambda/N)\lambda/N$. That penalty (appearing as the denominator in our Theorem 1) explicitly compensates for the data-dependence of the prior, thus preserving correctness and high-probability validity while significantly tightening the resulting bound.
>
> **Contributions Beyond Catoni’s Original Work**
>
> Catoni introduced the theoretical concept of "localization," showing in principle how PAC-Bayes bounds could be tightened via a softly-tilted reference prior dependent on empirical data. However, Catoni’s result remained purely theoretical—limited to abstract settings and binary 0–1 loss—without practical algorithms for modern deep networks.
>
> Our contributions fill these critical practical and theoretical gaps by extending Catoni's localization idea to deep learning:
>
> * **Cross-entropy extension:** generalized the bound rigorously to scaled cross-entropy loss, enabling direct gradient-based optimization.
> * **Stable estimation:** developed finite-sample corrected importance-sampling estimators for computationally stable log-partition approximations.
> * **One-pass practical implementation:** provided a lightweight, plug-and-play TensorFlow/Keras decorator integrating directly into standard training loops (no early stopping or extra phases).
> * **Per-prediction certification:** rigorously extended PAC-Bayes bounds to per-example certificates, enabling principled OOD detection.
> * **Wide-network analysis:** rigorously characterized generalization properties for wide neural networks trained with SGD (Theorem 3.2).
>
> These nontrivial advances transform Catoni’s theoretical insight into a practical and scalable generalization certification tool for modern deep learning workflows.
>
> ---
>
> We sincerely appreciate Reviewer 8Egq’s feedback highlighting opportunities to improve the paper’s clarity and accessibility. We will incorporate all clarifications into our manuscript. We share the reviewer's goal of making PAC-Bayes theory understandable and practically useful for a broad audience. We welcome any additional feedback or suggestions from Reviewer 8Egq.

---

> > ### Author Response · Authors · 2025-08-06
> >
> > Thank you again, Reviewer 8Egq, for your thoughtful review—your feedback on clarity and accessibility was particularly helpful. We will incorporate your suggestions and our rebuttal into the final revision. If you have any additional points or questions, we remain available through the end of the discussion period.

---

### Note · Authors · 2025-08-12

We address the lack of practical training-time certificates for deep neural nets. Our **localized, one-pass PAC-Bayes loss** jointly optimizes a softened data-dependent prior with model parameters, yielding tight, non-vacuous certificates under standard SGD.

Key concerns were resolved: uniformization justifies continuous optimization of $(\lambda,\xi)$; bounds are stable for small $K$; the “extra pass” note was an editing artifact. For per-prediction use without clean validation, we note conservative choices (fixed tail bounds; unsupervised quantiles/drift monitors), keeping calibration refinements orthogonal.

Proofs and code are provided. The approach is a drop-in loss with modest overhead in standard deep-learning workflows, bridging the theory-practice gap for deployable PAC-Bayes certification.

---

### Decision · Program_Chairs · 2025-09-17

**Decision:**

Accept (poster)

**Comment:**

This paper explores tighter generation bounds of deep neural networks by leveraging PAC-Bayes bounds along with a novel localized PAC-Bayes prior. The tightness is theoretically supported and empirically validated over diverse datasets and practical models.

**Strengths**:
* exploring a critical problem in learning theory: a non-vacuous generalization bound for deep networks
* proposing computationally efficient "one-pass" approach
* broad empirical evaluation


**Weaknesses**:
* presentation issues – the paper would be more readable by explaining layman's terms.

The paper is borderline but reviewers agree that this paper has sufficient contributions and advances the frontier of PAC-Bayes bounds. So, I vote for acceptance. For the final manuscript, it would be better to refine the paper presentation to make it more accessible for general readers and include valuable discussions from reviewers.